# TOTEM: TOkenized Time Series EMbeddings for General Time Series Analysis

**Sabera Talukder**                                              *sabera@caltech.edu*
**Yisong Yue**                                                     *yyue@caltech.edu*
**Georgia Gkioxari**                                           *georgia@caltech.edu*
*California Institute of Technology*

**Reviewed on OpenReview:** *https://openreview.net/forum?id=QlTLkH6xRC*

## Abstract

This work studies the problem of time series analysis with *generalist* (or foundation) models, which are models trained across many data domains. Drawing inspiration from the widespread success of large language models, we consider the simple strategy of discretely tokenizing time series data drawn from a myriad of datasets via self-supervision, then using the fixed tokenization to solve a variety of tasks across many data domains. Canonically, time series models are either trained on a single dataset or built in a task-specific manner (e.g., a forecasting-only model), where many use patches of time as inputs to the model. As such, performant generalist, discrete representation time series models explored across many tasks are of value. Our method, TOkenized Time Series EMbeddings (TOTEM), produces such generalist time series models with minimal or no fine-tuning while exhibiting strong zero-shot performance. We evaluate TOTEM extensively over nearly 500 experiments on three commonly-studied time series tasks with real-world data: imputation (17 baselines, 12 datasets), anomaly detection (19 baselines, 25 datasets), and forecasting (14 baselines, 12 datasets). We conclude that TOTEM matches or outperforms existing state-of-the-art models in both the canonical specialist setting (i.e., training one model on one domain) as well as the generalist setting (i.e., training a single model on many domains), which demonstrates the efficacy of tokenization for general time series analysis. The open-source implementation is available here: https://github.com/SaberaTalukder/TOTEM; a video summary is available here: https://www.youtube.com/watch?v=OqrCpdb6MJk.

## 1 Introduction

Time series are a fundamental data modality, generalizing large classes of time-varying data from many *domains*, like weather phenomena, electrical grid activity, or traffic flow. Most commonly, time series analysis is first restricted to one such domain, then to a specific *task*, like *imputation* (Luo et al., 2018; 2019; Talukder et al., 2022), *anomaly detection* (Xu et al., 2021; He & Zhao, 2019), or *forecasting* (Wu et al., 2021; Woo et al., 2022), among others. Though these domains and tasks are quite distinct, a natural question is whether it is possible to design domain-agnostic models adaptable to any task. This question is the subject of our work.

*Generalist models* are those trained on many data domains simultaneously (e.g., weather, electricity, traffic, etc.), while *specialist models* are those trained on a single time series domain (e.g., weather only), as shown in Figure 1A (Zhou et al., 2023; Wu et al., 2022; Nie et al., 2022). Both generalist and specialist models can be tested in two ways: *in-domain testing*, where a model is tested on the same domain(s) on which it was trained, and *zero-shot testing*, where it is tested on different domain(s) (see Figure 1B). Performing zero-shot testing on specialist models is not uncommon. For example, some works have studied zero-shot forecasting, where a forecaster trains on one dataset then predicts on a separate dataset (Zhou et al., 2023), or trains on a subset of channels (which we call *sensors*) from one dataset then forecasts zero-shot on the remaining sensors in the

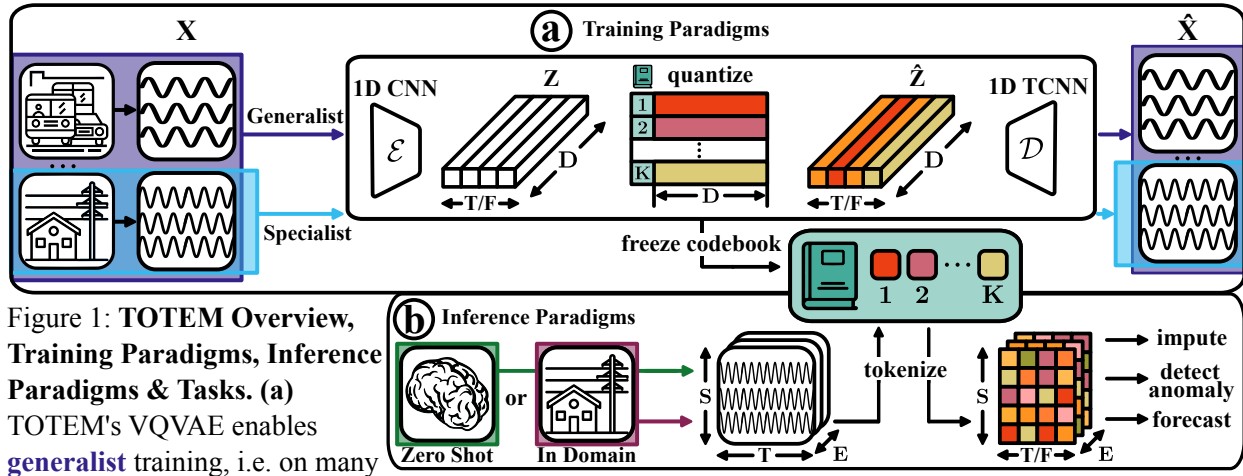

Figure 1: **TOTEM Overview, Training Paradigms, Inference Paradigms & Tasks. (a)** TOTEM's VQVAE enables **generalist** training, i.e. on many data domains jointly, and **specialist** training, i.e. on one data domain at a time. The TOTEM VQVAE architecture consists of a 1D strided CNN encoder $\mathcal{E}$, quantizer, latent codebook, and 1D strided transpose CNN decoder $\mathcal{D}$. (b) TOTEM's discrete, self-supervised codebook is frozen then leveraged for both **in domain** and **zero shot** testing across many tasks.

same dataset (Liu et al., 2023). However, we emphasize that both of the preceding examples are specialists, as they were trained on only one (or a subset of one) dataset. In contrast, our goal in this paper is instead the design of generalist models, which we evaluate in both the in-domain and zero-shot testing regimes.

Not only are most modern time series models specialists, they typically operate over patches (Zhou et al., 2023; Wu et al., 2022; Liu et al., 2023; Zhang & Yan, 2022; Nie et al., 2022; Li et al., 2019; Zhou et al., 2021; Wu et al., 2021) and are trained for only a single task (Das et al., 2023b; Ansari et al., 2024; Liu et al., 2023; Zhang & Yan, 2022; Zhou et al., 2021; Wu et al., 2021). Our core hypothesis is that many of the design decisions in prior works hinder the development of generalist models, and that by adopting practices more commonly used in language (Gage, 1994; Radford et al., 2018) and vision modeling (Van Den Oord et al., 2017; Esser et al., 2021; Rombach et al., 2022) we can boost the generalization performance of resulting time series models. While there exist works that train in an unsupervised manner (Yue et al., 2022; Yang & Hong, 2022; Tonekaboni et al., 2021; Barnum et al., 2020; Franceschi et al., 2019) or use discrete representations (Rabanser et al., 2020b; Van Den Oord et al., 2017; Lin et al., 2007), few works have explored the combination of generalist models and discrete representations over many tasks in a systematic manner (i.e., in both the in-domain and zero-shot testing regimes). Thus, the contributions of our work are twofold.

**TOTEM.** We develop **To**kenized **T**ime Series **Em**beddings (TOTEM), a simple tokenization method for time series data that employs a self-supervised pre-training stage to learn a fixed number of discrete tokens over a multi-domain corpus (Section 3.2). Surprisingly, we demonstrate that TOTEM is effective for solving a variety of downstream tasks in a domain-agnostic manner even though the tokens only encode the shape of univariate waveforms. This allows TOTEM to generically tokenize multivariate data of differing size by simply stacking collections of univariate tokens.

**Comprehensive Experiments.** We test our hypothesis extensively on three distinct tasks, each with their own datasets and baselines: imputation (17 baselines and 12 datasets), anomaly detection (19 baselines and 25 datasets), and forecasting (14 baselines and 12 datasets). We find that in the specialist settings, TOTEM matches or outperforms the performance of most state-of-the-art (SOTA) task-specific models, despite minimal or no task-specific design. Similarly, TOTEM also matches or outperforms SOTA generalist models. We conduct thorough ablations showing that discrete tokens outperform patches and that generalist training improves model performance independent of TOTEM's modeling choices. Our experiments are some of the most extensive in the literature, comprising hundreds of seeded runs (see Sections 5 and 6).

## 2 Related Work

We categorize related works in three ways: whether they (i) study specialists or generalists, (ii) use patched or discrete data representations, and (iii) train and evaluate models for multiple distinct time series tasks. Unlike TOTEM, no prior works study the use of discrete data representations for training generalists across multiple tasks (see Table 1 for a comparison).

**Specialist vs. Generalist Training.** Historically, the specialist (i.e., single-domain) training paradigm is most common amongst time series models (Zhou et al., 2023; Wu et al., 2022; Nie et al., 2022; Zhang & Yan, 2022). These specialist models are primarily evaluated via in-domain testing, where the test set is a held-out set from the same training domain. Some concurrent and subsequent works have begun exploring generalist time series foundation models, including forecasters from Google and Amazon (Das et al., 2023b; Ansari et al., 2024). We compare to the concurrent MOMENT model (Goswami et al., 2024) in limited evaluations (see Tables 13 and 20) as it also studies multiple tasks, and find that TOTEM generally outperforms it.

**Patched vs. Discrete Data Representations.** In order to pass time series data to a downstream model, it is necessary to choose some latent data representation. As in ViTs (Dosovitskiy et al., 2020), the prevailing strategy is to *patch* time series data, either temporally (Liu et al., 2023; Zhang & Yan, 2022; Nie et al., 2022) or spatially Li et al. (2019); Zhou et al. (2021); Wu et al. (2021), then to linearly project the patches to some latent embedding on which a model like a transformer or MLP can operate. We emphasize that patched representations are *dynamic* in the sense that the embedding associated with the patch is determined entirely by the layers in the downstream model which project the patches to the embedding space. Therefore, patched representations are trained end to end.

Patching is fundamentally at odds with tokenization, wherein a fixed "vocabulary" of embeddings is determined *before* training the downstream model, which then operates on the fixed, tokenized representations. Tokenization (learned or otherwise) has been leveraged for training models in fields like language and vision modeling (Gage, 1994; Radford et al., 2018; Van Den Oord et al., 2017; Esser et al., 2021; Rombach et al., 2022). Some prior work in time series modeling has explored discrete representations using binning (Rabanser et al., 2020b;a; Lin et al., 2007) or quantization (Baevski et al., 2020; Van Den Oord et al., 2017; Oord et al., 2016) in domain- or task-specific ways. Inspired by the success of vector quantized variational autoencoders (VQVAEs) in both audio and vision (Van Den Oord et al., 2017; Esser et al., 2021; Rombach et al., 2022), we build on these works by showing that the VQVAE is also effective for learning discrete representations for general time series modeling.

| | | Generalist Training | Discrete Tokenization | Multiple Tasks |
|---|---|:---:|:---:|:---:|
| Prior | GPT2 (Zhou et al., 2023) | ✗ | ✗ | ✓ |
| | TiNet (Wu et al., 2022) | ✗ | ✗ | ✓ |
| | W2V2.0 (Baevski et al., 2020) | ✗ | ✓ | ✗ |
| | SAX (Lin et al., 2007) | ✗ | ✓ | ✓ |
| C/S | TimesFM (Das et al., 2023b) | ✓ | ✗ | ✗ |
| | Chronos (Ansari et al., 2024) | ✓ | ✓ | ✗ |
| | MOMENT (Goswami et al., 2024) | ✓ | ✗ | ✓ |
| | **TOTEM (Ours)** | ✓ | ✓ | ✓ |

Table 1: **Related Work Overview.** TOTEM is designed for generalist training using discrete tokenization for any task. No prior and concurrent/subsequent (C/S) works study all three at once.

**Time Series Tasks.** Prior works on time series modeling study a variety of tasks, like forecasting, anomaly detection, imputation, and classification. Many prior and concurrent works focus on a single task (Zhang & Yan, 2022; Nie et al., 2022; Xu et al., 2021; Ansari et al., 2024; Das et al., 2023b), with a few exploring multiple specialist trained models on many tasks (Zhou et al., 2023; Wu et al., 2022). TOTEM is most closely related to concurrent works like MOMENT (Goswami et al., 2024), which are focused on generalist models which are effective on any one of the above tasks. For detail on each task, see Sections 3 and 4.

# 3 Method

## 3.1 Task Definitions

This work considers three tasks: imputation, anomaly detection, and forecasting. In *imputation*, models intake a masked time series $\mathbf{x_m} \in \mathbb{R}^{S \times T_{\text{in}}}$ and impute the missing values to recover the reconstruction $\hat{\mathbf{x}} \in \mathbb{R}^{S \times T_{\text{in}}}$. In *anomaly detection*, models intake a time series corrupted at a known level $\mathbf{x_{corr}} \in \mathbb{R}^{S \times T_{\text{in}}}$ and predict which times are anomalous, $\mathbf{y} \in \{0, 1\}^{T_{\text{in}}}$. Lastly, in *forecasting*, models intake a time series $\mathbf{x} \in \mathbb{R}^{S \times T_{\text{in}}}$ and predict future values $\mathbf{y} \in \mathbb{R}^{S \times T_{\text{out}}}$, where $T_{\text{in}}$ and $T_{\text{out}}$ signify the durations of the preceding and succeeding time series, respectively. A core design goal for TOTEM is to learn a representation suitable for any of these three tasks using the same architecture and without leveraging any task- or domain-specific knowledge.

## 3.2 Design Decisions

This section discusses TOTEM's key design features: a self-supervised training stage, exclusively-temporal tokenization, and no domain-specific data engineering.

**Self-supervised Tokenizer Training.** As described in Section 2, TOTEM learns a fixed codebook of tokens over a multi-domain corpus of time series data independently from the training of any downstream model. This disentangles the choice of data representation from the choice of task-specific architecture and permits the learning of representations from a large, diverse set of data, which aids in zero-shot generalization.

First, we elect to use a discrete, deterministic encoder to produce time series tokens. This decision is largely motivated by large language models (and in particular, tokenization methods in NLP like byte pair encoding (BPE) (Gage, 1994; Radford et al., 2018)), in which a downstream model learns on a finite number of distinct tokens. Moreover, in methods like BPE, the tokenization operation is lossless and reversible because it is deterministic (though non-unique). This suggests that vector quantization-based models could be effective for tokenizing time series data. Two popular vector quantization methods are VQVAEs (Van Den Oord et al., 2017) and VQGANs (Esser et al., 2021). In this work, we choose to use a VQVAE, as VQGANs are more commonly used for encoding images. Moreover, the use of VQVAEs has been studied in neural audio models (Oord et al., 2016; Van Den Oord et al., 2017), including followup works with audio-specific models (Baevski et al., 2020), which suggests that they may be effective for modeling general time series.

**Exclusively-Temporal Tokenization.** A time series dataset consists of $E$ examples, $S$ sensor channels, and $T$ time steps, and can be formally expressed as $\{\mathbf{x}_j\}_{j=1}^{E} \subset \mathbb{R}^{S \times T}$. Prior work commonly patches along either the sensor dimension (Li et al., 2019; Zhou et al., 2021; Wu et al., 2021; Liu et al., 2021), or time dimension (Liu et al., 2023; Zhang & Yan, 2022; Nie et al., 2022). When training specialists, it is reasonable to tokenize across any combination of these or the example dimension (e.g., in neuroscience data, it is common to group recordings by day, where the subject exhibits different behavior on a daily basis (Talukder et al., 2022)).

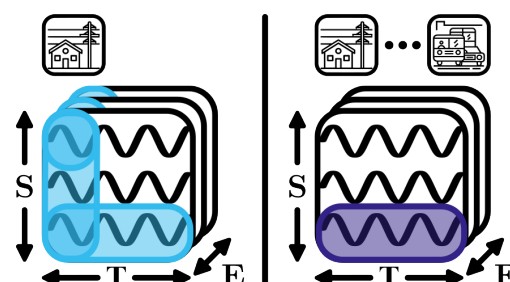

Figure 2: **Left. Specialist** models can tokenize along any of the $E$, $S$, or $T$ dimensions. **Right. Generalist** models can only tokenize along $T$, since the learned tokenization must apply to a diverse set of domains with any possible data dimensionality.

However, in the generalist case, because the sensors associated with each domain have distinct semantic meanings, performing sensor- or example-wise tokenization will capture domain-specific relations, hindering the tokenizer's generalization. Thus, we choose to exclusively tokenize over the temporal dimension, such that the tokens represent univariate waveforms. Further, this is crucial for testing the tokenizer in the zero-shot regime, where the dimensionality of the testing domain may differ significantly from that of the training domain(s). Specifically, TOTEM tokenizes time series data with non-overlapping temporal chunks of length $T/F$, where $F$ is some compression factor for downsampling the data.

**No Domain-specific Data Engineering.** Many prior works (especially in time series forecasting) leverage domain-specific knowledge to handcraft features that encode critical information. For instance, works that study calendar-based time series often add auxiliary features that denote landmarks like the first day of a month or holidays (Chen et al., 2023; Salinas et al., 2020). Other works propose highly-engineered architectures that convert time series into frequency space representations. For example, TimesNet operates on the assumption that most time series exhibit multi-resolutional periodicity, and convert a time series into a frequency-space image by computing the Fourier transform on several subsets of the time series (Wu et al., 2022). Similarly, FedFormer represents a time series with a random subset of its Fourier components and a complex mixture-of-experts model (Zhou et al., 2022). In contrast, TOTEM uses only reverse instance normalization (RevIN) (Kim et al., 2021) to represent temporal waveforms in a normalized space (see Figure 3), which requires no assumptions on the form of the data. This allows TOTEM to generalize across domains and outperform the prior handcrafted methods on many distinct tasks using simple, generic architectures.

### 3.3  Tokenizer Implementation

Though TOTEM is a VQVAE, the design of the encoder and decoder differ substantially from the original model and similar works like WaveNet, which use dilated convolutions (Oord et al., 2016; Van Den Oord et al., 2017). The dilations in these architectures skip many time steps, allowing the convolutional filters to operate on a larger input area at a coarser scale, improving model efficiency. However, this design decision is motivated by the high sampling rates of digital audio waveforms,

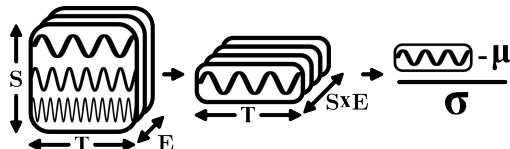

Figure 3: TOTEM flattens the sensor and example dimensions and learns a discrete representation along the time dimension in a normalized space.

which is not a universal trait across time series domains (see Table 6). In contrast, TOTEM uses a stack of strided 1D convolutions with a dilation of 1 such that it can account for every time step. Using a long input (e.g., 96 time steps for standard forecasting tasks) allows TOTEM to maintain a large receptive field. Lastly, the use of RevIN allows TOTEM to remain effective by only learning a small set of normalized waveforms, and if the unnormalized reconstruction is required for a downstream task, the normalization parameters can also be passed to the decoder (see Figure 4).

Formally, TOTEM accepts a batch of univariate time series $\{\mathbf{x}_i \in \mathbb{R}^T\}_{i=1}^{E \cdot S}$ obtained by flattening the sensor channel of the multivariate data. An encoder $\mathcal{E}$ consisting of a stack of strided 1D convolutions then temporally compresses the data by a total factor of $F$ to recover a latent variable $\mathbf{z} = \mathcal{E}(\mathbf{x}) \in \mathbb{R}^{T/F \times D}$, where $D$ is the latent feature dimension. The latent variable $\mathbf{z}$ is then quantized into an element $\hat{\mathbf{z}}$ of the codebook $\mathcal{C} = \{\mathbf{c}_i\}_{i=1}^K$ consisting of $K$ $D$-dimensional codewords $\mathbf{c}_i \in \mathbb{R}^D$ following the relation $\hat{\mathbf{z}} = \mathbf{c}_\ell$, where $\ell = \arg\min_i ||\mathbf{z} - c_i||_2^2$. The decoder $\mathcal{D}$ mirrors the encoder's architecture, mapping the quantized embedding $\hat{\mathbf{z}}$ to a reconstructed time series $\hat{\mathbf{x}} = \mathcal{D}(\hat{\mathbf{z}}) \in \mathbb{R}^T$.

As in Van Den Oord et al. (2017), we train $\mathcal{E}$, $\mathcal{D}$, and $\mathcal{C}$ by optimizing the objective

$$\mathcal{L} = \underbrace{\frac{1}{E \cdot S} \sum_i ||\mathbf{x}_i - \hat{\mathbf{x}}_i||_2^2}_{\mathcal{L}_{\text{rec}}} + \underbrace{||\mathbf{sg}[\mathbf{z}] - \hat{\mathbf{z}}||_2^2}_{\mathcal{L}_{\text{vq}}} + \beta \underbrace{||\mathbf{z} - \mathbf{sg}[\hat{\mathbf{z}}]||_2^2}_{\mathcal{L}_{\text{cmt}}}, \tag{1}$$

where $\mathbf{sg}[\cdot]$ is the stop-gradient operator and $\beta$ is the commitment loss weight. For additional details, see Appendices A.11 and A.12. In all experiments, we use a compression factor of $F = 4$, (see Table 32).

### 3.4  Forecasting Model Implementation

In contrast with prior works, **TOTEM is capable of solving the imputation and anomaly detection tasks with the tokenizer alone** (see Figures 11 and 12). Therefore, the only downstream model we must design is the forecasting model. First, each sensor's observations $\mathbf{x}_i \in \mathbb{R}^{T_{\text{in}}}$ are converted into a sequence of $T_{\text{in}}/F$ discrete tokens $\hat{\mathbf{z}}_i$. The forecaster processes adds temporal positional embeddings to these tokens, passing them through a transformer encoder consisting of a series of multi-head attention layers that attend along the time dimension to predict normalized measurements $\bar{\mathbf{y}}_i \in \mathbb{R}^{T_{\text{out}}}$ for $i = 1, ..., S$.

A separate prediction head predicts the mean $\mu_i$ and standard deviation $\sigma_i$ associated with each univariate time series $\mathbf{x}_i$ such that the final forecasted prediction is $\mathbf{y}_i = \sigma_i \cdot \bar{\mathbf{y}}_i + \mu_i$ for $i = 1, \ldots, S$. The forecaster is trained in a supervised fashion by minimizing three smooth L1 losses between predictions $\{\bar{\mathbf{y}}_i, \mu_i, \sigma_i\}_{i=1}^S$ and their ground truth values respectively. Crucially, this architecture is used for *all* domains in our forecasting experiments, demonstrating that TOTEM can competitively perform forecasting in a domain-agnostic manner.

## 4 Experimental Setup

This section explains the experimental setup for each task, including the baselines and datasets used for evaluation. The results and analyses are presented in Section 5. We compare to two families of approaches: methods designed for multiple tasks (**multi-task**),

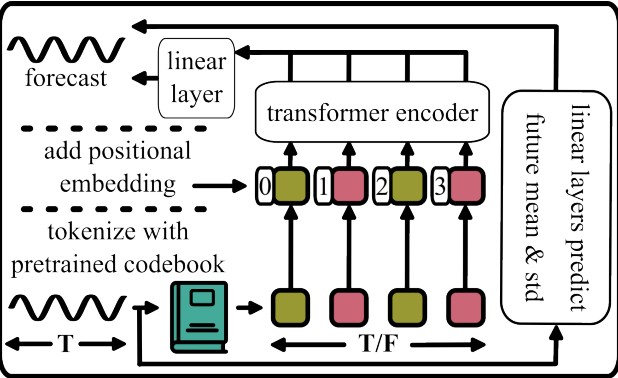

Figure 4: **The Forecaster Model.** The forecaster takes in a tokenized version of normalized time series observations (obtained using TOTEM's encoder) and predicts a normalized time series over some specified horizon along with parameters that allow the model to unnormalize the prediction.

like TOTEM, and methods designed for a specific task (**single-task**). Many single-task methods have frequently been adapted by others to tasks besides the ones for which they were originally designed, and in those cases, we compare against the best reported results for the adapted model. For all tasks, we trained a GPT2 generalist baseline from scratch, and for forecasting, we additionally trained a GPT2 specialist.

### 4.1 Imputation

**Baselines.** In the main text, we compare TOTEM against 12 baselines with varying model architectures. We further compare against 5 additional baselines with different architectures for completeness in the Appendix A.2. *In total, we evaluate against 17 baselines.* See Table 2A for a summary.

**Datasets.** For the in-domain testing regime, we test on 6 datasets, and for the zero-shot testing regime, we evaluate on an additional 5 datasets. We also perform additional evaluations in the appendix on the PhysioNet Challenge 2012 dataset. *In total, we evaluate on 12 distinct datasets.* See Table 2B for a summary.

**Metrics.** We report the mean squared error (`MSE`) and mean absolute error (`MAE`) of the imputed versus the ground truth signals.

### A. Imputation Baselines

| Type | Model | Abbr. | Arch. | Citation |
|------|-------|-------|-------|----------|
| Multi-task | GPT2 - Generalist | GPT2 | TF | Trained by us |
| | GPT2 - Specialist | GPT2 | TF | Zhou et al. (2023) |
| | TimesNet | TiNet | Conv | Wu et al. (2022) |
| Single-task | PatchTST | Patch | TF | Nie et al. (2022) |
| | ETSFormer | ETS | TF | Woo et al. (2022) |
| | Fedformer | FED | TF | Zhou et al. (2022) |
| | Non-stationary Trans. | Stat | TF | Liu et al. (2022b) |
| | Autoformer | Auto | TF | Wu et al. (2021) |
| | Informer | Inf | TF | Zhou et al. (2021) |
| | Reformer | Re | TF | Kitaev et al. (2020) |
| | LightTS | LiTS | Linear | Zhang et al. (2022) |
| | DLinear | DLin | Linear | Zeng et al. (2023) |
| Appendix | V-RIN | - | VAE | Mulyadi et al. (2021) |
| | BRITS | - | RNN | Cao et al. (2018) |
| | RDIS | - | Diffusion | Choi et al. (2023) |
| | Unconditional CSDI | - | Diffusion | Tashiro et al. (2021) |
| | CSDI | - | Diffusion | Tashiro et al. (2021) |

### B. Imputation Datasets

| Regime | Dataset | Abbr. | Citation |
|--------|---------|-------|----------|
| In-Domain | Weather | W | Zhou et al. (2023) |
| | Electricity | E | Zhou et al. (2023) |
| | ETTm1 | m1 | Zhou et al. (2023) |
| | ETTm2 | m2 | Zhou et al. (2023) |
| | ETTh1 | h1 | Zhou et al. (2023) |
| | ETTh2 | h2 | Zhou et al. (2023) |
| Zero-Shot | Neuro2 | N2 | Peterson et al. (2022) |
| | Neuro5 | N5 | Peterson et al. (2022) |
| | Saugeen River Flow | R | Godahewa et al. (2021) |
| | U.S. Births | B | Godahewa et al. (2021) |
| | Sunspot | S | Godahewa et al. (2021) |
| Appendix | PhysioNet | - | Silva et al. (2012) |

Table 2: Imputation baselines and datasets.

## 4.2 Anomaly Detection

**Baselines.** In the main text, we compare TOTEM against 16 baselines, and in the Appendix A.3 an additional 3 for a *total of 19 baselines* (see Table 13). See Table 3A for a summary.

**Datasets.** For the in-domain testing regime, we test on 5 datasets, and for the zero-shot regime, we test on another 5. For additional signal, we also test on 15 distinct anomaly detection datasets from Wu & Keogh (2021) in Appendix A.3 (see Table 13). *In total, we evaluate on 25 datasets.* See Table 3B for a summary.

**Metrics.** We report the precision (`P`), recall (`MSE`), and adjusted `F1` score.

**A. Anomaly Detection Baselines**

| Type | Model | Abbr. | Arch. | Citation |
|---|---|---|---|---|
| Multi-task | GPT2 - Generalist | GPT2 | TF | Trained by us |
| | GPT2 - Specialist | GPT2 | TF | Zhou et al. (2023) |
| | TimesNet | TiNet | Conv | Wu et al. (2022) |
| Single-task | Anomaly Trans. | ATran | TF | Xu et al. (2021) |
| | PatchTST | Patch | TF | Nie et al. (2022) |
| | ETSFormer | ETS | TF | Woo et al. (2022) |
| | Fedformer | FED | TF | Zhou et al. (2022) |
| | Non-stationary Trans. | Stat | TF | Liu et al. (2022b) |
| | Autoformer | Auto | TF | Wu et al. (2021) |
| | Pyraformer | Pyra | TF | Liu et al. (2021) |
| | Informer | Inf | TF | Zhou et al. (2021) |
| | Reformer | Re | TF | Kitaev et al. (2020) |
| | LogTrans | LogTr | TF | Li et al. (2019) |
| | Transformer | Trans | TF | Vaswani et al. (2017) |
| | LightTS | LiTS | Linear | Zhang et al. (2022) |
| | DLinear | DLin | Linear | Zeng et al. (2023) |
| Appendix | DGHL | - | - | Challu et al. (2022) |
| | MOMENT-0 | - | - | Goswami et al. (2024) |
| | MOMENT-LP | - | - | Goswami et al. (2024) |

**B. Anomaly Detection Datasets**

| Regime | Dataset | Abbr. | Citation |
|---|---|---|---|
| In-Domain | SMD | - | Zhou et al. (2023) |
| | MSL | - | Zhou et al. (2023) |
| | SMAP | - | Zhou et al. (2023) |
| | SWAT | - | Zhou et al. (2023) |
| | PSM | - | Zhou et al. (2023) |
| Zero-Shot | Neuro2 | N2 | Peterson et al. (2022) |
| | Neuro5 | N5 | Peterson et al. (2022) |
| | Saugeen River Flow | R | Godahewa et al. (2021) |
| | U.S. Births | B | Godahewa et al. (2021) |
| | Sunspot | S | Godahewa et al. (2021) |
| Appendix | 15 Wu et al. Datasets | - | Wu & Keogh (2021) |

Table 3: Anomaly detection baselines and datasets.

## 4.3 Forecasting

**Baselines.** In the main text, we compare against 12 baselines, with an additional 2 in Appendix A.4 (see Table 20). For the GPT2 specialist that we trained from scratch, we choose a lookback length of 96 for fair comparison with the other models in this paper. *In total, we have 14 baselines.* See Table 4A for a summary.

**Datasets.** For the in-domain testing regime, we test on 7 datasets, and for the zero-shot regime, we test on an additional 5. *In total, we evaluate on 12 datasets.* See Table 4B for a summary.

**Metrics.** We report the mean squared error (`MSE`) and mean absolute error (`MAE`) of the predicted versus the true forecast values.

**A. Forecasting Baselines**

| Type | Model | Abbr. | Arch. | Citation |
|---|---|---|---|---|
| Multi-task | GPT2 - Generalist | GPT2 | TF | Trained by us |
| | GPT2 - Specialist | GPT2 | TF | Trained by us w/ 96 lookback length |
| | TimesNet | TiNet | Conv | Wu et al. (2022) |
| Single-task | iTransformer | iTrans | TF | Liu et al. (2023) |
| | PatchTST | Patch | TF | Nie et al. (2022) |
| | Crossformer | Cross | TF | Zhang & Yan (2022) |
| | Fedformer | FED | TF | Zhou et al. (2022) |
| | Non-stationary Trans. | Stat | TF | Liu et al. (2022b) |
| | TiDE | TiDE | - | Das et al. (2023a) |
| | RLinear | RLin | Linear | Li et al. (2023) |
| | DLinear | DLin | Linear | Zeng et al. (2023) |
| | SciNet | SCi | - | Liu et al. (2022a) |
| Appendix | N-Beats | - | - | Oreshkin et al. (2019) |
| | MOMENT | - | - | Goswami et al. (2024) |

**B. Forecasting Datasets**

| Regime | Dataset | Abbr. | Citation |
|---|---|---|---|
| In-Domain | Weather | W | Liu et al. (2023) |
| | Electricity | E | Liu et al. (2023) |
| | Traffic | T | Liu et al. (2023) |
| | ETTm1 | m1 | Liu et al. (2023) |
| | ETTm2 | m2 | Liu et al. (2023) |
| | ETTh1 | h1 | Liu et al. (2023) |
| | ETTh2 | h2 | Liu et al. (2023) |
| Zero-Shot | Neuro2 | N2 | Peterson et al. (2022) |
| | Neuro5 | N5 | Peterson et al. (2022) |
| | Saugeen River Flow | R | Godahewa et al. (2021) |
| | U.S. Births | B | Godahewa et al. (2021) |
| | Sunspot | S | Godahewa et al. (2021) |

Table 4: Forecasting baselines and datasets.

### 4.4 Task Selection

In the time series literature, there are five canonically studied tasks: imputation, anomaly detection, short- and long-term forecasting, and classification. In this work, we study imputation, anomaly detection, and long-term forecasting. We exclude short-term forecasting and classification for the following reasons.

**Non-Standardized Baselines.** The *long-term forecasting* task uses standardized input and output lengths across all datasets (in particular an input length of 96 timesteps and output lengths of 96, 192, 336, and 720 timesteps), as enforced by a large body of existing work Liu et al. (2023); Wu et al. (2022); Liu et al. (2022b); Zhou et al. (2022) among others[2]. This allows us to fairly baseline TOTEM without rerunning thousands of experiments on dozens of models trained from scratch.

In contrast, the *short-term forecasting* task typically uses non-standard and dataset-specific input and output dimensionalities (see Table 19 for details), which makes systematic, fair comparisons of TOTEM against prior works extremely challenging in the generalist setting[3]. Thus, we exclude short-term forecasting from our main results.

**Leaky Baselines.** In both classification and anomaly detection, the modern SOTA baselines are leaky (Zhou et al., 2023; Wu et al., 2022; Xu et al., 2021), where leakage is defined as using the test set as the validation set during training. In particular, the cited works that report SOTA results all use models that were trained with either early stopping or with the best model checkpoint on the validation (i.e., the test) set. We felt strongly that we should not propagate faulty baselines, so we did not compare to these models in our work. Subsequent to the initial release of this paper, followup works have demonstrated on neural classification tasks that TOTEM, when compared to baselines trained in a non-leaky manner, achieves SOTA performance (Chau et al., 2024a;b).

For anomaly detection, the benchmark datasets used by Zhou et al. (2023); Wu et al. (2022); Xu et al. (2021) contain numerous flaws besides training leakage flawed (see Wu & Keogh (2021) for a detailed account). However, since Wu & Keogh (2021) released a large set of new, unflawed benchmarks, we elected to compare TOTEM to both the flawed and a subset of the unflawed baselines (see the comparisons to Wu & Keogh (2021) in the Appendix). Because we find that TOTEM convincingly achieves SOTA performance in both cases, we report our results to establish an unflawed baseline for future comparison.

In summary, due to non-standardized and leaky baselines, we only report systematic results on the imputation, anomaly detection, and long-term forecasting tasks.

## 5 Main Results

The primary goal of our experiments is to systematically evaluate TOTEM on multiple tasks simultaneously against new generalist benchmarks and strong specialist baselines (i.e., models trained on data from many domains versus one domain). In particular, for each task, we report evaluations against (i) specialists on the in-domain testing regime, (ii) generalists on the in-domain regime, and (iii) generalists on the zero-shot regime. We emphasize that no domain, sampling rate, or sensor dimension is shared between the training sets and zero-shot testing sets (see Table 6 for additional dataset details).

Throughout the main text, we report summary results. The full numerical results can be found throughout the Appendix. Moreover, all results are reported as the mean of 3 seeded runs, with standard deviations available in the Appendix. Since evaluation metrics differ across tasks, ($\downarrow$) will denote a metric where lower is better and ($\uparrow$) will denote a metric where higher is better. Given the varied metrics, we calculate the average number of best results, or `AvgWins`, for each method and highlight the **best** method. For a summary of training and testing domains, see Table 7; for a comparison of generalist parameter counts and training times, see Section A.8; for additional architecture and training details, see Sections A.11 and A.12.

---

[2]Some methods utilize a 512-dimensional input, which makes consistent comparisons challenging; despite this field-wide inconsistency, we include some of these results in Appendix 20. TOTEM outperforms other methods across lookback lengths $96, 512$ at 58.3% `AvgWins` , while the next best model is GPT2 at 8.3% `AvgWins` .

[3]Despite this, we show that TOTEM and GPT2 outperform all other methods on a small set of short term forecasting lengths and datasets in Appendix 18.

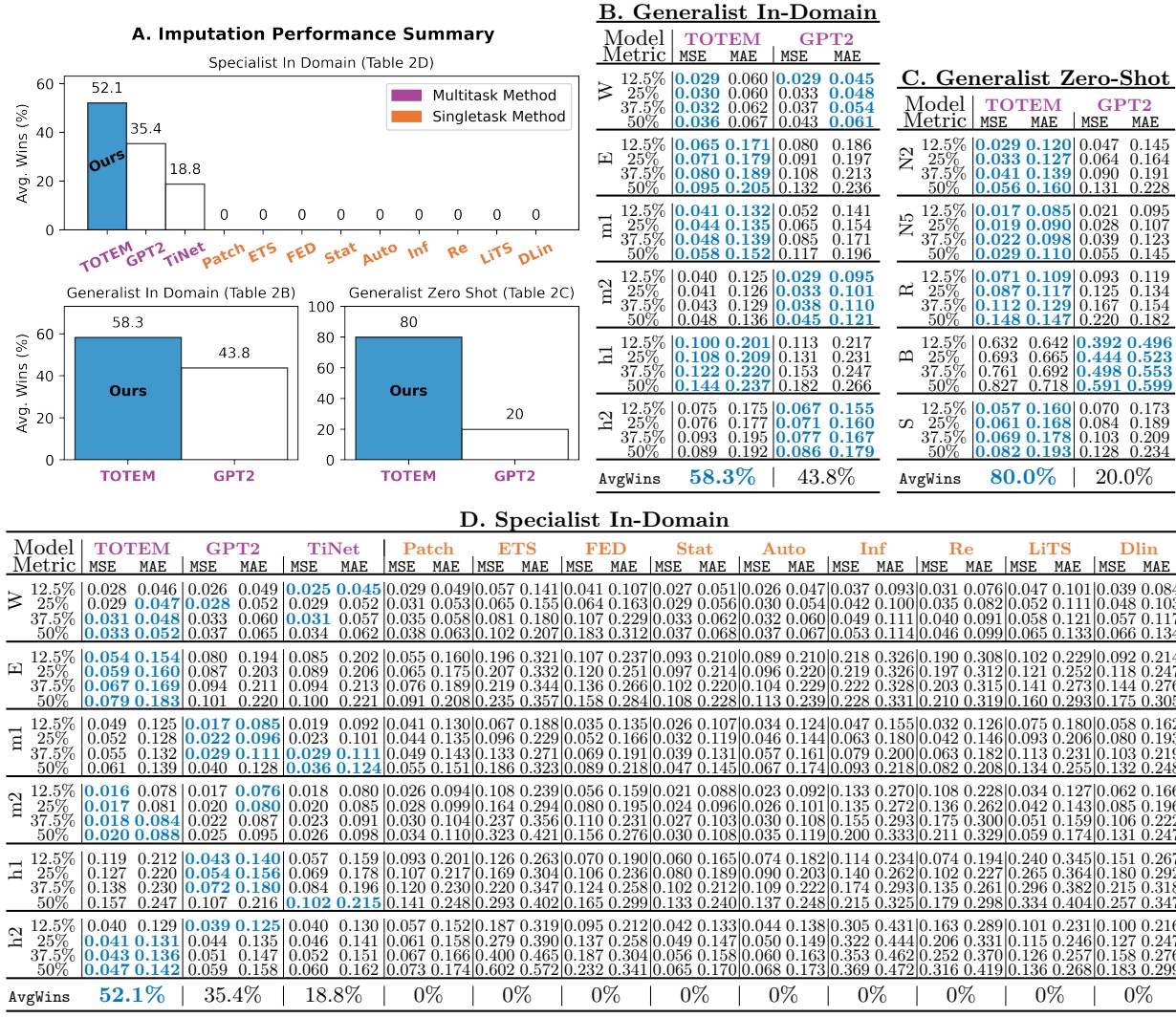

**A. Imputation Performance Summary**

**B. Generalist In-Domain**

| Model | TOTEM | | GPT2 | |
|---|---|---|---|---|
| Metric | MSE | MAE | MSE | MAE |
| **W** 12.5% | **0.029** | 0.060 | **0.029** | **0.045** |
| 25% | **0.030** | 0.060 | 0.033 | **0.048** |
| 37.5% | **0.032** | 0.062 | 0.037 | **0.054** |
| 50% | **0.036** | 0.067 | 0.043 | **0.061** |
| **E** 12.5% | **0.065** | **0.171** | 0.080 | 0.186 |
| 25% | **0.071** | **0.179** | 0.091 | 0.197 |
| 37.5% | **0.080** | **0.189** | 0.108 | 0.213 |
| 50% | **0.095** | **0.205** | 0.132 | 0.236 |
| **m1** 12.5% | **0.041** | **0.132** | 0.052 | 0.141 |
| 25% | **0.044** | **0.135** | 0.065 | 0.154 |
| 37.5% | **0.048** | **0.139** | 0.085 | 0.171 |
| 50% | **0.058** | **0.152** | 0.117 | 0.196 |
| **m2** 12.5% | 0.040 | 0.125 | **0.029** | **0.095** |
| 25% | 0.041 | 0.126 | **0.033** | **0.101** |
| 37.5% | 0.043 | 0.129 | **0.038** | **0.110** |
| 50% | 0.048 | 0.136 | **0.045** | **0.121** |
| **h1** 12.5% | **0.100** | **0.201** | 0.113 | 0.217 |
| 25% | **0.108** | **0.209** | 0.131 | 0.231 |
| 37.5% | **0.122** | **0.220** | 0.153 | 0.247 |
| 50% | **0.144** | **0.237** | 0.182 | 0.266 |
| **h2** 12.5% | 0.075 | 0.175 | **0.067** | **0.155** |
| 25% | 0.076 | 0.177 | **0.071** | **0.160** |
| 37.5% | 0.093 | 0.195 | **0.077** | **0.167** |
| 50% | 0.089 | 0.192 | **0.086** | **0.179** |
| **AvgWins** | **58.3%** | | 43.8% | |

**C. Generalist Zero-Shot**

| Model | TOTEM | | GPT2 | |
|---|---|---|---|---|
| Metric | MSE | MAE | MSE | MAE |
| **N2** 12.5% | **0.029** | **0.120** | 0.047 | 0.145 |
| 25% | **0.033** | **0.127** | 0.064 | 0.164 |
| 37.5% | **0.041** | **0.139** | 0.090 | 0.191 |
| 50% | **0.056** | **0.160** | 0.131 | 0.228 |
| **N5** 12.5% | **0.017** | **0.085** | 0.021 | 0.095 |
| 25% | **0.019** | **0.090** | 0.028 | 0.107 |
| 37.5% | **0.022** | **0.098** | 0.039 | 0.123 |
| 50% | **0.029** | **0.110** | 0.055 | 0.145 |
| **R** 12.5% | **0.071** | **0.109** | 0.093 | 0.119 |
| 25% | **0.087** | **0.117** | 0.125 | 0.134 |
| 37.5% | **0.112** | **0.129** | 0.167 | 0.154 |
| 50% | **0.148** | **0.147** | 0.220 | 0.182 |
| **B** 12.5% | 0.632 | 0.642 | **0.392** | **0.496** |
| 25% | 0.693 | 0.665 | **0.444** | **0.523** |
| 37.5% | 0.761 | 0.692 | **0.498** | **0.553** |
| 50% | 0.827 | 0.718 | **0.591** | **0.599** |
| **S** 12.5% | **0.057** | **0.160** | 0.070 | 0.173 |
| 25% | **0.061** | **0.168** | 0.084 | 0.189 |
| 37.5% | **0.069** | **0.178** | 0.103 | 0.209 |
| 50% | **0.082** | **0.193** | 0.128 | 0.234 |
| **AvgWins** | **80.0%** | | 20.0% | |

**D. Specialist In-Domain**

| Model | TOTEM | | GPT2 | | TiNet | | Patch | | ETS | | FED | | Stat | | Auto | | Inf | | Re | | LiTS | | Dlin | |
|---|---|---|---|---|---|---|---|---|---|---|---|---|---|---|---|---|---|---|---|---|---|---|---|---|
| Metric | MSE | MAE | MSE | MAE | MSE | MAE | MSE | MAE | MSE | MAE | MSE | MAE | MSE | MAE | MSE | MAE | MSE | MAE | MSE | MAE | MSE | MAE | MSE | MAE |
| **W** 12.5% | 0.028 | 0.046 | 0.026 | 0.049 | **0.025** | **0.045** | 0.029 | 0.049 | 0.057 | 0.141 | 0.041 | 0.107 | 0.027 | 0.051 | 0.026 | 0.047 | 0.037 | 0.093 | 0.031 | 0.076 | 0.047 | 0.101 | 0.039 | 0.084 |
| 25% | 0.029 | **0.047** | **0.028** | 0.052 | 0.029 | 0.052 | 0.031 | 0.053 | 0.065 | 0.155 | 0.064 | 0.163 | 0.029 | 0.056 | 0.030 | 0.054 | 0.042 | 0.100 | 0.035 | 0.082 | 0.052 | 0.111 | 0.048 | 0.103 |
| 37.5% | **0.031** | **0.048** | 0.033 | 0.060 | **0.031** | 0.057 | 0.035 | 0.058 | 0.081 | 0.180 | 0.107 | 0.229 | 0.033 | 0.062 | 0.032 | 0.060 | 0.049 | 0.111 | 0.040 | 0.091 | 0.058 | 0.121 | 0.057 | 0.117 |
| 50% | **0.033** | **0.052** | 0.037 | 0.065 | 0.034 | 0.062 | 0.038 | 0.063 | 0.102 | 0.207 | 0.183 | 0.312 | 0.037 | 0.068 | 0.037 | 0.067 | 0.053 | 0.114 | 0.046 | 0.099 | 0.065 | 0.133 | 0.066 | 0.134 |
| **E** 12.5% | **0.054** | **0.154** | 0.080 | 0.194 | 0.085 | 0.202 | 0.055 | 0.160 | 0.196 | 0.321 | 0.107 | 0.237 | 0.093 | 0.210 | 0.089 | 0.210 | 0.218 | 0.326 | 0.190 | 0.308 | 0.102 | 0.229 | 0.092 | 0.214 |
| 25% | **0.059** | **0.160** | 0.087 | 0.203 | 0.089 | 0.206 | 0.065 | 0.175 | 0.207 | 0.332 | 0.120 | 0.251 | 0.097 | 0.214 | 0.096 | 0.220 | 0.219 | 0.326 | 0.197 | 0.312 | 0.121 | 0.252 | 0.118 | 0.247 |
| 37.5% | **0.067** | **0.169** | 0.094 | 0.211 | 0.094 | 0.213 | 0.076 | 0.189 | 0.219 | 0.344 | 0.136 | 0.266 | 0.102 | 0.220 | 0.104 | 0.229 | 0.222 | 0.328 | 0.203 | 0.315 | 0.141 | 0.273 | 0.144 | 0.276 |
| 50% | **0.079** | **0.183** | 0.101 | 0.220 | 0.100 | 0.221 | 0.091 | 0.208 | 0.235 | 0.357 | 0.158 | 0.284 | 0.108 | 0.228 | 0.113 | 0.239 | 0.228 | 0.331 | 0.210 | 0.319 | 0.160 | 0.293 | 0.175 | 0.305 |
| **m1** 12.5% | 0.049 | 0.125 | **0.017** | **0.085** | 0.019 | 0.092 | 0.041 | 0.130 | 0.067 | 0.188 | 0.035 | 0.135 | 0.026 | 0.107 | 0.034 | 0.124 | 0.047 | 0.155 | 0.032 | 0.126 | 0.075 | 0.180 | 0.058 | 0.162 |
| 25% | 0.052 | 0.128 | **0.022** | **0.096** | 0.023 | 0.101 | 0.044 | 0.135 | 0.096 | 0.229 | 0.052 | 0.166 | 0.032 | 0.119 | 0.046 | 0.144 | 0.063 | 0.180 | 0.042 | 0.146 | 0.093 | 0.206 | 0.080 | 0.193 |
| 37.5% | 0.055 | 0.132 | **0.029** | **0.111** | 0.029 | 0.111 | 0.049 | 0.143 | 0.133 | 0.271 | 0.069 | 0.191 | 0.039 | 0.131 | 0.057 | 0.161 | 0.079 | 0.200 | 0.063 | 0.182 | 0.113 | 0.231 | 0.103 | 0.219 |
| 50% | 0.061 | 0.139 | 0.040 | 0.128 | **0.036** | **0.124** | 0.055 | 0.151 | 0.186 | 0.323 | 0.089 | 0.218 | 0.047 | 0.145 | 0.067 | 0.174 | 0.093 | 0.218 | 0.082 | 0.208 | 0.134 | 0.255 | 0.132 | 0.248 |
| **m2** 12.5% | **0.016** | 0.078 | 0.017 | **0.076** | 0.018 | 0.080 | 0.026 | 0.094 | 0.108 | 0.239 | 0.056 | 0.159 | 0.021 | 0.088 | 0.023 | 0.092 | 0.133 | 0.270 | 0.108 | 0.228 | 0.034 | 0.127 | 0.062 | 0.166 |
| 25% | **0.017** | 0.081 | 0.020 | **0.080** | 0.020 | 0.085 | 0.028 | 0.099 | 0.164 | 0.294 | 0.080 | 0.195 | 0.024 | 0.096 | 0.026 | 0.101 | 0.135 | 0.272 | 0.136 | 0.262 | 0.042 | 0.143 | 0.085 | 0.196 |
| 37.5% | **0.018** | **0.084** | 0.022 | 0.087 | 0.023 | 0.091 | 0.030 | 0.104 | 0.237 | 0.356 | 0.110 | 0.231 | 0.027 | 0.103 | 0.030 | 0.108 | 0.155 | 0.293 | 0.175 | 0.300 | 0.051 | 0.159 | 0.106 | 0.222 |
| 50% | **0.020** | **0.088** | 0.025 | 0.095 | 0.026 | 0.098 | 0.034 | 0.110 | 0.323 | 0.421 | 0.156 | 0.276 | 0.030 | 0.108 | 0.035 | 0.119 | 0.200 | 0.333 | 0.211 | 0.329 | 0.059 | 0.174 | 0.131 | 0.247 |
| **h1** 12.5% | 0.119 | 0.212 | **0.043** | **0.140** | 0.057 | 0.159 | 0.093 | 0.201 | 0.126 | 0.263 | 0.070 | 0.190 | 0.060 | 0.165 | 0.074 | 0.182 | 0.114 | 0.234 | 0.074 | 0.194 | 0.240 | 0.345 | 0.151 | 0.267 |
| 25% | 0.127 | 0.220 | **0.054** | **0.156** | 0.069 | 0.178 | 0.107 | 0.217 | 0.169 | 0.304 | 0.106 | 0.236 | 0.080 | 0.189 | 0.090 | 0.203 | 0.140 | 0.262 | 0.102 | 0.227 | 0.265 | 0.364 | 0.180 | 0.292 |
| 37.5% | 0.138 | 0.230 | **0.072** | **0.180** | 0.084 | 0.196 | 0.120 | 0.230 | 0.220 | 0.347 | 0.124 | 0.258 | 0.102 | 0.212 | 0.109 | 0.222 | 0.174 | 0.293 | 0.135 | 0.261 | 0.296 | 0.382 | 0.215 | 0.318 |
| 50% | 0.157 | 0.247 | 0.107 | 0.216 | **0.102** | **0.215** | 0.141 | 0.248 | 0.293 | 0.402 | 0.165 | 0.299 | 0.133 | 0.240 | 0.137 | 0.248 | 0.215 | 0.325 | 0.179 | 0.298 | 0.334 | 0.404 | 0.257 | 0.347 |
| **h2** 12.5% | 0.040 | 0.129 | **0.039** | **0.125** | 0.040 | 0.130 | 0.057 | 0.152 | 0.187 | 0.319 | 0.095 | 0.212 | 0.042 | 0.133 | 0.044 | 0.138 | 0.305 | 0.431 | 0.163 | 0.289 | 0.101 | 0.231 | 0.100 | 0.216 |
| 25% | **0.041** | **0.131** | 0.044 | 0.135 | 0.046 | 0.141 | 0.061 | 0.158 | 0.279 | 0.390 | 0.137 | 0.258 | 0.049 | 0.147 | 0.050 | 0.149 | 0.322 | 0.444 | 0.206 | 0.331 | 0.115 | 0.246 | 0.127 | 0.247 |
| 37.5% | **0.043** | **0.136** | 0.051 | 0.147 | 0.052 | 0.151 | 0.067 | 0.166 | 0.400 | 0.465 | 0.187 | 0.304 | 0.056 | 0.158 | 0.060 | 0.163 | 0.353 | 0.462 | 0.252 | 0.370 | 0.126 | 0.257 | 0.158 | 0.276 |
| 50% | **0.047** | **0.142** | 0.059 | 0.158 | 0.060 | 0.162 | 0.073 | 0.174 | 0.602 | 0.572 | 0.232 | 0.341 | 0.065 | 0.170 | 0.068 | 0.173 | 0.369 | 0.472 | 0.316 | 0.419 | 0.136 | 0.268 | 0.183 | 0.299 |
| **AvgWins** | **52.1%** | | 35.4% | | 18.8% | | 0% | | 0% | | 0% | | 0% | | 0% | | 0% | | 0% | | 0% | | 0% | |

Table 5: **Imputation Summary.** In all categories TOTEM has SOTA `AvgWins`. In the specialist TOTEM has 52.1% `AvgWins`; in generalist in domain TOTEM has 58.3%; in generalist zero shot TOTEM has 80.0%.

Since we only use 3 seeds, we run a non-parametric *permutation test* on the generalist models in Appendix A.6 to analyze the performance of TOTEM vs. GPT2 (Table 24), and TOTEM vs. PatchTOTEM (Table 25). We find that TOTEM statistically significantly ($p \leq 0.05$) outperforms GPT2 in terms of `AvgWins` on all tasks for both the in-domain and zero-shot testing paradigms. Additionally, TOTEM outperforms PatchTOTEM in a statistically significant ($p \leq 0.05$) manner for in-domain and zero-shot testing.

## 5.1 Imputation

In imputation, models intake a masked time series $\mathbf{x_m} \in \mathbb{R}^{S \times T_{\text{in}}}$, and then impute the signal $\hat{\mathbf{x}} \in \mathbb{R}^{S \times T_{\text{in}}}$ (see Figure 11). We experiment with four canonical masking percentages at $12.5\%, 25\%, 37.5\%, 50\%$, and report the resulting `MSE` and `MAE`.

**Specialists.** Figure 5A and Table 5D compare TOTEM to specialist baselines. All models are trained and evaluated on the same dataset (in-domain). TOTEM has the highest `AvgWins` with 52.1%, followed by GPT2 at 35.4%, and TiNet at 18.8%. TOTEM's performance on m1 and h1 is lower, but since these datasets are the minute and hour resampling of the same raw data respectively, we expect their results to be correlated.

**Generalists.** Figure 5A and Tables 5B&C compare TOTEM to GPT2 (the best two models in the specialist in-domain regime) in the generalist setting, when both models are trained on the aggregate of the W, E, m1, m2, h1, and h2 datasets. We evaluate them on both the in-domain and zero-shot test sets. TOTEM outperforms GPT2 in-domain, 58.3% vs. 43.8%, and by a much larger margin zero-shot, 80% vs. 20%. TOTEM's performance across all experiments demonstrate that tokens are a performant representation for imputation. We visualize codebook examples in Figure 13, and imputation examples in Figure 14.

## 5.2 Anomaly Detection

In anomaly detection, models intake a corrupted time series $\mathbf{x_{corr}} \in \mathbb{R}^{S \times T_{in}}$ and predict which times correspond to anomalies via a binary mask $\hat{\mathbf{y}} \in \{0,1\}^{T_{in}}$, where the amount of corruption is considered known, at A% (see Figure 12). We report Precision P ($\uparrow$), Recall R ($\uparrow$), and F1 Score ($\uparrow$). In the main text, we compare against the flawed baselines from prior work (see Section 4.4) for ease of comparison. We compare against a subset of 15 "correct" baselines from Wu & Keogh (2021) in Table 13. In both cases, we find that TOTEM achieves SOTA results.

**Specialists.** Figure 5 and Table 10 test TOTEM against specialist baselines. TOTEM has the highest `AvgWins` at 33.3% followed by a tie between GPT2, TiNet, ATrans, ETS, and LogTr at 13.3%.

**Generalists.** Figure 5 and Table 11 compare generalist-trained TOTEM and GPT2. On the in-domain and zero-shot regimes, TOTEM outperforms GPT2 80% to 20% and 73.3% to 26.7% respectively. TOTEM's `AvgWins` across the specialist and generalist settings demonstrate that tokens are a performant representation for anomaly detection.

Figure 5: **Anomaly Detection Results.** In all cases, TOTEM has SOTA `AvgWins`. Vs. specialists, TOTEM has 33.3%; vs. generalists in-domain, TOTEM has 80.0%; vs. generalists zero-shot, TOTEM has 73.3%.

## 5.3 Forecasting

In forecasting, models intake a time series $\mathbf{x} \in \mathbb{R}^{S \times T_{in}}$ and predict future readings $\mathbf{y} \in \mathbb{R}^{S \times T_{out}}$, where $S$ is the sensor dimension and $T_{in}, T_{out}$ signify the durations of the preceding and succeeding time series, respectively. All models have a lookback window of $T_{in} = 96$, with prediction lengths $T_{out} = \{96, 192, 336, 720\}$. Results for baselines are from Liu et al. (2023). We run GPT2 with $T_{in} = 96$ as Zhou et al. (2023) originally use inconsistent dataset-specific lookback lengths. See Figure 6 for a summary.

**Specialists.** Figure 6 and Table 14 show that TOTEM achieves the highest `AvgWins` at 28.6% followed by iTrans at 26.8%. In particular, TOTEM has first-place finishes in five datasets while iTrans' first-place finishes are concentrated in only the electricity and traffic datasets.

**Generalists.** Figure 6 and Table 15 compare the generalist-trained TOTEM and GPT2 models. TOTEM outperforms GPT2 in both the in-domain (67.9% vs. 33.9%) and zero-shot (90.0% vs. 12.5%) regimes. TOTEM's `AvgWins` across both regimes show that tokens are a performant representation for forecasting.

## 6 Ablations

We present 4 ablation studies: (i) testing tokens vs. patches for a fixed TOTEM architecture, (ii) testing tokens vs. patches using both transformer and MLP forecasters, (iii) a codebook size study, and (iv) a study of TOTEM's zero-shot performance when trained on datasets of different sizes.

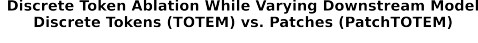

Figure 6: **Forecasting Summary.** In all categories TOTEM has SOTA `AvgWins`. In the specialist TOTEM has 28.6%; in generalist in domain TOTEM has 67.9%; in generalist zero shot TOTEM has 90.0%.

Figure 7: **Discrete Token Ablation.** In all categories, the discrete token representation (TOTEM) has SOTA `AvgWins` over the patch representation (PatchTOTEM).

**Tokens vs. Patches.** The experiments in Section 5 show that the combination of discrete tokenization and TOTEM's generalist architecture achieve SOTA performance. We now fix the architecture while varying only the representation (TOTEM vs. PatchTOTEM) on a forecasting task to test what proportion of the performance is attributable to tokenization. We find that in all testing regimes used in the main results, TOTEM greatly outperforms PatchTOTEM, with 67.9% vs. 39.3% `AvgWins` in the specialist in-domain regime, 78.6% vs. 23.2% `AvgWins` in the generalist in-domain regime, and 67.5% vs. 35.0% `AvgWins` in the generalist zero-shot regime (see Figure 7 and Table 21).

**Downstream Architecture Study.** In Figure 8 & Table 21, we explore the effect of discrete tokens vs. patches for each of two common downstream forecasting models: the transformer encoder introduced in Section 3.3 and Figure 4, and an MLP (Ekambaram et al., 2023; Das et al., 2023a; Zeng et al., 2023). The MLP has 3-layers ReLU activations, uses dropout with $p = 0.1$ after the second layer, and concludes with a layernorm; this architecture is modeled after similar architectures in the literature like Das et al. (2023a). The patch-based MLP takes in an uncompressed time series. We find that for both the MLP and transformer architectures, the discrete token representation outperforms the patch representation (in the transformer 67.9% to 39.3% `AvgWins` and MLP 66.1% to 37.5% `AvgWins`). This shows that TOTEM's strength in forecasting is not due to the strength of the transformer forecaster, but because of the choice to use discrete tokens.

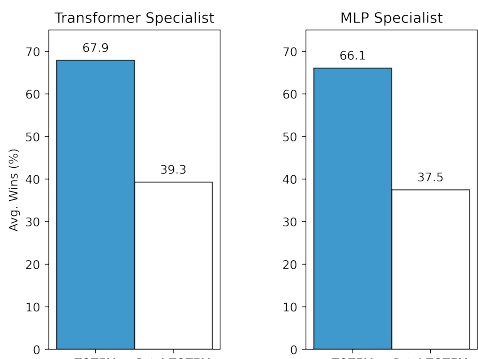

Figure 8: **Discrete Token vs. Patches with MLP.** For both the transformer (left) and MLP (right) the discrete token representation (TOTEM) outperforms the patch respresentation (PatchTOTEM).

**Codebook Size.** In Figure 9, we explore the effect of the codebook size $K$ on the VQVAE's reconstruction performance. As expected, we find that as $K$ increases from 32 to 256 to 512, the reconstruction performance improves. However, for downstream tasks like forecasting, it is more parsimonious to model interactions between fewer codewords. Thus, we elect to use $K = 256$ codewords, as the reconstruction performance is similar to that of $K = 512$. We note that the the average generalist codebook error (see Table 21D), is substantially lower than the corresponding downstream forecasting error, demonstrating that a larger proportion of error is attributable to the difficulty of the forecasting task rather than poor reconstruction. This provides evidence that time

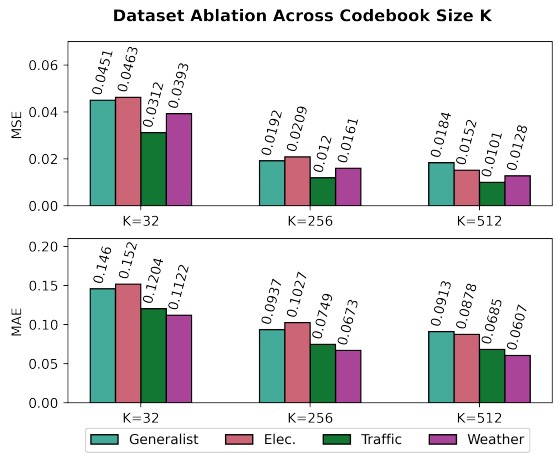

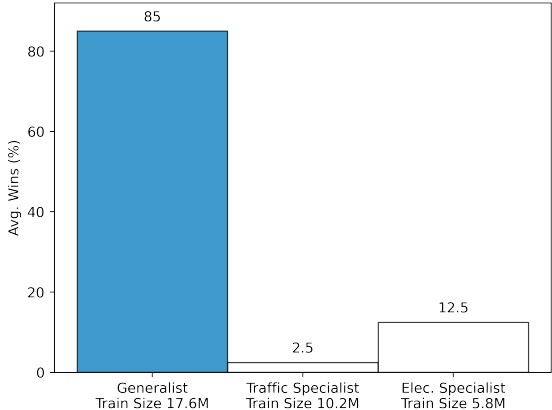

Figure 9: **Codebook Size Ablation.** As the codebook size $K$ increases, the reconstruction loss of the VQVAE decreases on a variety of datasets.

Figure 10: **Dataset Size Study.** As expected, the generalist has the highest zero-shot performance at 85.0% `AvgWins`, but the electricity specialist outperforms the traffic specialist even with a smaller training dataset. This confirms that dataset diversity may be more important than dataset scale for generalization.

series can have a single unified representation across multiple domains, akin to BPE in language modeling. We note that this same trend holds for the specialist models as well.

**Dataset Size Study.** One natural question is whether TOTEM's strong generalization performance is driven by the size of the dataset or the diversity of the training samples. We study this in a minimal setting by comparing the TOTEM generalist model against two TOTEM specialists trained on the two largest domain-specific datasets: traffic (10.2M examples) and electricity (5.8M examples). As expected, the results in Figure 10 show that the TOTEM generalist significantly outperforms the two specialists in the zero-shot setting. However, the electricity specialist outperformed the traffic specialist even though the training dataset was about half the size. This provides some preliminary evidence that simply training on *more* data is insufficient for achieving generalization - the types of data are also crucial. For related exploratory studies on generalist models, see Appendix A.7.

# 7 Conclusion

We present TOTEM: a simple, performant tokenizer that is designed to learn domain-agnostic discrete representations for time series data, paving the way for time series foundation models. TOTEM demonstrates strong in-domain and zero-shot capabilities versus a large array of both generalist and specialist baselines across dozens of domains and datasets over hundreds of seeded experiments. Overall, TOTEM unlocks domain generalization while performing at or above existing SOTA levels, demonstrating the potential of adopting training and modeling techniques from language and vision modeling for time series modeling.

There are many exciting directions for future work. First, our proposed architectural design decisions were very simple, which suggests that there are many possible performant extensions. Further, while we have collated millions of existing time series, TOTEM's promising initial results suggest that scaling up the generalist training dataset size by an order of magnitude or more could unlock true domain- and task-agnostic generalizability. Such followup works could allow a more systematic study of the relationships between generalist data representations, token length, data size, and domain diversity.

## 8 Broader Impact Statement

There are no immediate ethical concerns that arise from our work. However, as with all data driven methods, certain societal consequences are important to be discussed, in this case surrounding time series modeling. A few are reported below:

**Privacy Concerns.** Time series data, especially when sourced from personal devices or applications, can contain sensitive information about individuals, e.g. for health domains. In this work, no time series were sourced from personal devices.

**Misuse.** Time series forecast models can be misused. For instance, if a model forecasts stock prices or market movements, it could be exploited for insider trading or other illegal financial activities. In this work, we are focused on domains pertinent to scientific disciplines.

**Economic Impacts.** Automated forecasts and decisions based on time series models can significantly impact industries and labor markets both positively and negatively. For instance, if a model can accurately predict weather patterns, it might affect farmers and their crop decisions, or if it can forecast energy consumption, it could impact the energy sector.

## 9 Acknowledgments

We thank Albert Hao Li for helpful discussions and edits, Addison Hu for insights into statistical modeling, and Angela Gao and Jack Wilding for discussions surrounding applications to earthquake data.

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

# A    Appendix

## A.1    Dataset.

| Dataset | \|Sampling Rate\| | Number Sensors |
|---|---|---|
| Imputation & Forecasting Training Sets | | |
| Weather | Every 10 min | 21 |
| Traffic | Every hour | 862 |
| Electricity | Every hour | 321 |
| Etth1, ETTh2 | Every hour | 7 |
| Ettm1, ETTm2 | Every 15 min | 7 |
| Anomaly Detection Training Sets | | |
| SMD (Sever Machine) | Every min | 38 |
| MSL (Mars Rover) | Every min | 55 |
| SMAP (Soil Moisture) | Every min | 25 |
| SWAT (Water Treatment) | Every sec | 51 |
| PSM (Pooled Server) | Every min | 25 |
| Zero Shot Testing Sets for Imputation, Forecasting & Anomaly Detection | | |
| Neuro2 | Every 0.002 sec | 72 |
| Neuro5 | Every 0.002 sec | 106 |
| Saugeen River Flow | Every day | 1 |
| US Birth Rate | Every day | 1 |
| Sunspot | Every day | 1 |

Table 6: Dataset Information Table. Notably no sampling rate or sensor number is shared between the training sets and testing sets for any task.

| Task | Training Domains Explored (Main Paper) | Training Domains Explored (Appendix) | Zero Shot Testing Domains Explored |
|---|---|---|---|
| Imputation | Weather, Electricity, Transformer Temperature | Healthcare | Neuroscience (ECoG), River Flow, U.S. Birth Rate, Sunspot |
| Anomaly Detection | Server Machines, Mars Science Lab, Soil Moisture, Water Treatment | Insect Feeding, Walking Acceleration, Air Temperature, 3D Gait Phase, Heart Beat (including ECG datasets), Parkinsons Asymmetry, Tilt Table Beat, Internal Bleeding, Accelerometer on Whale, NASA SpaceCraft Increase Rate | Neuroscience (ECoG), River Flow, U.S. Birth Rate, Sunspot |
| Forecasting | Traffic, Weather, Electricity, Transformer Temperature | Tourism, Imports & Exports, Real Estate, etc. [aggregated in the W4, W3, etc. datasets], Demographics | Neuroscience (ECoG), River Flow, U.S. Birth Rate, Sunspot |

Table 7: Domain Diversity Table. Here we list various domains that are used for each task across training and testing in both the main paper and Appendix.

## A.2    Imputation.

**Imputer**

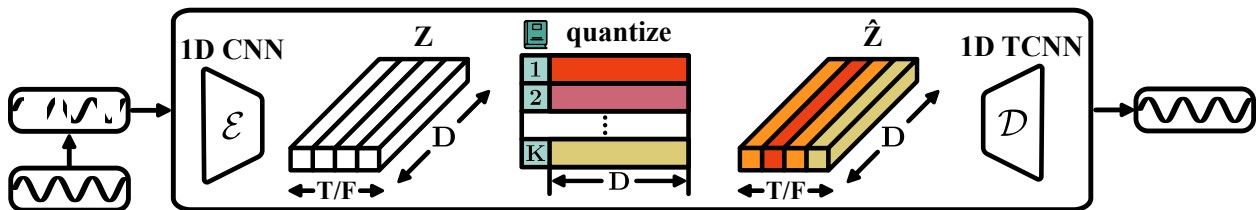

Figure 11: **Imputation Visualization.** The VQVAE architecture does not change for the imputation task. The data passed in has a mask applied to it so that the VQVAE solves the task of reconstruction and imputation simultaneously.

Table 8: **Means & Stds. for the Imputation Task.** A. is the TOTEM specialist, B. is the TOTEM generalist, C. is the GPT2 generalist which we setup to run in a generalist manner.

**B. TOTEM - Generalist Imputation (↓)**

| Metric | | MSE | MAE |
|---|---|---|---|
| W | 12.5% | $0.029 \pm 0.0012$ | $0.060 \pm 0.0047$ |
| | 25% | $0.030 \pm 0.0006$ | $0.060 \pm 0.0047$ |
| | 37.5% | $0.032 \pm 0.0006$ | $0.062 \pm 0.0030$ |
| | 50% | $0.036 \pm 0.0006$ | $0.067 \pm 0.0036$ |
| E | 12.5% | $0.065 \pm 0.0020$ | $0.171 \pm 0.0032$ |
| | 25% | $0.071 \pm 0.0015$ | $0.179 \pm 0.0031$ |
| | 37.5% | $0.080 \pm 0.0025$ | $0.189 \pm 0.0032$ |
| | 50% | $0.095 \pm 0.0026$ | $0.205 \pm 0.0032$ |
| m1 | 12.5% | $0.041 \pm 0.0006$ | $0.132 \pm 0.0015$ |
| | 25% | $0.044 \pm 0.0000$ | $0.135 \pm 0.0010$ |
| | 37.5% | $0.048 \pm 0.0006$ | $0.139 \pm 0.0040$ |
| | 50% | $0.058 \pm 0.0010$ | $0.152 \pm 0.0000$ |
| m2 | 12.5% | $0.040 \pm 0.0020$ | $0.125 \pm 0.0067$ |
| | 25% | $0.041 \pm 0.0015$ | $0.126 \pm 0.0058$ |
| | 37.5% | $0.043 \pm 0.0015$ | $0.129 \pm 0.0049$ |
| | 50% | $0.048 \pm 0.0010$ | $0.136 \pm 0.0038$ |
| h1 | 12.5% | $0.100 \pm 0.0049$ | $0.201 \pm 0.0049$ |
| | 25% | $0.108 \pm 0.0049$ | $0.209 \pm 0.0038$ |
| | 37.5% | $0.122 \pm 0.0064$ | $0.220 \pm 0.0044$ |
| | 50% | $0.144 \pm 0.0078$ | $0.237 \pm 0.0049$ |
| h2 | 12.5% | $0.075 \pm 0.0012$ | $0.175 \pm 0.0053$ |
| | 25% | $0.076 \pm 0.0006$ | $0.177 \pm 0.0036$ |
| | 37.5% | $0.093 \pm 0.0222$ | $0.195 \pm 0.0200$ |
| | 50% | $0.089 \pm 0.0010$ | $0.192 \pm 0.0035$ |

**Zero-Shot**

| Metric | | MSE | MAE |
|---|---|---|---|
| N2 | 12.5% | $0.029 \pm 0.0015$ | $0.120 \pm 0.0045$ |
| | 25% | $0.033 \pm 0.0010$ | $0.127 \pm 0.0035$ |
| | 37.5% | $0.041 \pm 0.0006$ | $0.139 \pm 0.0025$ |
| | 50% | $0.056 \pm 0.0006$ | $0.160 \pm 0.0012$ |
| N5 | 12.5% | $0.017 \pm 0.0010$ | $0.085 \pm 0.0030$ |
| | 25% | $0.019 \pm 0.0010$ | $0.090 \pm 0.0030$ |
| | 37.5% | $0.022 \pm 0.0006$ | $0.098 \pm 0.0025$ |
| | 50% | $0.029 \pm 0.0006$ | $0.110 \pm 0.0025$ |
| R | 12.5% | $0.071 \pm 0.0070$ | $0.109 \pm 0.0040$ |
| | 25% | $0.087 \pm 0.0064$ | $0.117 \pm 0.0031$ |
| | 37.5% | $0.112 \pm 0.0050$ | $0.129 \pm 0.0035$ |
| | 50% | $0.148 \pm 0.0032$ | $0.147 \pm 0.0023$ |
| B | 12.5% | $0.632 \pm 0.0087$ | $0.642 \pm 0.0068$ |
| | 25% | $0.693 \pm 0.0070$ | $0.665 \pm 0.0047$ |
| | 37.5% | $0.761 \pm 0.0055$ | $0.692 \pm 0.0023$ |
| | 50% | $0.827 \pm 0.0044$ | $0.718 \pm 0.0000$ |
| S | 12.5% | $0.057 \pm 0.0012$ | $0.160 \pm 0.0023$ |
| | 25% | $0.061 \pm 0.0006$ | $0.168 \pm 0.0021$ |
| | 37.5% | $0.069 \pm 0.0006$ | $0.178 \pm 0.0021$ |
| | 50% | $0.082 \pm 0.0010$ | $0.193 \pm 0.0015$ |

**C. GPT2 - Generalist Imputation (↓)**

| Metric | | MSE | MAE |
|---|---|---|---|
| W | 12.5% | $0.029 \pm 0.0000$ | $0.045 \pm 0.0006$ |
| | 25% | $0.033 \pm 0.0006$ | $0.048 \pm 0.0006$ |
| | 37.5% | $0.037 \pm 0.0006$ | $0.054 \pm 0.0012$ |
| | 50% | $0.043 \pm 0.0012$ | $0.061 \pm 0.0017$ |
| E | 12.5% | $0.008 \pm 0.0020$ | $0.186 \pm 0.0035$ |
| | 25% | $0.091 \pm 0.0020$ | $0.197 \pm 0.0025$ |
| | 37.5% | $0.108 \pm 0.0021$ | $0.213 \pm 0.0026$ |
| | 50% | $0.132 \pm 0.0026$ | $0.236 \pm 0.0026$ |
| m1 | 12.5% | $0.052 \pm 0.0012$ | $0.141 \pm 0.0016$ |
| | 25% | $0.065 \pm 0.0021$ | $0.154 \pm 0.0021$ |
| | 37.5% | $0.085 \pm 0.0038$ | $0.171 \pm 0.0026$ |
| | 50% | $0.117 \pm 0.0052$ | $0.196 \pm 0.0026$ |
| m2 | 12.5% | $0.029 \pm 0.0000$ | $0.095 \pm 0.0006$ |
| | 25% | $0.033 \pm 0.0006$ | $0.101 \pm 0.0006$ |
| | 37.5% | $0.038 \pm 0.0006$ | $0.110 \pm 0.0012$ |
| | 50% | $0.045 \pm 0.0006$ | $0.121 \pm 0.0012$ |
| h1 | 12.5% | $0.113 \pm 0.0012$ | $0.217 \pm 0.0021$ |
| | 25% | $0.131 \pm 0.0010$ | $0.231 \pm 0.0015$ |
| | 37.5% | $0.153 \pm 0.0012$ | $0.247 \pm 0.0017$ |
| | 50% | $0.182 \pm 0.0006$ | $0.266 \pm 0.0012$ |
| h2 | 12.5% | $0.067 \pm 0.0010$ | $0.155 \pm 0.0015$ |
| | 25% | $0.071 \pm 0.0006$ | $0.160 \pm 0.0015$ |
| | 37.5% | $0.077 \pm 0.0010$ | $0.167 \pm 0.0015$ |
| | 50% | $0.086 \pm 0.0032$ | $0.179 \pm 0.0038$ |

**Zero-Shot**

| Metric | | MSE | MAE |
|---|---|---|---|
| N2 | 12.5% | $0.047 \pm 0.0006$ | $0.145 \pm 0.0015$ |
| | 25% | $0.064 \pm 0.0017$ | $0.164 \pm 0.0015$ |
| | 37.5% | $0.090 \pm 0.0036$ | $0.191 \pm 0.0032$ |
| | 50% | $0.131 \pm 0.0051$ | $0.228 \pm 0.0044$ |
| N5 | 12.5% | $0.021 \pm 0.0006$ | $0.095 \pm 0.0012$ |
| | 25% | $0.028 \pm 0.0006$ | $0.107 \pm 0.0010$ |
| | 37.5% | $0.039 \pm 0.0015$ | $0.123 \pm 0.0015$ |
| | 50% | $0.055 \pm 0.0015$ | $0.145 \pm 0.0023$ |
| R | 12.5% | $0.093 \pm 0.0010$ | $0.119 \pm 0.0015$ |
| | 25% | $0.125 \pm 0.0006$ | $0.134 \pm 0.0026$ |
| | 37.5% | $0.167 \pm 0.0021$ | $0.154 \pm 0.0042$ |
| | 50% | $0.220 \pm 0.0045$ | $0.182 \pm 0.0057$ |
| B | 12.5% | $0.392 \pm 0.0064$ | $0.496 \pm 0.0023$ |
| | 25% | $0.444 \pm 0.0071$ | $0.523 \pm 0.0029$ |
| | 37.5% | $0.498 \pm 0.0080$ | $0.553 \pm 0.0023$ |
| | 50% | $0.591 \pm 0.0700$ | $0.599 \pm 0.0275$ |
| S | 12.5% | $0.070 \pm 0.0012$ | $0.173 \pm 0.0017$ |
| | 25% | $0.084 \pm 0.0010$ | $0.189 \pm 0.0015$ |
| | 37.5% | $0.103 \pm 0.0010$ | $0.209 \pm 0.0021$ |
| | 50% | $0.128 \pm 0.0015$ | $0.234 \pm 0.0021$ |

**A. TOTEM - Specialist Imputation (↓)**

| Metric | | MSE | MAE |
|---|---|---|---|
| W | 12.5% | $0.028 \pm 0.0000$ | $0.046 \pm 0.0006$ |
| | 37.5% | $0.029 \pm 0.0000$ | $0.047 \pm 0.0010$ |
| | 50% | $0.031 \pm 0.0006$ | $0.048 \pm 0.0015$ |
| | 25% | $0.033 \pm 0.0006$ | $0.052 \pm 0.0006$ |
| E | 12.5% | $0.054 \pm 0.0006$ | $0.154 \pm 0.0015$ |
| | 25% | $0.059 \pm 0.0006$ | $0.160 \pm 0.0010$ |
| | 37.5% | $0.067 \pm 0.0006$ | $0.169 \pm 0.0012$ |
| | 50% | $0.079 \pm 0.0012$ | $0.183 \pm 0.0012$ |
| m1 | 12.5% | $0.049 \pm 0.0000$ | $0.125 \pm 0.0006$ |
| | 25% | $0.052 \pm 0.0006$ | $0.128 \pm 0.0006$ |
| | 37.5% | $0.055 \pm 0.0000$ | $0.132 \pm 0.0006$ |
| | 50% | $0.061 \pm 0.0006$ | $0.139 \pm 0.0006$ |
| m2 | 12.5% | $0.016 \pm 0.0006$ | $0.078 \pm 0.0010$ |
| | 25% | $0.017 \pm 0.0006$ | $0.081 \pm 0.0006$ |
| | 37.5% | $0.018 \pm 0.0000$ | $0.084 \pm 0.0006$ |
| | 50% | $0.020 \pm 0.0000$ | $0.088 \pm 0.0000$ |
| h1 | 12.5% | $0.119 \pm 0.0010$ | $0.212 \pm 0.0006$ |
| | 25% | $0.127 \pm 0.0015$ | $0.220 \pm 0.0006$ |
| | 37.5% | $0.138 \pm 0.0012$ | $0.230 \pm 0.0006$ |
| | 50% | $0.157 \pm 0.0006$ | $0.247 \pm 0.0010$ |
| h2 | 12.5% | $0.040 \pm 0.0006$ | $0.129 \pm 0.0017$ |
| | 25% | $0.041 \pm 0.0010$ | $0.131 \pm 0.0012$ |
| | 37.5% | $0.043 \pm 0.0006$ | $0.136 \pm 0.0006$ |
| | 50% | $0.047 \pm 0.0006$ | $0.142 \pm 0.0012$ |

Table 9: **Imputation on PhysioNet 2012 Dataset.** We report MAE where lower is better. TOTEM has the best performance in all three scenarios of percent missing.

| Method | 10% Missing | 50% Missing | 90% Missing |
|---|---|---|---|
| V-Rin | 0.271 | 0.365 | 0.606 |
| BRITS | 0.284 | 0.368 | 0.517 |
| RDIS | 0.319 | 0.419 | 0.613 |
| Unconditional | 0.326 | 0.417 | 0.625 |
| CSDI | 0.217 | 0.301 | 0.481 |
| TOTEM (Ours) | **0.126** | **0.134** | **0.143** |

## A.3 Anomaly Detection.

**Anomaly Detector**

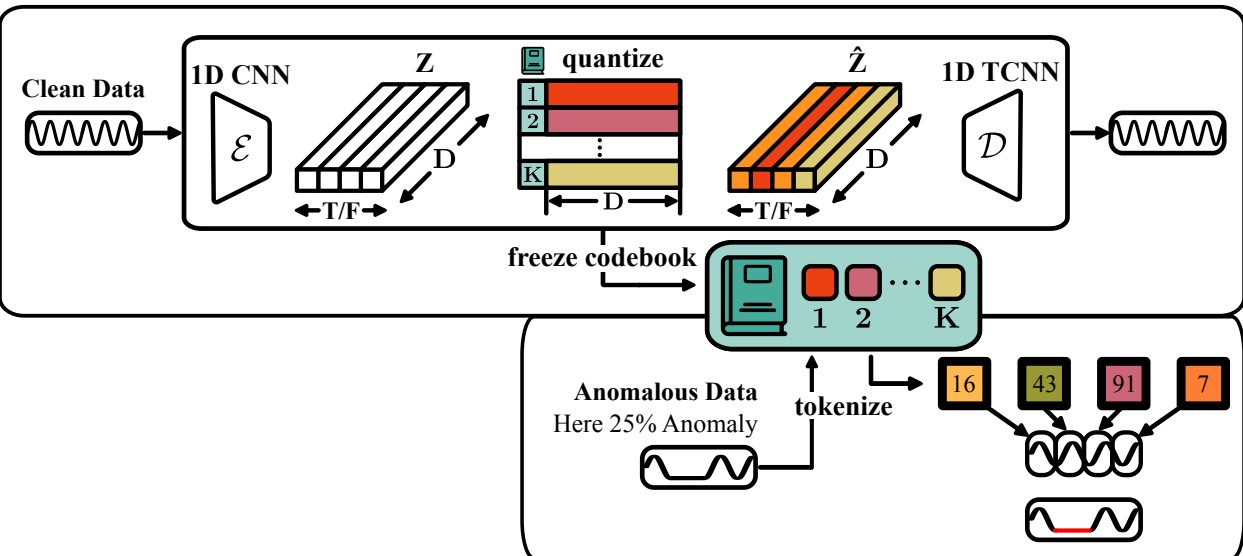

Figure 12: **Anomaly Detection Visualization.** The VQVAE architecture does not change for the anomaly detection task. The training data passed in must be clean such that the VQVAE can learn clean representations. At test time, when anomaly data is passed in with anomaly A% (in this case 25%), the worst A% reconstructed is set to the anomaly.

Table 10: **Specialist Anomaly Detection** (↑). TOTEM has the highest `AvgWins` at **33.3%** followed by a five-way tie between GPT2, TiNet, ATrans, ETS, and LogTr at 13.3%. Some prior methods use the test set as a validation set for early stopping of the learning algorithm, which can inflate performance. We do not adopt this practice and train TOTEM for a set number of iterations.

| | Model | TOTEM | GPT2 | TiNet | ATran | Patch | ETS | FED | Stat | Auto | Pyra | Inf | Re | LogTr | Trans | LiTS | DLin |
|---|---|---|---|---|---|---|---|---|---|---|---|---|---|---|---|---|---|
| **F1** | SMD | 79.62 | **86.89** | 84.61 | 85.49 | 84.62 | 83.13 | 85.08 | 84.62 | 85.11 | 83.04 | 81.65 | 75.32 | 76.21 | 79.56 | 82.53 | 77.10 |
| | MSL | 82.58 | 82.45 | 81.84 | 83.31 | 78.70 | **85.03** | 78.57 | 77.50 | 79.05 | 84.86 | 84.06 | 84.40 | 79.57 | 78.68 | 78.95 | 84.88 |
| | SMAP | **94.02** | 72.88 | 69.39 | 71.18 | 68.82 | 69.50 | 70.76 | 71.09 | 71.12 | 71.09 | 69.92 | 70.40 | 69.97 | 69.70 | 69.21 | 69.26 |
| | SWAT | **94.27** | 94.23 | 93.02 | 83.10 | 85.72 | 84.91 | 93.19 | 79.88 | 92.74 | 91.78 | 81.43 | 82.80 | 80.52 | 80.37 | 93.33 | 87.52 |
| | PSM | 95.87 | 97.13 | **97.34** | 79.40 | 96.08 | 91.76 | 97.23 | 97.29 | 93.29 | 82.08 | 77.10 | 73.61 | 76.74 | 76.07 | 97.15 | 93.55 |
| **R** | SMD | 76.06 | **84.98** | 81.54 | 82.23 | 82.14 | 79.23 | 82.39 | 81.21 | 82.35 | 80.61 | 77.23 | 69.24 | 70.13 | 76.13 | 78.42 | 71.52 |
| | MSL | 82.85 | 82.91 | 75.36 | 87.37 | 70.96 | 84.93 | 80.07 | **89.14** | 80.92 | 85.93 | 86.48 | 83.31 | 87.37 | 87.37 | 75.78 | 85.42 |
| | SMAP | **94.04** | 60.95 | 56.40 | 58.11 | 55.46 | 55.75 | 58.10 | 59.02 | 58.62 | 57.71 | 57.13 | 57.44 | 57.59 | 57.12 | 55.27 | 55.41 |
| | SWAT | 95.91 | 96.34 | 95.40 | **97.32** | 80.94 | 80.36 | 96.42 | 96.75 | 95.81 | 96.00 | 96.75 | 96.53 | **97.32** | 96.53 | 94.72 | 95.30 |
| | PSM | 94.21 | 95.68 | 96.20 | 94.72 | 93.47 | 85.28 | 97.16 | 96.76 | 88.15 | 96.02 | 96.33 | 95.38 | **98.00** | 96.56 | 95.97 | 89.26 |
| **P** | SMD | 83.54 | 88.89 | 87.91 | **88.91** | 87.26 | 87.44 | 87.95 | 88.33 | 88.06 | 85.61 | 86.60 | 82.58 | 83.46 | 83.58 | 87.10 | 83.62 |
| | MSL | 82.32 | 82.00 | **89.54** | 79.61 | 88.34 | 85.13 | 77.14 | 68.55 | 77.27 | 83.81 | 81.77 | 85.51 | 73.05 | 71.57 | 82.40 | 84.34 |
| | SMAP | **94.00** | 90.60 | 90.14 | 91.85 | 90.64 | 92.25 | 90.47 | 89.37 | 90.40 | 92.54 | 90.11 | 90.91 | 89.15 | 89.37 | 92.58 | 92.32 |
| | SWAT | **92.68** | 92.20 | 90.75 | 72.51 | 91.10 | 90.02 | 90.17 | 68.03 | 89.85 | 87.92 | 70.29 | 72.50 | 68.67 | 68.84 | 91.98 | 80.91 |
| | PSM | 97.58 | 98.62 | 98.51 | 68.35 | 98.84 | **99.31** | 97.31 | 97.82 | 99.08 | 71.67 | 64.27 | 59.93 | 63.06 | 62.75 | 98.37 | 98.28 |
| `AvgWins` | | **33.3%** | 13.3% | 13.3% | 13.3% | 0% | 13.3% | 0% | 6.7% | 0% | 0% | 0% | 0% | 13.3% | 0% | 0% | 0% |

Table 11: **Generalist Anomaly Detection** (↑). We train TOTEM & GPT2 on all datasets and then perform in-domain and zero-shot evaluations. **A. In-Domain Performance.** TOTEM outperforms GPT2: **80.0%** vs. 20.0%. **B. Zero-Shot Performance.** TOTEM again outperforms GPT2: **73.3%** vs. 26.7%.

**A. In-Domain Performance**

| | Model | TOTEM | GPT2 |
|---|---|---|---|
| F1 | SMD | 78.64 | **79.73** |
| | MSL | **83.29** | 80.17 |
| | SMAP | **92.51** | 67.05 |
| | SWAT | **94.37** | 89.62 |
| | PSM | **95.78** | 90.47 |
| R | SMD | 72.07 | **73.42** |
| | MSL | **82.96** | 78.48 |
| | SMAP | **91.48** | 53.42 |
| | SWAT | **96.13** | 87.53 |
| | PSM | **93.90** | 87.76 |
| P | SMD | 86.66 | **87.44** |
| | MSL | **83.64** | 81.95 |
| | SMAP | **93.56** | 90.01 |
| | SWAT | **92.68** | 91.83 |
| | PSM | **97.74** | 93.39 |
| AvgWins | | **80.0%** | 20.0% |

**B. Zero-Shot Performance**

| | Model | TOTEM | GPT2 |
|---|---|---|---|
| F1 | N2 | **51.29** | 39.02 |
| | N5 | **51.28** | 42.19 |
| | R | **49.39** | 36.14 |
| | B | **49.15** | 20.81 |
| | S | **52.17** | 38.12 |
| R | N2 | **76.88** | 33.69 |
| | N5 | **76.84** | 36.77 |
| | R | **70.49** | 29.66 |
| | B | **73.71** | 17.67 |
| | S | **77.36** | 31.83 |
| P | N2 | 38.49 | **46.43** |
| | N5 | 38.48 | **49.58** |
| | R | 38.02 | **46.30** |
| | B | **36.86** | 25.33 |
| | S | 39.35 | **47.72** |
| AvgWins | | **73.3%** | 26.7% |

Table 12: **Means & Stds. for the Anomaly Detection Task.** A. is the TOTEM specialist, B. is the TOTEM generalist, C. is the GPT2 generalist which we setup to run in a generalist manner.

**A. TOTEM - Specialist Anomaly Detection (↑)**

| | | Mean ± Std |
|---|---|---|
| F1 | SMD | $0.7962 \pm 0.0137$ |
| | MSL | $0.8258 \pm 0.0052$ |
| | SMAP | $0.9402 \pm 0.0008$ |
| | SWAT | $0.9427 \pm 0.0006$ |
| | PSM | $0.9587 \pm 0.0008$ |
| R | SMD | $0.7606 \pm 0.0207$ |
| | MSL | $0.8285 \pm 0.0071$ |
| | SMAP | $0.9404 \pm 0.0013$ |
| | SWAT | $0.9591 \pm 0.0012$ |
| | PSM | $0.9421 \pm 0.0004$ |
| P | SMD | $0.8354 \pm 0.0054$ |
| | MSL | $0.8232 \pm 0.0033$ |
| | SMAP | $0.9400 \pm 0.0004$ |
| | SWAT | $0.9268 \pm 0.0003$ |
| | PSM | $0.9758 \pm 0.0012$ |

**B. TOTEM - Generalist Anomaly Detection (↑)** and **C. GPT2 - Generalist Anomaly Detection (↑)**

| | | B. Mean ± Std | C. Mean ± Std |
|---|---|---|---|
| F1 | SMD | $0.7864 \pm 0.0386$ | $0.7973 \pm 0.0326$ |
| | MSL | $0.8329 \pm 0.0020$ | $0.8017 \pm 0.0205$ |
| | SMAP | $0.9251 \pm 0.0014$ | $0.6705 \pm 0.0041$ |
| | SWAT | $0.9437 \pm 0.0005$ | $0.8962 \pm 0.0016$ |
| | PSM | $0.9578 \pm 0.0002$ | $0.9047 \pm 0.0759$ |
| | N2 | $0.5129 \pm 0.0397$ | $0.3902 \pm 0.0596$ |
| | N5 | $0.5128 \pm 0.0390$ | $0.4219 \pm 0.0047$ |
| | R | $0.4939 \pm 0.0625$ | $0.3614 \pm 0.0204$ |
| | B | $0.4915 \pm 0.0229$ | $0.2081 \pm 0.0462$ |
| | S | $0.5217 \pm 0.0418$ | $0.3812 \pm 0.0621$ |
| R | SMD | $0.7207 \pm 0.0565$ | $0.7342 \pm 0.0559$ |
| | MSL | $0.8296 \pm 0.0046$ | $0.7848 \pm 0.0277$ |
| | SMAP | $0.9148 \pm 0.0020$ | $0.5342 \pm 0.0051$ |
| | SWAT | $0.9613 \pm 0.0010$ | $0.8753 \pm 0.0033$ |
| | PSM | $0.9390 \pm 0.0004$ | $0.8776 \pm 0.0624$ |
| | N2 | $0.7688 \pm 0.0594$ | $0.3369 \pm 0.0592$ |
| | N5 | $0.7684 \pm 0.0582$ | $0.3677 \pm 0.0498$ |
| | R | $0.7049 \pm 0.0825$ | $0.2966 \pm 0.0218$ |
| | B | $0.7371 \pm 0.0340$ | $0.1767 \pm 0.0426$ |
| | S | $0.7736 \pm 0.0581$ | $0.3183 \pm 0.0648$ |
| P | SMD | $0.8666 \pm 0.0114$ | $0.8744 \pm 0.0029$ |
| | MSL | $0.8364 \pm 0.0014$ | $0.8195 \pm 0.0130$ |
| | SMAP | $0.9356 \pm 0.0009$ | $0.9001 \pm 0.0007$ |
| | SWAT | $0.9268 \pm 0.0001$ | $0.9183 \pm 0.0006$ |
| | PSM | $0.9774 \pm 0.0002$ | $0.9339 \pm 0.0925$ |
| | N2 | $0.3849 \pm 0.0299$ | $0.4643 \pm 0.0561$ |
| | N5 | $0.3848 \pm 0.0294$ | $0.4958 \pm 0.0396$ |
| | R | $0.3802 \pm 0.0502$ | $0.4630 \pm 0.0139$ |
| | B | $0.3686 \pm 0.0172$ | $0.2533 \pm 0.0498$ |
| | S | $0.3935 \pm 0.0325$ | $0.4772 \pm 0.5000$ |

Table 13: **Extra Anomaly Detection** (↑)**.** We present the the Adj. F1 metric the table (higher is better), then calculate the `AvgWins` . The selection criteria for the 15 datasets from (Wu & Keogh, 2021; Goswami et al., 2024) was the following. First, based only on the names in (Goswami et al., 2024), it was often ambiguous which data file was used. In these cases, we excluded the dataset. Second, we had difficulty verifying whether the default train/val/test ratios specified in the (Goswami et al., 2024) code matched what was reported. We found for the majority of datasets that the defaults resulted in test sets with no anomalies, when anomalies should be present. These were also excluded. From the results we could obtain, TOTEM matches or beats all other methods.

| Model | **TOTEM** | ATran | MNT-0 | MNT-LP | DGHL | GPT2 | TiNet |
|---|---|---|---|---|---|---|---|
| CIMIS44AirTemperature3 | 73.8 | 6.0 | **100.0** | 98.0 | 50.0 | 18.0 | 47.0 |
| GP711MarkerLFM5z4 | 96.7 | 76.0 | 69.0 | **97.0** | 31.0 | 48.0 | 90.0 |
| InternalBleeding5 | **100.0** | 94.0 | **100.0** | **100.0** | **100.0** | 92.0 | **100.0** |
| MesoplodonDensirostris | 99.4 | **100.0** | 91.0 | 84.0 | 79.0 | **100.0** | **100.0** |
| TKeepSecondMARS | **100.0** | 83.0 | 95.0 | **100.0** | 16.0 | 12.0 | 95.0 |
| WalkingAceleration5 | **100.0** | 99.0 | **100.0** | **100.0** | 91.0 | 87.0 | 93.0 |
| insectEPG2 | **100.0** | 12.0 | 11.0 | 23.0 | 14.0 | 81.0 | 96.0 |
| ltstdbs30791AS | **100.0** | **100.0** | **100.0** | **100.0** | **100.0** | **100.0** | **100.0** |
| park3m | 67.2 | 15.0 | 56.0 | 64.0 | 20.0 | 63.0 | **93.0** |
| s20101mML2 | **100.0** | 69.0 | 65.0 | 71.0 | 15.0 | 5.0 | 8.0 |
| sddb49 | 99.8 | 89.0 | **100.0** | **100.0** | 88.0 | 94.0 | **100.0** |
| sel840mECG1 | **99.5** | 16.0 | 61.0 | 66.0 | 28.0 | 21.0 | 36.0 |
| sel840mECG2 | **86.8** | 15.0 | 36.0 | 39.0 | 32.0 | 28.0 | 21.0 |
| tiltAPB2 | 68.5 | 92.0 | 96.0 | **98.0** | 36.0 | 83.0 | 38.0 |
| tiltAPB3 | 23.4 | 17.0 | 48.0 | **85.0** | 3.0 | 5.0 | 9.0 |
| AvgWins | **53.5%** | 13.3% | 33.3% | **53.5%** | 13.3% | 13.3% | 33.3% |
| Avg. Best Adj. F1 | **87.7** | 58.9 | 75.2 | 81.7 | 46.9 | 55.8 | 68.4 |

## A.4 Forecasting.

Table 14: **Specialist Forecasting** (↓)**.** TOTEM has the best `AvgWins` (**28.6%**), followed by iTrans (26.8%). Notably, TOTEM has first place finishes in 5 datasets, while iTrans' first places are concentrated in only electricity and traffic. All models have lookback $T_{in} = 96$.

| Model Metric | | TOTEM | | GPT2 | | TiNet | | iTrans | | Patch | | Cross | | FED | | Stat | | TiDE | | RLin | | DLin | | SCi |
|---|---|---|---|---|---|---|---|---|---|---|---|---|---|---|---|---|---|---|---|---|---|---|---|---|
| | | MSE | MAE | MSE | MAE | MSE | MAE | MSE | MAE | MSE | MAE | MSE | MAE | MSE | MAE | MSE | MAE | MSE | MAE | MSE | MAE | MSE | MAE | MSE | MAE |
| W | 96 | 0.165 | **0.208** | 0.184 | 0.224 | 0.172 | 0.220 | 0.174 | 0.214 | 0.177 | 0.218 | **0.158** | 0.230 | 0.217 | 0.296 | 0.173 | 0.223 | 0.202 | 0.261 | 0.192 | 0.232 | 0.196 | 0.255 | 0.221 | 0.306 |
| | 192 | 0.207 | **0.250** | 0.231 | 0.263 | 0.219 | 0.261 | 0.221 | 0.254 | 0.225 | 0.259 | **0.206** | 0.277 | 0.276 | 0.336 | 0.245 | 0.285 | 0.242 | 0.298 | 0.240 | 0.271 | 0.237 | 0.296 | 0.261 | 0.340 |
| | 336 | **0.257** | **0.291** | 0.285 | 0.302 | 0.280 | 0.306 | 0.278 | 0.296 | 0.278 | 0.297 | 0.272 | 0.335 | 0.339 | 0.380 | 0.321 | 0.338 | 0.287 | 0.335 | 0.292 | 0.307 | 0.283 | 0.335 | 0.309 | 0.378 |
| | 720 | **0.326** | **0.340** | 0.362 | 0.351 | 0.365 | 0.359 | 0.358 | 0.349 | 0.354 | 0.348 | 0.398 | 0.418 | 0.403 | 0.428 | 0.414 | 0.410 | 0.351 | 0.386 | 0.364 | 0.353 | 0.345 | 0.381 | 0.377 | 0.427 |
| E | 96 | 0.178 | 0.263 | 0.186 | 0.272 | 0.168 | 0.272 | **0.148** | **0.240** | 0.195 | 0.285 | 0.219 | 0.314 | 0.193 | 0.308 | 0.169 | 0.273 | 0.237 | 0.329 | 0.201 | 0.281 | 0.197 | 0.282 | 0.247 | 0.345 |
| | 192 | 0.187 | 0.272 | 0.190 | 0.278 | 0.184 | 0.289 | **0.162** | **0.253** | 0.199 | 0.289 | 0.231 | 0.322 | 0.201 | 0.315 | 0.182 | 0.286 | 0.236 | 0.330 | 0.201 | 0.283 | 0.196 | 0.285 | 0.257 | 0.355 |
| | 336 | 0.199 | 0.285 | 0.204 | 0.291 | 0.198 | 0.300 | **0.178** | **0.269** | 0.215 | 0.305 | 0.246 | 0.337 | 0.214 | 0.329 | 0.200 | 0.304 | 0.249 | 0.344 | 0.215 | 0.298 | 0.209 | 0.301 | 0.269 | 0.369 |
| | 720 | 0.236 | 0.318 | 0.245 | 0.324 | **0.220** | 0.320 | 0.225 | **0.317** | 0.256 | 0.337 | 0.280 | 0.363 | 0.246 | 0.355 | 0.222 | 0.321 | 0.284 | 0.373 | 0.257 | 0.331 | 0.245 | 0.333 | 0.299 | 0.390 |
| T | 96 | 0.523 | 0.303 | 0.471 | 0.311 | 0.593 | 0.321 | **0.395** | **0.268** | 0.544 | 0.359 | 0.522 | 0.290 | 0.587 | 0.366 | 0.612 | 0.338 | 0.805 | 0.493 | 0.649 | 0.389 | 0.650 | 0.396 | 0.788 | 0.499 |
| | 192 | 0.530 | 0.303 | 0.479 | 0.312 | 0.617 | 0.336 | **0.417** | **0.276** | 0.540 | 0.354 | 0.530 | 0.293 | 0.604 | 0.373 | 0.613 | 0.340 | 0.756 | 0.474 | 0.601 | 0.366 | 0.598 | 0.370 | 0.789 | 0.505 |
| | 336 | 0.549 | 0.311 | 0.490 | 0.317 | 0.629 | 0.336 | **0.433** | **0.283** | 0.551 | 0.358 | 0.558 | 0.305 | 0.621 | 0.383 | 0.618 | 0.328 | 0.762 | 0.477 | 0.609 | 0.369 | 0.605 | 0.373 | 0.797 | 0.508 |
| | 720 | 0.598 | 0.331 | 0.524 | 0.336 | 0.640 | 0.350 | **0.467** | **0.302** | 0.586 | 0.375 | 0.589 | 0.328 | 0.626 | 0.382 | 0.653 | 0.355 | 0.719 | 0.449 | 0.647 | 0.387 | 0.645 | 0.394 | 0.841 | 0.523 |
| m1 | 96 | **0.320** | **0.347** | 0.328 | 0.363 | 0.338 | 0.375 | 0.334 | 0.368 | 0.329 | 0.367 | 0.404 | 0.426 | 0.379 | 0.419 | 0.386 | 0.398 | 0.364 | 0.387 | 0.355 | 0.376 | 0.345 | 0.372 | 0.418 | 0.438 |
| | 192 | 0.379 | **0.382** | 0.368 | **0.382** | 0.374 | 0.387 | 0.377 | 0.391 | **0.367** | 0.385 | 0.450 | 0.451 | 0.426 | 0.441 | 0.459 | 0.444 | 0.398 | 0.404 | 0.391 | 0.392 | 0.380 | 0.389 | 0.439 | 0.450 |
| | 336 | 0.406 | **0.402** | 0.400 | 0.404 | 0.410 | 0.411 | 0.426 | 0.420 | **0.399** | 0.410 | 0.532 | 0.515 | 0.445 | 0.459 | 0.495 | 0.464 | 0.428 | 0.425 | 0.424 | 0.415 | 0.413 | 0.413 | 0.490 | 0.485 |
| | 720 | 0.471 | **0.438** | 0.462 | 0.440 | 0.478 | 0.450 | 0.491 | 0.459 | **0.454** | 0.439 | 0.666 | 0.589 | 0.543 | 0.490 | 0.585 | 0.516 | 0.487 | 0.461 | 0.487 | 0.450 | 0.474 | 0.453 | 0.595 | 0.550 |
| m2 | 96 | 0.176 | **0.253** | 0.178 | 0.263 | 0.187 | 0.267 | 0.180 | 0.264 | **0.175** | 0.259 | 0.287 | 0.366 | 0.203 | 0.287 | 0.192 | 0.274 | 0.207 | 0.305 | 0.182 | 0.265 | 0.193 | 0.292 | 0.286 | 0.377 |
| | 192 | 0.247 | **0.302** | 0.245 | 0.307 | 0.249 | 0.309 | 0.250 | 0.309 | **0.241** | **0.302** | 0.414 | 0.492 | 0.269 | 0.328 | 0.280 | 0.339 | 0.290 | 0.364 | 0.246 | 0.304 | 0.284 | 0.362 | 0.399 | 0.445 |
| | 336 | 0.317 | 0.348 | 0.307 | 0.346 | 0.321 | 0.351 | 0.311 | 0.348 | **0.305** | 0.343 | 0.597 | 0.542 | 0.325 | 0.366 | 0.334 | 0.361 | 0.377 | 0.422 | 0.307 | **0.342** | 0.369 | 0.427 | 0.637 | 0.591 |
| | 720 | 0.426 | 0.410 | 0.410 | 0.410 | 0.408 | 0.403 | 0.412 | 0.407 | **0.402** | 0.400 | 1.730 | 1.042 | 0.421 | 0.415 | 0.417 | 0.413 | 0.558 | 0.524 | 0.407 | **0.398** | 0.554 | 0.522 | 0.960 | 0.735 |
| h1 | 96 | 0.380 | **0.394** | 0.379 | 0.397 | 0.384 | 0.402 | 0.386 | 0.405 | 0.414 | 0.419 | 0.423 | 0.448 | **0.376** | 0.419 | 0.513 | 0.491 | 0.479 | 0.464 | 0.386 | 0.395 | 0.386 | 0.400 | 0.654 | 0.599 |
| | 192 | 0.434 | 0.427 | 0.434 | 0.427 | 0.436 | 0.429 | 0.441 | 0.436 | 0.460 | 0.445 | 0.471 | 0.474 | **0.420** | 0.448 | 0.534 | 0.504 | 0.525 | 0.492 | 0.437 | **0.424** | 0.437 | 0.432 | 0.719 | 0.631 |
| | 336 | 0.490 | 0.459 | 0.474 | 0.448 | 0.491 | 0.469 | 0.487 | 0.458 | 0.501 | 0.466 | 0.570 | 0.546 | **0.459** | 0.465 | 0.588 | 0.535 | 0.565 | 0.515 | 0.479 | **0.446** | 0.481 | 0.459 | 0.778 | 0.659 |
| | 720 | 0.539 | 0.513 | 0.496 | 0.475 | 0.521 | 0.500 | 0.503 | 0.491 | 0.500 | 0.488 | 0.653 | 0.621 | 0.506 | 0.507 | 0.643 | 0.616 | 0.594 | 0.558 | **0.481** | **0.470** | 0.519 | 0.516 | 0.836 | 0.699 |
| h2 | 96 | 0.293 | **0.338** | 0.295 | 0.348 | 0.340 | 0.374 | 0.297 | 0.349 | 0.302 | 0.348 | 0.745 | 0.584 | 0.358 | 0.397 | 0.476 | 0.458 | 0.400 | 0.440 | **0.288** | **0.338** | 0.333 | 0.387 | 0.707 | 0.621 |
| | 192 | 0.375 | **0.390** | 0.384 | 0.402 | 0.402 | 0.414 | 0.380 | 0.400 | 0.388 | 0.400 | 0.877 | 0.656 | 0.429 | 0.439 | 0.512 | 0.493 | 0.528 | 0.509 | **0.374** | **0.390** | 0.477 | 0.476 | 0.860 | 0.689 |
| | 336 | 0.422 | 0.431 | 0.418 | 0.432 | 0.452 | 0.452 | 0.428 | 0.432 | 0.426 | 0.433 | 1.043 | 0.731 | 0.496 | 0.487 | 0.552 | 0.551 | 0.643 | 0.571 | **0.415** | **0.426** | 0.594 | 0.541 | 1.000 | 0.744 |
| | 720 | 0.610 | 0.567 | 0.423 | 0.446 | 0.462 | 0.468 | 0.427 | 0.445 | 0.431 | 0.446 | 1.104 | 0.763 | 0.463 | 0.474 | 0.562 | 0.560 | 0.874 | 0.679 | **0.420** | **0.440** | 0.831 | 0.657 | 1.249 | 0.838 |
| AvgWins | | **28.6%** | | 1.8% | | 1.8% | | 26.8% | | 14.3% | | 3.6% | | 5.4% | | 0% | | 0% | | 25% | | 0% | | 0% | |

Table 15: **Generalist Forecasting** (↓)**.** Here we evaluate the generalist TOTEM and GPT2 models. **A. In-Domain Performance.** TOTEM outperforms GPT2: 67.9% to 33.9%. **B. Zero-Shot Performance.** TOTEM outperforms GPT2: 90.0% to 12.5%.

**A. In-Domain Performance**

| Model Metric | | TOTEM | | GPT2 | |
|---|---|---|---|---|---|
| | | MSE | MAE | MSE | MAE |
| W | 96 | **0.172** | **0.216** | 0.201 | 0.237 |
| | 192 | **0.217** | **0.256** | 0.247 | 0.275 |
| | 336 | **0.266** | **0.295** | 0.298 | 0.311 |
| | 720 | **0.334** | **0.342** | 0.372 | 0.360 |
| E | 96 | **0.179** | **0.264** | 0.194 | 0.278 |
| | 192 | **0.181** | **0.267** | 0.199 | 0.284 |
| | 336 | **0.196** | **0.283** | 0.214 | 0.300 |
| | 720 | **0.230** | **0.314** | 0.255 | 0.331 |
| T | 96 | 0.507 | **0.284** | **0.484** | 0.320 |
| | 192 | 0.511 | **0.282** | **0.488** | 0.320 |
| | 336 | 0.535 | **0.292** | **0.502** | 0.326 |
| | 720 | 0.580 | **0.309** | **0.534** | 0.343 |
| m1 | 96 | **0.374** | **0.384** | 0.487 | 0.468 |
| | 192 | **0.400** | **0.399** | 0.516 | 0.480 |
| | 336 | **0.432** | **0.424** | 0.548 | 0.499 |
| | 720 | **0.487** | **0.460** | 0.581 | 0.511 |
| m2 | 96 | **0.198** | **0.275** | 0.243 | 0.315 |
| | 192 | **0.266** | **0.319** | 0.297 | 0.346 |
| | 336 | 0.365 | 0.377 | **0.349** | **0.376** |
| | 720 | 0.588 | 0.511 | **0.439** | **0.423** |
| h1 | 96 | **0.382** | **0.404** | 0.421 | 0.408 |
| | 192 | **0.463** | **0.435** | 0.480 | 0.436 |
| | 336 | **0.507** | 0.463 | 0.518 | **0.453** |
| | 720 | **0.517** | 0.500 | **0.517** | **0.467** |
| h2 | 96 | 0.307 | 0.345 | **0.298** | **0.343** |
| | 192 | 0.406 | 0.403 | **0.381** | **0.392** |
| | 336 | 0.505 | 0.460 | **0.406** | **0.419** |
| | 720 | 0.661 | 0.557 | **0.423** | **0.438** |
| AvgWins | | **67.9%** | | 33.9% | |

**B. Zero-Shot Performance**

| Model Metric | | TOTEM | | GPT2 | |
|---|---|---|---|---|---|
| | | MSE | MAE | MSE | MAE |
| N2 | 96 | **1.138** | **0.777** | 1.332 | 0.830 |
| | 192 | **1.149** | **0.785** | 1.416 | 0.863 |
| | 336 | **1.092** | **0.770** | 1.358 | 0.851 |
| | 720 | **1.045** | **0.754** | 1.308 | 0.840 |
| N5 | 96 | **0.483** | **0.484** | 0.528 | 0.499 |
| | 192 | **0.495** | **0.491** | 0.578 | 0.524 |
| | 336 | **0.468** | **0.483** | 0.548 | 0.515 |
| | 720 | **0.451** | **0.477** | 0.537 | 0.511 |
| R | 96 | **1.120** | **0.582** | 1.465 | 0.725 |
| | 192 | **1.242** | **0.635** | 1.638 | 0.785 |
| | 336 | **1.237** | **0.626** | 1.601 | 0.769 |
| | 720 | **1.182** | **0.604** | 1.552 | 0.760 |
| B | 96 | **0.805** | **0.739** | 0.838 | 0.762 |
| | 192 | **0.836** | 0.752 | 0.837 | **0.752** |
| | 336 | 0.809 | 0.748 | **0.792** | **0.738** |
| | 720 | **0.896** | **0.794** | 0.927 | 0.806 |
| S | 96 | 0.446 | 0.482 | **0.443** | **0.478** |
| | 192 | **0.462** | **0.491** | 0.481 | 0.499 |
| | 336 | **0.521** | **0.525** | 0.541 | 0.533 |
| | 720 | **0.717** | **0.625** | 0.773 | 0.643 |
| AvgWins | | **90.0%** | | 12.5% | |

Table 16: **Means and Stds. for the Forecasting Specialits.** A. is the TOTEM specialist, B. is the GPT2 specialist which we setup to run with a consistent lookback.

**A. TOTEM - Specialist Forecasting (↓)**

| Metric | | MSE (Mean ± Std) | MAE (Mean ± Std) |
|---|---|---|---|
| W | 96 | 0.165 ± 0.0015 | 0.208 ± 0.0012 |
| | 192 | 0.207 ± 0.0006 | 0.250 ± 0.0012 |
| | 336 | 0.257 ± 0.0002 | 0.291 ± 0.0006 |
| | 720 | 0.326 ± 0.0035 | 0.340 ± 0.0023 |
| E | 96 | 0.178 ± 0.0015 | 0.263 ± 0.0010 |
| | 192 | 0.187 ± 0.0015 | 0.272 ± 0.0015 |
| | 336 | 0.199 ± 0.0012 | 0.285 ± 0.0012 |
| | 720 | 0.236 ± 0.0035 | 0.318 ± 0.0031 |
| T | 96 | 0.523 ± 0.0010 | 0.303 ± 0.0006 |
| | 192 | 0.530 ± 0.0030 | 0.303 ± 0.0017 |
| | 336 | 0.549 ± 0.0017 | 0.311 ± 0.0021 |
| | 720 | 0.598 ± 0.0095 | 0.331 ± 0.0062 |
| m1 | 96 | 0.320 ± 0.0006 | 0.347 ± 0.0006 |
| | 192 | 0.379 ± 0.0017 | 0.382 ± 0.0012 |
| | 336 | 0.406 ± 0.0040 | 0.402 ± 0.0026 |
| | 720 | 0.471 ± 0.0006 | 0.438 ± 0.0010 |
| m2 | 96 | 0.176 ± 0.0006 | 0.253 ± 0.0010 |
| | 192 | 0.247 ± 0.0012 | 0.302 ± 0.0015 |
| | 336 | 0.317 ± 0.0046 | 0.348 ± 0.0031 |
| | 720 | 0.426 ± 0.0085 | 0.410 ± 0.0062 |
| h1 | 96 | 0.380 ± 0.0006 | 0.394 ± 0.0000 |
| | 192 | 0.434 ± 0.0010 | 0.427 ± 0.0006 |
| | 336 | 0.490 ± 0.0023 | 0.448 ± 0.0015 |
| | 720 | 0.539 ± 0.0031 | 0.513 ± 0.0020 |
| h2 | 96 | 0.293 ± 0.0015 | 0.338 ± 0.0006 |
| | 192 | 0.375 ± 0.0031 | 0.390 ± 0.0026 |
| | 336 | 0.422 ± 0.0046 | 0.431 ± 0.0031 |
| | 720 | 0.610 ± 0.0095 | 0.567 ± 0.0081 |

**B. GPT2 - Specialist Forecasting, Lookback of 96 ↓**

| Metric | | MSE (Mean ± Std) | MAE (Mean ± Std) |
|---|---|---|---|
| W | 96 | 0.184 ± 0.0013 | 0.224 ± 0.0014 |
| | 192 | 0.231 ± 0.0012 | 0.263 ± 0.0009 |
| | 336 | 0.285 ± 0.0015 | 0.302 ± 0.0013 |
| | 720 | 0.362 ± 0.0016 | 0.351 ± 0.0008 |
| E | 96 | 0.186 ± 0.0004 | 0.272 ± 0.0005 |
| | 192 | 0.190 ± 0.0007 | 0.278 ± 0.0008 |
| | 336 | 0.204 ± 0.0003 | 0.291 ± 0.0005 |
| | 720 | 0.245 ± 0.0012 | 0.324 ± 0.0014 |
| T | 96 | 0.471 ± 0.0016 | 0.311 ± 0.0016 |
| | 192 | 0.479 ± 0.0017 | 0.312 ± 0.0010 |
| | 336 | 0.490 ± 0.0009 | 0.317 ± 0.0010 |
| | 720 | 0.524 ± 0.0019 | 0.336 ± 0.0018 |
| m1 | 96 | 0.328 ± 0.0022 | 0.363 ± 0.0014 |
| | 192 | 0.368 ± 0.0006 | 0.382 ± 0.0004 |
| | 336 | 0.400 ± 0.0013 | 0.404 ± 0.0011 |
| | 720 | 0.462 ± 0.0010 | 0.440 ± 0.0009 |
| m2 | 96 | 0.178 ± 0.0000 | 0.263 ± 0.0000 |
| | 192 | 0.245 ± 0.0000 | 0.307 ± 0.0000 |
| | 336 | 0.307 ± 0.0000 | 0.346 ± 0.0000 |
| | 720 | 0.410 ± 0.0000 | 0.409 ± 0.0000 |
| h1 | 96 | 0.379 ± 0.0032 | 0.397 ± 0.0007 |
| | 192 | 0.438 ± 0.0037 | 0.427 ± 0.0004 |
| | 336 | 0.474 ± 0.0045 | 0.448 ± 0.0004 |
| | 720 | 0.496 ± 0.0066 | 0.475 ± 0.0033 |
| h2 | 96 | 0.295 ± 0.0000 | 0.348 ± 0.0000 |
| | 192 | 0.384 ± 0.0000 | 0.402 ± 0.0000 |
| | 336 | 0.418 ± 0.0000 | 0.432 ± 0.0000 |
| | 720 | 0.423 ± 0.0000 | 0.446 ± 0.0000 |

Table 17: **Means and Stds. for the Forecasting Generalist.** A. is the TOTEM generalist, B. is the GPT2 generalist which we setup to run in a generalist manner.

**A. TOTEM - Generalist and Zero-Shot Forecasting (↓)**

| Metric | | MSE (Mean ± Std) | MAE (Mean ± Std) |
|---|---|---|---|
| W | 96 | 0.172 ± 0.0010 | 0.216 ± 0.0006 |
| | 192 | 0.217 ± 0.0006 | 0.256 ± 0.0006 |
| | 336 | 0.266 ± 0.0015 | 0.295 ± 0.0015 |
| | 720 | 0.334 ± 0.0010 | 0.342 ± 0.0012 |
| E | 96 | 0.179 ± 0.0006 | 0.264 ± 0.0012 |
| | 192 | 0.181 ± 0.0006 | 0.267 ± 0.0006 |
| | 336 | 0.196 ± 0.0020 | 0.283 ± 0.0015 |
| | 720 | 0.230 ± 0.0035 | 0.314 ± 0.0029 |
| T | 96 | 0.507 ± 0.0020 | 0.284 ± 0.0006 |
| | 192 | 0.511 ± 0.0030 | 0.282 ± 0.0006 |
| | 336 | 0.535 ± 0.0076 | 0.292 ± 0.0012 |
| | 720 | 0.580 ± 0.0046 | 0.309 ± 0.0006 |
| m1 | 96 | 0.374 ± 0.0000 | 0.384 ± 0.0006 |
| | 192 | 0.400 ± 0.0015 | 0.399 ± 0.0023 |
| | 336 | 0.432 ± 0.0040 | 0.424 ± 0.0015 |
| | 720 | 0.487 ± 0.0081 | 0.460 ± 0.0017 |
| m2 | 96 | 0.198 ± 0.0006 | 0.275 ± 0.0012 |
| | 192 | 0.266 ± 0.0035 | 0.319 ± 0.0021 |
| | 336 | 0.365 ± 0.0115 | 0.349 ± 0.0025 |
| | 720 | 0.588 ± 0.0699 | 0.511 ± 0.0281 |
| h1 | 96 | 0.382 ± 0.0364 | 0.404 ± 0.0012 |
| | 192 | 0.463 ± 0.0025 | 0.435 ± 0.0006 |
| | 336 | 0.507 ± 0.0025 | 0.463 ± 0.0161 |
| | 720 | 0.570 ± 0.0010 | 0.508 ± 0.0017 |
| h2 | 96 | 0.307 ± 0.0012 | 0.345 ± 0.0015 |
| | 192 | 0.406 ± 0.0038 | 0.403 ± 0.0023 |
| | 336 | 0.505 ± 0.0114 | 0.460 ± 0.0035 |
| | 720 | 0.661 ± 0.0514 | 0.557 ± 0.0215 |
| *Zero-Shot* | | | |
| N2 | 96 | 1.138 ± 0.0032 | 0.777 ± 0.0012 |
| | 192 | 1.149 ± 0.0026 | 0.785 ± 0.0012 |
| | 336 | 1.092 ± 0.0062 | 0.770 ± 0.0023 |
| | 720 | 1.045 ± 0.0040 | 0.754 ± 0.0023 |
| N5 | 96 | 0.483 ± 0.0012 | 0.484 ± 0.0012 |
| | 192 | 0.495 ± 0.0021 | 0.491 ± 0.0015 |
| | 336 | 0.468 ± 0.0035 | 0.483 ± 0.0029 |
| | 720 | 0.451 ± 0.0023 | 0.477 ± 0.0023 |
| R | 96 | 1.120 ± 0.0081 | 0.582 ± 0.0036 |
| | 192 | 1.242 ± 0.0151 | 0.635 ± 0.0074 |
| | 336 | 1.237 ± 0.0153 | 0.626 ± 0.0076 |
| | 720 | 1.182 ± 0.0151 | 0.604 ± 0.0050 |
| B | 96 | 0.805 ± 0.0070 | 0.739 ± 0.0035 |
| | 192 | 0.836 ± 0.0040 | 0.752 ± 0.0021 |
| | 336 | 0.809 ± 0.0038 | 0.748 ± 0.0021 |
| | 720 | 0.896 ± 0.0137 | 0.794 ± 0.0085 |
| S | 96 | 0.446 ± 0.0032 | 0.482 ± 0.0017 |
| | 192 | 0.462 ± 0.0015 | 0.491 ± 0.0010 |
| | 336 | 0.551 ± 0.0122 | 0.529 ± 0.0008 |
| | 720 | 0.717 ± 0.0096 | 0.625 ± 0.0040 |

**B. GPT2 - Generalist and Zero-Shot Forecasting (↓)**

| Metric | | MSE (Mean ± Std) | MAE (Mean ± Std) |
|---|---|---|---|
| W | 96 | 0.201 ± 0.0017 | 0.237 ± 0.0012 |
| | 192 | 0.247 ± 0.0020 | 0.275 ± 0.0015 |
| | 336 | 0.298 ± 0.0006 | 0.311 ± 0.0006 |
| | 720 | 0.372 ± 0.0010 | 0.360 ± 0.0006 |
| E | 96 | 0.194 ± 0.0012 | 0.275 ± 0.0021 |
| | 192 | 0.199 ± 0.0006 | 0.283 ± 0.0006 |
| | 336 | 0.214 ± 0.0012 | 0.300 ± 0.0015 |
| | 720 | 0.255 ± 0.0006 | 0.331 ± 0.0012 |
| T | 96 | 0.484 ± 0.0046 | 0.320 ± 0.0042 |
| | 192 | 0.488 ± 0.0006 | 0.320 ± 0.0006 |
| | 336 | 0.502 ± 0.0020 | 0.326 ± 0.0021 |
| | 720 | 0.534 ± 0.0021 | 0.343 ± 0.0021 |
| m1 | 96 | 0.487 ± 0.0106 | 0.468 ± 0.0035 |
| | 192 | 0.516 ± 0.0071 | 0.480 ± 0.0021 |
| | 336 | 0.548 ± 0.0015 | 0.499 ± 0.0012 |
| | 720 | 0.581 ± 0.0031 | 0.511 ± 0.0012 |
| m2 | 96 | 0.243 ± 0.0021 | 0.315 ± 0.0021 |
| | 192 | 0.297 ± 0.0012 | 0.346 ± 0.0010 |
| | 336 | 0.349 ± 0.0025 | 0.376 ± 0.0020 |
| | 720 | 0.439 ± 0.0010 | 0.423 ± 0.0010 |
| h1 | 96 | 0.421 ± 0.0058 | 0.408 ± 0.0010 |
| | 192 | 0.480 ± 0.0026 | 0.436 ± 0.0020 |
| | 336 | 0.518 ± 0.0161 | 0.453 ± 0.0070 |
| | 720 | 0.517 ± 0.0036 | 0.467 ± 0.0035 |
| h2 | 96 | 0.298 ± 0.0090 | 0.343 ± 0.0049 |
| | 192 | 0.381 ± 0.0153 | 0.392 ± 0.0072 |
| | 336 | 0.406 ± 0.0271 | 0.419 ± 0.0144 |
| | 720 | 0.423 ± 0.0078 | 0.438 ± 0.0051 |
| *Zero-Shot* | | | |
| N2 | 96 | 1.332 ± 0.0012 | 0.830 ± 0.0010 |
| | 192 | 1.416 ± 0.0080 | 0.863 ± 0.0025 |
| | 336 | 1.358 ± 0.0123 | 0.851 ± 0.0042 |
| | 720 | 1.308 ± 0.0026 | 0.840 ± 0.0010 |
| N5 | 96 | 0.528 ± 0.0006 | 0.499 ± 0.0010 |
| | 192 | 0.578 ± 0.0015 | 0.524 ± 0.0006 |
| | 336 | 0.548 ± 0.0040 | 0.515 ± 0.0015 |
| | 720 | 0.537 ± 0.0006 | 0.511 ± 0.0006 |
| R | 96 | 1.465 ± 0.0185 | 0.725 ± 0.0031 |
| | 192 | 1.638 ± 0.0280 | 0.785 ± 0.0078 |
| | 336 | 1.601 ± 0.0244 | 0.769 ± 0.0060 |
| | 720 | 1.552 ± 0.0110 | 0.760 ± 0.0035 |
| B | 96 | 0.838 ± 0.0149 | 0.762 ± 0.0071 |
| | 192 | 0.837 ± 0.0095 | 0.752 ± 0.0040 |
| | 336 | 0.792 ± 0.0104 | 0.738 ± 0.0050 |
| | 720 | 0.927 ± 0.0066 | 0.806 ± 0.0038 |
| S | 96 | 0.443 ± 0.0010 | 0.478 ± 0.0006 |
| | 192 | 0.481 ± 0.0006 | 0.499 ± 0.0006 |
| | 336 | 0.541 ± 0.0010 | 0.533 ± 0.0006 |
| | 720 | 0.773 ± 0.0020 | 0.643 ± 0.0010 |

| Metric | Tin →Tout | Train | Test | TOTEM (Ours) | GPT2 | TiNet | Patch | DLin | Re | Inf | Auto | Fed | LiTS |
|---|---|---|---|---|---|---|---|---|---|---|---|---|---|
| sMAPE | 24 →18 | M4-M | M3-M | 14.4 | 14.1 | 14.0 | 14.7 | 15.7 | 14.8 | 15.9 | 16.9 | 15.1 | 24.6 |
| sMAPE | 48 →24 | M3-M | M4-M | 14.6 | 14.6 | 16.2 | 14.7 | 14.8 | 15.6 | 23.5 | 25.1 | 18.2 | 15.2 |
| MAPE | 12 →4 | M4-Y | Tour.-Y | 31.8 | 27.2 | 35.6 | 33.2 | 39.6 | 33.9 | 41.2 | 51.2 | 43.4 | 138.2 |
| NDx100 | 30 →168 | M4-H | Elec.-H | 17.6 | 17.2 | 19.3 | 17.3 | 17.6 | 21.6 | 21.2 | 33.9 | 18.4 | 19.6 |

Table 18: Short term forecasting results (lower is better). We randomly choose settings across varying input-to-output dimensionalites, train and test datasets, and find that TOTEM (Ours) and GPT2 outperform all other methods.

Table 19: **Long term vs. short term forecasting lookback and lookahead lengths.** We see that long term forecasting is far more stereotyped, and therefore easier to build generalist models for, than short term forecasting.

| Dataset | Input →Output |
|---|---|
| Long Term Forecasting; In-Domain Testing | |
| All Datasets (enforced by us, Liu et al. (2023); Wu et al. (2022); Liu et al. (2022b); Zhou et al. (2022) | $96 \to 96, 192, 336, 720$ |
| Long Term Forecasting; Zero Shot Testing | |
| All Datasets | $96 \to 96, 192, 336, 720$ |
| Short Term Forecasting; In Domain Testing | |
| M4-Y | $12 \to 6$ |
| M4-Q | $16 \to 8$ |
| M4-M | $36 \to 18$ |
| M4-W | $26 \to 13$ |
| M4-D | $28 \to 14$ |
| M4-H | $96 \to 48$ |
| Short Term Forecasting; Zero Shot Testing | |
| M4-Y, M3-Y | $12 \to 6$ |
| M4-Q, M3-Q | $24 \to 8$ |
| M4-M, M3-M | $24 \to 18$ |
| M4-M, M3-O | $16 \to 8$ |
| M3-Q, M4-Q | $16 \to 8$ |
| M3-M, M4-M | $48 \to 24$ |
| M3-Y, M4-Y | $9 \to 6$ |
| M3-M, M4-W | $65 \to 13$ |
| M3-M, M4-D | $9 \to 14$ |
| M3-O, M4-H | $2 \to 48$ |
| M4-Y, Tour.-Y | $12 \to 4$ |
| M4-Q, Tour.-Q | $24 \to 8$ |
| M4-M, Tour.-M | $36 \to 24$ |
| M4-H, Elec.-H | $30 \to 168$ |
| *Y=Yearly, Q=Quarterly, M=Monthly, W=Weekly, D=Daily, H=Hourly, O=Other | |

Table 20: **96 and 512 Lookback Lengths.** We compare various forecasters with a lookback length of 96 and 512, across all lookback lengths and datasets TOTEM has the most `AvgWins` at 58.3% followed by GPT2 at 8.3%.

| Tin=512 | Model | TOTEM (Ours) | GPT2 | MNT | Patch | N-Beats |
|---|---|---|---|---|---|---|
| | Run By | TOTEM (Ours) | GPT2 | MNT | Patch | MNT |
| Dataset | Metric | MSE, MAE | MSE, MAE | MSE, MAE | MSE, MAE | MSE, MAE |
| W | 96 | 0.147, 0.196 | 0.162, 0.212 | 0.154, 0.209 | 0.149, 0.198 | 0.152, 0.210 |
| W | 192 | 0.195, 0.242 | 0.204, 0.248 | N/A, N/A | 0.194, 0.241 | N/A, N/A |
| W | 336 | 0.248, 0.283 | 0.254, 0.286 | N/A, N/A | 0.245, 0.282 | N/A, N/A |
| W | 720 | 0.314, 0.330 | 0.326, 0.337 | 0.315, 0.336 | 0.314, 0.334 | 0.331, 0.359 |
| E | 96 | 0.135, 0.231 | 0.139, 0.238 | 0.138, 0.242 | 0.129, 0.222 | 0.131, 0.228 |
| E | 192 | 0.151, 0.245 | 0.153, 0.251 | N/A, N/A | 0.147, 0.240 | N/A, N/A |
| E | 336 | 0.168, 0.265 | 0.169, 0.266 | N/A, N/A | 0.163, 0.259 | N/A, N/A |
| E | 720 | 0.200, 0.292 | 0.206, 0.297 | 0.211, 0.305 | 0.197, 0.290 | 0.208, 0.298 |
| T | 96 | 0.369, 0.241 | 0.388, 0.282 | 0.391, 0.282 | 0.360, 0.249 | 0.375, 0.259 |
| T | 192 | 0.383, 0.242 | 0.407, 0.290 | N/A, N/A | 0.379, 0.256 | N/A, N/A |
| T | 336 | 0.397, 0.248 | 0.412, 0.294 | N/A, N/A | 0.392, 0.264 | N/A, N/A |
| T | 720 | 0.446, 0.275 | 0.450, 0.312 | 0.450, 0.310 | 0.431, 0.286 | 0.508, 0.335 |
| Tin=96 | Model | TOTEM (Ours) | GPT2 | MNT | Patch | N-Beats |
| | Run By | TOTEM (Ours) | TOTEM (Ours) | N/A | Trans | N/A |
| W | 96 | 0.165, 0.208 | 0.184, 0.224 | N/A, N/A | 0.177, 0.218 | N/A, N/A |
| W | 192 | 0.207, 0.250 | 0.231, 0.263 | N/A, N/A | 0.225, 0.259 | N/A, N/A |
| W | 336 | 0.257, 0.291 | 0.285, 0.302 | N/A, N/A | 0.278, 0.297 | N/A, N/A |
| W | 720 | 0.326, 0.340 | 0.362, 0.351 | N/A, N/A | 0.354, 0.348 | N/A, N/A |
| E | 96 | 0.178, 0.263 | 0.186, 0.272 | N/A, N/A | 0.195, 0.285 | N/A, N/A |
| E | 192 | 0.187, 0.272 | 0.190, 0.278 | N/A, N/A | 0.199, 0.289 | N/A, N/A |
| E | 336 | 0.199, 0.285 | 0.204, 0.291 | N/A, N/A | 0.215, 0.305 | N/A, N/A |
| E | 720 | 0.236, 0.318 | 0.245, 0.324 | N/A, N/A | 0.256, 0.337 | N/A, N/A |
| T | 96 | 0.523, 0.303 | 0.471, 0.311 | N/A, N/A | 0.544, 0.359 | N/A, N/A |
| T | 192 | 0.530, 0.303 | 0.479, 0.312 | N/A, N/A | 0.540, 0.354 | N/A, N/A |
| T | 336 | 0.549, 0.311 | 0.490, 0.317 | N/A, N/A | 0.551, 0.358 | N/A, N/A |
| T | 720 | 0.598, 0.331 | 0.524, 0.336 | N/A, N/A | 0.586, 0.375 | N/A, N/A |
| | AvgWins | 58.3% | 8.3% | 0% | 35.4% | 0% |

## A.5 Ablation Details.

Table 21: **Ablations** (↓)**.** Across the Tokens vs. Time (TvT) experiments tokens out perform time. (A) specialist: 67.9% to 39.3%, (B) in-domain generalist: 78.6% to 23.2% , and (C) zero-shot generalist: 67.5% to 35%. (D) As the codebook size $K$ increases the VQVAE reconstruction performance improves.

### A. TvT Specialist

| Metric | Horizon | TOTEM MSE | TOTEM MAE | TimeTOTEM MSE | TimeTOTEM MAE |
|---|---|---|---|---|---|
| W | 96 | 0.165 | **0.208** | **0.164** | 0.209 |
| | 192 | **0.207** | **0.250** | 0.209 | 0.251 |
| | 336 | **0.257** | **0.291** | 0.261 | 0.293 |
| | 720 | **0.326** | **0.340** | 0.332 | **0.340** |
| E | 96 | **0.178** | 0.263 | 0.179 | **0.262** |
| | 192 | 0.187 | 0.272 | **0.185** | **0.269** |
| | 336 | **0.199** | **0.285** | 0.204 | 0.289 |
| | 720 | **0.236** | **0.318** | 0.244 | 0.325 |
| T | 96 | **0.523** | **0.303** | 0.528 | 0.310 |
| | 192 | 0.530 | **0.303** | **0.500** | 0.349 |
| | 336 | 0.549 | **0.311** | 0.531 | 0.365 |
| | 720 | 0.598 | **0.331** | **0.578** | 0.398 |
| m1 | 96 | **0.320** | **0.347** | 0.326 | 0.355 |
| | 192 | 0.379 | **0.382** | **0.377** | 0.386 |
| | 336 | **0.406** | **0.402** | 0.409 | 0.409 |
| | 720 | 0.471 | **0.438** | **0.469** | 0.441 |
| m2 | 96 | **0.176** | **0.253** | 0.176 | 0.254 |
| | 192 | **0.247** | **0.302** | 0.247 | 0.303 |
| | 336 | **0.317** | **0.348** | 0.318 | 0.350 |
| | 720 | 0.426 | **0.410** | 0.419 | 0.411 |
| h1 | 96 | 0.380 | **0.394** | **0.377** | 0.395 |
| | 192 | 0.434 | **0.427** | **0.428** | 0.428 |
| | 336 | 0.490 | **0.459** | **0.480** | 0.462 |
| | 720 | 0.539 | **0.513** | **0.530** | 0.522 |
| h2 | 96 | **0.293** | **0.338** | 0.294 | **0.338** |
| | 192 | 0.375 | 0.390 | **0.373** | **0.389** |
| | 336 | **0.422** | **0.431** | 0.423 | 0.433 |
| | 720 | 0.610 | 0.567 | **0.591** | **0.556** |
| AvgWins | | **67.9%** | | 39.3% | |

### B. TvT In-Domain Generalist

| Metric | Horizon | TOTEM MSE | TOTEM MAE | TimeTOTEM MSE | TimeTOTEM MAE |
|---|---|---|---|---|---|
| W | 96 | **0.172** | **0.216** | 0.173 | 0.218 |
| | 192 | **0.217** | **0.256** | 0.218 | 0.261 |
| | 336 | **0.266** | **0.295** | 0.267 | 0.299 |
| | 720 | **0.334** | **0.342** | 0.337 | 0.347 |
| E | 96 | **0.179** | **0.264** | 0.183 | 0.267 |
| | 192 | **0.181** | **0.267** | 0.189 | 0.275 |
| | 336 | **0.196** | **0.283** | 0.204 | 0.291 |
| | 720 | **0.230** | **0.314** | 0.242 | 0.325 |
| T | 96 | **0.507** | **0.284** | 0.517 | 0.293 |
| | 192 | **0.511** | **0.282** | 0.526 | 0.296 |
| | 336 | **0.535** | **0.292** | 0.552 | 0.304 |
| | 720 | **0.580** | **0.309** | 0.602 | 0.326 |
| m1 | 96 | **0.374** | **0.384** | 0.428 | 0.420 |
| | 192 | **0.400** | **0.399** | 0.438 | 0.427 |
| | 336 | **0.432** | **0.424** | 0.469 | 0.447 |
| | 720 | **0.487** | **0.460** | 0.546 | 0.493 |
| m2 | 96 | **0.198** | **0.275** | 0.207 | 0.286 |
| | 192 | **0.266** | **0.319** | 0.269 | 0.325 |
| | 336 | 0.365 | **0.377** | 0.358 | **0.377** |
| | 720 | 0.588 | 0.511 | **0.521** | **0.482** |
| h1 | 96 | **0.382** | **0.404** | 0.401 | 0.410 |
| | 192 | 0.463 | **0.435** | **0.453** | 0.441 |
| | 336 | 0.507 | **0.463** | **0.496** | 0.468 |
| | 720 | **0.517** | **0.500** | 0.518 | 0.510 |
| h2 | 96 | 0.307 | **0.345** | **0.305** | 0.346 |
| | 192 | 0.406 | 0.403 | **0.396** | **0.402** |
| | 336 | 0.505 | 0.460 | **0.492** | **0.458** |
| | 720 | 0.661 | 0.557 | **0.599** | **0.531** |
| AvgWins | | **78.6%** | | 23.2% | |

### C. TvT Zero-Shot Generalist

| Metric | Horizon | TOTEM MSE | TOTEM MAE | TimeTOTEM MSE | TimeTOTEM MAE |
|---|---|---|---|---|---|
| N2 | 96 | 1.138 | 0.777 | **1.127** | **0.773** |
| | 192 | **1.149** | **0.785** | 1.169 | 0.793 |
| | 336 | **1.092** | **0.770** | 1.115 | 0.780 |
| | 720 | **1.045** | **0.754** | 1.070 | 0.766 |
| N5 | 96 | 0.483 | 0.484 | **0.481** | **0.483** |
| | 192 | **0.495** | **0.491** | 0.508 | 0.500 |
| | 336 | **0.468** | **0.483** | 0.481 | 0.491 |
| | 720 | **0.451** | **0.477** | 0.467 | 0.488 |
| R | 96 | 1.120 | 0.582 | **1.102** | **0.578** |
| | 192 | 1.242 | 0.635 | **1.207** | **0.628** |
| | 336 | 1.237 | 0.626 | **1.190** | **0.613** |
| | 720 | 1.182 | 0.604 | **1.149** | **0.596** |
| B | 96 | **0.805** | **0.739** | 0.825 | 0.751 |
| | 192 | **0.836** | **0.752** | 0.847 | 0.761 |
| | 336 | **0.809** | **0.748** | 0.831 | 0.764 |
| | 720 | **0.896** | **0.794** | 0.928 | 0.813 |
| S | 96 | **0.446** | 0.482 | **0.446** | **0.481** |
| | 192 | **0.462** | **0.491** | 0.478 | 0.499 |
| | 336 | **0.521** | **0.525** | 0.535 | 0.532 |
| | 720 | **0.717** | **0.625** | 0.736 | 0.631 |
| AvgWins | | **67.5%** | | 35.0% | |

### D. Codebook Size Ablations

| | Codebook Size $K$ = 32 | 256 | 512 |
|---|---|---|---|
| **MSE** | | | |
| All | 0.0451 | 0.0192 | **0.0184** |
| T | 0.0312 | 0.0120 | **0.0101** |
| E | 0.0463 | 0.0209 | **0.0152** |
| W | 0.0393 | 0.0161 | **0.0128** |
| **MAE** | | | |
| All | 0.1460 | 0.0937 | **0.0913** |
| T | 0.1204 | 0.0749 | **0.0685** |
| E | 0.1520 | 0.1027 | **0.0878** |
| W | 0.1122 | 0.0673 | **0.0607** |
| AvgWins | 0% | 0% | 100% |

### E. TvT MLP Specialist

| Metric | Horizon | TOTEM MSE | TOTEM MAE | TimeTOTEM MSE | TimeTOTEM MAE |
|---|---|---|---|---|---|
| W | 96 | **0.164** | **0.210** | 0.180 | 0.224 |
| | 192 | **0.207** | **0.252** | 0.212 | 0.254 |
| | 336 | **0.259** | **0.293** | 0.273 | 0.302 |
| | 720 | **0.330** | **0.342** | 0.345 | 0.350 |
| E | 96 | **0.183** | 0.268 | 0.186 | **0.265** |
| | 192 | **0.188** | 0.275 | 0.190 | **0.271** |
| | 336 | **0.203** | 0.290 | **0.203** | **0.285** |
| | 720 | **0.240** | 0.323 | **0.240** | **0.319** |
| T | 96 | **0.539** | **0.330** | 0.556 | 0.332 |
| | 192 | **0.551** | 0.332 | 0.567 | **0.326** |
| | 336 | **0.565** | 0.336 | 0.577 | **0.329** |
| | 720 | **0.608** | 0.354 | 0.622 | **0.351** |
| m1 | 96 | **0.332** | **0.362** | 0.335 | 0.368 |
| | 192 | **0.379** | **0.390** | 0.392 | 0.404 |
| | 336 | **0.418** | 0.423 | 0.421 | **0.421** |
| | 720 | **0.466** | **0.454** | 0.470 | 0.456 |
| m2 | 96 | **0.178** | **0.257** | 0.179 | 0.259 |
| | 192 | **0.253** | **0.307** | 0.258 | 0.313 |
| | 336 | 0.336 | 0.361 | **0.333** | **0.359** |
| | 720 | 0.475 | 0.426 | **0.423** | **0.467** |
| h1 | 96 | **0.391** | **0.409** | 0.407 | 0.419 |
| | 192 | 0.493 | **0.441** | **0.481** | 0.446 |
| | 336 | 0.642 | 0.506 | **0.541** | **0.468** |
| | 720 | **0.679** | **0.523** | 0.727 | 0.572 |
| h2 | 96 | 0.362 | 0.368 | **0.326** | **0.353** |
| | 192 | 0.438 | **0.410** | **0.436** | 0.411 |
| | 336 | **0.543** | **0.457** | 0.922 | 0.676 |
| | 720 | 1.007 | 0.614 | **0.824** | **0.577** |
| AvgWins | | **66.1%** | | 37.5% | |

Table 22: **Mean & Stds. for the PatchTOTEM Ablation.** Left is the specialist, right is the generalist.

**Generalist In Domain & Zero Shot Forecasting**

| Metric | | MSE (Mean ± Std) | MAE (Mean ± Std) |
|---|---|---|---|
| W | 96 | 0.173 ± 0.0012 | 0.218 ± 0.0006 |
|   | 192 | 0.218 ± 0.0006 | 0.261 ± 0.0006 |
|   | 336 | 0.267 ± 0.0006 | 0.299 ± 0.0006 |
|   | 720 | 0.337 ± 0.0010 | 0.347 ± 0.0006 |
| E | 96 | 0.183 ± 0.0012 | 0.267 ± 0.0012 |
|   | 192 | 0.189 ± 0.0006 | 0.275 ± 0.0000 |
|   | 336 | 0.204 ± 0.0010 | 0.291 ± 0.0010 |
|   | 720 | 0.242 ± 0.0006 | 0.325 ± 0.0006 |
| T | 96 | 0.517 ± 0.0000 | 0.293 ± 0.0029 |
|   | 192 | 0.526 ± 0.0030 | 0.296 ± 0.0006 |
|   | 336 | 0.552 ± 0.0015 | 0.304 ± 0.0015 |
|   | 720 | 0.602 ± 0.0046 | 0.326 ± 0.0015 |
| m1 | 96 | 0.428 ± 0.0090 | 0.420 ± 0.0040 |
|   | 192 | 0.438 ± 0.0015 | 0.427 ± 0.0010 |
|   | 336 | 0.469 ± 0.0062 | 0.447 ± 0.0042 |
|   | 720 | 0.546 ± 0.0081 | 0.493 ± 0.0017 |
| m2 | 96 | 0.207 ± 0.0015 | 0.286 ± 0.0020 |
|   | 192 | 0.269 ± 0.0015 | 0.325 ± 0.0010 |
|   | 336 | 0.358 ± 0.0199 | 0.377 ± 0.0091 |
|   | 720 | 0.521 ± 0.0165 | 0.482 ± 0.0026 |
| h1 | 96 | 0.401 ± 0.0006 | 0.410 ± 0.0006 |
|   | 192 | 0.453 ± 0.0010 | 0.441 ± 0.0010 |
|   | 336 | 0.496 ± 0.0017 | 0.468 ± 0.0006 |
|   | 720 | 0.518 ± 0.0020 | 0.510 ± 0.0017 |
| h2 | 96 | 0.305 ± 0.0006 | 0.346 ± 0.0006 |
|   | 192 | 0.396 ± 0.0015 | 0.402 ± 0.0001 |
|   | 336 | 0.492 ± 0.0310 | 0.458 ± 0.0131 |
|   | 720 | 0.599 ± 0.0105 | 0.531 ± 0.0026 |
| N2 | 96 | 1.127 ± 0.0017 | 0.773 ± 0.0006 |
|   | 192 | 1.169 ± 0.0032 | 0.793 ± 0.0010 |
|   | 336 | 1.115 ± 0.0010 | 0.780 ± 0.0006 |
|   | 720 | 1.070 ± 0.0035 | 0.766 ± 0.0010 |
| N5 | 96 | 0.481 ± 0.0015 | 0.483 ± 0.0006 |
|   | 192 | 0.508 ± 0.0012 | 0.500 ± 0.0000 |
|   | 336 | 0.481 ± 0.0006 | 0.491 ± 0.0006 |
|   | 720 | 0.467 ± 0.0010 | 0.488 ± 0.0010 |
| R | 96 | 1.102 ± 0.0031 | 0.578 ± 0.0021 |
|   | 192 | 1.207 ± 0.0036 | 0.628 ± 0.0017 |
|   | 336 | 1.190 ± 0.0021 | 0.613 ± 0.0010 |
|   | 720 | 1.149 ± 0.0017 | 0.596 ± 0.0020 |
| B | 96 | 0.825 ± 0.0079 | 0.751 ± 0.0076 |
|   | 192 | 0.847 ± 0.0021 | 0.761 ± 0.0012 |
|   | 336 | 0.831 ± 0.0066 | 0.764 ± 0.0042 |
|   | 720 | 0.928 ± 0.0131 | 0.813 ± 0.0050 |
| S | 96 | 0.446 ± 0.0015 | 0.481 ± 0.0010 |
|   | 192 | 0.478 ± 0.0015 | 0.499 ± 0.0000 |
|   | 336 | 0.535 ± 0.0012 | 0.532 ± 0.0006 |
|   | 720 | 0.736 ± 0.0025 | 0.631 ± 0.0006 |

**Specialist Forecasting**

| | | MSE (Mean ± Std) | MAE (Mean ± Std) |
|---|---|---|---|
| W | 96 | 0.164 ± 0.0006 | 0.209 ± 0.0006 |
|   | 192 | 0.209 ± 0.0017 | 0.251 ± 0.0023 |
|   | 336 | 0.261 ± 0.0012 | 0.293 ± 0.0017 |
|   | 720 | 0.332 ± 0.0023 | 0.340 ± 0.0006 |
| E | 96 | 0.179 ± 0.0015 | 0.262 ± 0.0015 |
|   | 192 | 0.185 ± 0.0006 | 0.269 ± 0.0000 |
|   | 336 | 0.204 ± 0.0055 | 0.289 ± 0.0061 |
|   | 720 | 0.244 ± 0.0040 | 0.325 ± 0.0036 |
| T | 96 | 0.528 ± 0.0081 | 0.310 ± 0.0092 |
|   | 192 | 0.500 ± 0.0606 | 0.349 ± 0.0699 |
|   | 336 | 0.531 ± 0.0424 | 0.365 ± 0.0852 |
|   | 720 | 0.578 ± 0.0361 | 0.398 ± 0.1103 |
| m1 | 96 | 0.326 ± 0.0006 | 0.355 ± 0.0006 |
|   | 192 | 0.377 ± 0.0023 | 0.386 ± 0.0012 |
|   | 336 | 0.409 ± 0.0006 | 0.409 ± 0.0006 |
|   | 720 | 0.469 ± 0.0015 | 0.441 ± 0.0000 |
| m2 | 96 | 0.176 ± 0.0010 | 0.254 ± 0.0006 |
|   | 192 | 0.247 ± 0.0031 | 0.303 ± 0.0026 |
|   | 336 | 0.318 ± 0.0006 | 0.350 ± 0.0021 |
|   | 720 | 0.419 ± 0.0067 | 0.411 ± 0.0044 |
| h1 | 96 | 0.377 ± 0.0010 | 0.395 ± 0.0006 |
|   | 192 | 0.428 ± 0.0015 | 0.428 ± 0.0015 |
|   | 336 | 0.480 ± 0.0021 | 0.462 ± 0.0012 |
|   | 720 | 0.530 ± 0.0110 | 0.522 ± 0.0108 |
| h2 | 96 | 0.294 ± 0.0021 | 0.338 ± 0.0010 |
|   | 192 | 0.373 ± 0.0023 | 0.389 ± 0.0032 |
|   | 336 | 0.423 ± 0.0031 | 0.433 ± 0.0025 |
|   | 720 | 0.591 ± 0.0145 | 0.556 ± 0.0051 |

Table 23: **Mean and Stds. for the Codebook Ablation** ($\downarrow$)

| | $K$ | MSE (Mean ± Std) | MAE (Mean ± Std) |
|---|---|---|---|
| All | 32 | 0.0451 ± 0.0014 | 0.1460 ± 0.0030 |
|   | 256 | 0.0192 ± 0.0003 | 0.0937 ± 0.0007 |
|   | 512 | 0.0184 ± 0.0025 | 0.0913 ± 0.0062 |
| W | 32 | 0.0393 ± 0.0005 | 0.1122 ± 0.0064 |
|   | 256 | 0.0161 ± 0.0004 | 0.0673 ± 0.0011 |
|   | 512 | 0.0128 ± 0.0011 | 0.0607 ± 0.0032 |
| E | 32 | 0.0463 ± 0.0007 | 0.1520 ± 0.0016 |
|   | 256 | 0.0209 ± 0.0012 | 0.1027 ± 0.0029 |
|   | 512 | 0.0152 ± 0.0005 | 0.0878 ± 0.0014 |
| T | 32 | 0.0312 ± 0.0007 | 0.1204 ± 0.0008 |
|   | 256 | 0.0120 ± 0.0003 | 0.0749 ± 0.0007 |
|   | 512 | 0.0101 ± 0.0012 | 0.0685 ± 0.0044 |

### A.6 Statistical Significance.

Despite the fact that much prior and concurrent work only reports results on 1 seed Wu et al. (2022); Goswami et al. (2024) and Zhou et al. (2023) (except for Table 15), we perform a statistical analysis on our generalist results.

Our results are statistically significant with 3 seeds using an exact one-sided permutation test. The exact one-sided permutation test repeatedly randomly permutes the reported metrics (e.g., MSE) between two competing methods (e.g., TOTEM vs. GPT2) and generates a distribution of all the possible metric assignments. This is an appropriate test because it returns a p-value indicating how rare it is to observe our reported results relative to all the permuted outcomes. Importantly, this test is non-parametric, so it is valid in a low-sample regime. The reason we expect this statistic to return $p <= 0.05$, even with 3 seeds, is because the training algorithms in this setting are actually quite stable (unlike other areas of ML such as deep RL (Henderson et al., 2018)).

We perform this analysis on the generalist models for each experimental trial/metric (e.g., for the MSE metric of the Weather dataset in the forecasting task), for all tasks (where we compare TOTEM vs. GPT2), and for the token vs. patch analysis (where we compare TOTEM vs. PatchTOTEM).

The following tables report the proportion of trials where the p-value is $<= 0.05$, i.e., where TOTEM statistically significantly outperforms the competing method (GPT2 24 or PatchTOTEM 25). While the results in the main paper double-count the ties between method (e.g., if TOTEM and GPT2 tied, a win was counted for both) to prove the strength of TOTEM, the tables below compare the percentage of experiments in which TOTEM strictly outperforms the baseline (we also provide the reported win percentage from the main paper for ease of comparison). It is clear that even in the statistical setting, TOTEM still wins more than baselines.

Table 24: **TOTEM vs. GPT2 Generalist Statistical Significance.** Here we compare the TOTEM and GPT2 generalists and calculate the proportion of trials where the p-value is $<= 0.05$, i.e., where TOTEM statistically significantly outperforms GPT2 (right column). These results are in line with those reported in the main paper.

| | AvgWins | | |
|---|---|---|---|
| | Original with ties (as in 5, 5, 6) | Original no ties | Statistical no ties |
| Imputation In Domain | 58.3% | 56.3% | 56.3% |
| Imputation Zero Shot | 80.0% | 80.0% | 80.0% |
| Anomaly Detection In Domain | 80.0% | 80.0% | 66.6% |
| Anomaly Detection Zero Shot | 73.3% | 73.3% | 73.3% |
| Forecasting In Domain | 67.9% | 66.1% | 62.5% |
| Forecasting Zero Shot | 90.0% | 87.5% | 82.5% |

Table 25: **Tokens vs. Patches Generalist Statistical Significance.** Here we compare the TOTEM and PatchTOTEM generalists and calculate the proportion of trials where the p-value is $<= 0.05$, i.e., where TOTEM statistically significantly outperforms PatchTOTEM (right column). These results are in line with those reported in the main paper.

| | AvgWins | | |
|---|---|---|---|
| | Original with ties (as in 7) | Original no ties | Statistical no ties |
| In Domain | 78.6% | 76.9% | 66.1% |
| Zero Shot | 67.5% | 65.0% | 60.0% |

### A.7 Further Exploration Details.

**Generalist Codebooks.** To further explore the capabilities of a generalist codebook data representation we train models that utilize a general codebook but dataset-specific transformer forecasters, i.e., a TOTEM VQVAE trained on multiple domains with a forecaster trained only on electricity, Table 26. We compare these mixed models to generalist and specialist models trained on the same domains. All models use the same

codebook hyperparameters (number of codewords $K = 256$, compression factor $F = 4$, code dimensionality $D = 64$) as well as the forecaster transformer architecture to ensure a fair comparison.

Since we are evaluating specialists, mixed-models, and a generalist on in-domain test data, one might expect the TOTEM specialists to significantly outperform all models in all domains. Surprisingly, this intuition is not correct. We find that the fully-generalist model (right Table 26) significantly outperforms the mixed-models (middle Table 26) in traffic (T) and electricity (E). This performance is puzzling until considering the training sizes.

The largest training set across domains belongs to traffic (T) at $10.2M$ training examples. In dataset T, the fully generalist models achieves 100% `AvgWins`. The second-largest training set belongs to electricity (E) at $5.8M$ training examples, with 75% `AvgWins` for the fully-generalist model. Unfortunately, there is a sharp drop off in training set sizes, with the rest of the data domains collectively comprising $1.6M$ training examples. These results evoke questions. For instance: does training on the smaller datasets act like a form of regularization? How does in-domain generalist performance scale with dataset size? We leave these exciting directions for future work. The generalist codebook's performance across datasets highlights the potential of unified, discrete, token representations for in-domain evaluations.

Table 26: Specialist models, mixed models, and generalist models.

| Codebook Forecaster Metric | | Specialist Specialist MSE | MAE | Generalist Specialist MSE | MAE | Generalist Generalist MSE | MAE |
|---|---|---|---|---|---|---|---|
| W | 96 | 0.165 | **0.208** | **0.164** | **0.208** | 0.172 | 0.216 |
| | 192 | **0.207** | **0.250** | 0.208 | 0.251 | 0.217 | 0.256 |
| | 336 | **0.257** | 0.291 | 0.258 | **0.290** | 0.266 | 0.295 |
| | 720 | **0.326** | 0.340 | 0.329 | **0.338** | 0.334 | 0.342 |
| E | 96 | **0.178** | **0.263** | **0.178** | **0.263** | 0.179 | 0.264 |
| | 192 | 0.187 | 0.272 | 0.187 | 0.273 | **0.181** | **0.267** |
| | 336 | 0.199 | 0.285 | 0.199 | 0.285 | **0.196** | **0.283** |
| | 720 | 0.236 | 0.318 | 0.238 | 0.320 | **0.230** | **0.314** |
| T | 96 | 0.523 | 0.303 | 0.521 | 0.301 | **0.507** | **0.284** |
| | 192 | 0.530 | 0.303 | 0.530 | 0.303 | **0.511** | **0.282** |
| | 336 | 0.549 | 0.311 | 0.555 | 0.313 | **0.535** | **0.292** |
| | 720 | 0.598 | 0.331 | 0.605 | 0.337 | **0.580** | **0.309** |
| m1 | 96 | **0.320** | **0.347** | 0.328 | 0.352 | 0.374 | 0.384 |
| | 192 | 0.379 | **0.382** | **0.377** | 0.383 | 0.400 | 0.399 |
| | 336 | **0.406** | **0.402** | 0.408 | 0.404 | 0.432 | 0.424 |
| | 720 | 0.471 | **0.438** | **0.470** | 0.440 | 0.487 | 0.460 |
| m2 | 96 | 0.176 | **0.253** | **0.175** | **0.253** | 0.198 | 0.275 |
| | 192 | 0.247 | **0.302** | 0.247 | **0.302** | **0.266** | 0.319 |
| | 336 | **0.317** | **0.348** | 0.318 | **0.348** | 0.365 | 0.377 |
| | 720 | **0.426** | **0.410** | 0.427 | **0.410** | 0.588 | 0.511 |
| h1 | 96 | **0.380** | **0.394** | 0.382 | 0.395 | 0.382 | 0.404 |
| | 192 | **0.434** | **0.427** | 0.437 | **0.427** | 0.463 | 0.435 |
| | 336 | **0.490** | **0.459** | **0.490** | 0.460 | 0.507 | 0.463 |
| | 720 | 0.539 | 0.513 | 0.536 | 0.512 | **0.517** | **0.500** |
| h2 | 96 | **0.293** | **0.338** | 0.294 | 0.339 | 0.307 | 0.345 |
| | 192 | **0.375** | **0.390** | 0.375 | 0.391 | 0.406 | 0.403 |
| | 336 | 0.422 | **0.431** | **0.421** | **0.431** | 0.505 | 0.460 |
| | 720 | **0.610** | **0.567** | **0.610** | **0.567** | 0.661 | 0.557 |
| `AvgWins` | | 57.1% | | 35.7% | | 30.4% | |

Table 27: Zero Shot Vignette: Training Size & Diversity

| Model | TOTEM Generalist | | TOTEM Specialist | | TOTEM Specialist | |
|---|---|---|---|---|---|---|
| Train Domain | ALL | | Traffic | | Electricity | |
| Sensor Num ($S$) | - | | 862 | | 321 | |
| Raw Length ($T$) | - | | 17544 | | 26304 | |
| Train Size | 17.6M | | 10.2M | | 5.8M | |
| Metric | MSE | MAE | MSE | MAE | MSE | MAE |
| **N2** 96 | **1.138** | **0.777** | 1.194 | 0.798 | 1.193 | 0.802 |
| 192 | **1.149** | **0.785** | 1.218 | 0.808 | 1.300 | 0.845 |
| 336 | **1.092** | **0.770** | 1.190 | 0.804 | 1.260 | 0.837 |
| 720 | **1.045** | **0.754** | 1.117 | 0.784 | 1.234 | 0.832 |
| **N5** 96 | **0.483** | **0.484** | 0.515 | 0.505 | 0.489 | 0.490 |
| 192 | **0.495** | **0.491** | 0.535 | 0.514 | 0.535 | 0.527 |
| 336 | **0.468** | **0.483** | 0.524 | 0.513 | 0.538 | 0.525 |
| 720 | **0.451** | **0.477** | 0.500 | 0.507 | 0.533 | 0.527 |
| **R** 96 | **1.120** | 0.582 | 1.171 | 0.635 | 1.141 | **0.579** |
| 192 | **1.242** | **0.635** | 1.273 | 0.673 | 1.297 | 0.652 |
| 336 | 1.237 | **0.626** | **1.232** | 0.653 | 1.247 | 0.628 |
| 720 | **1.182** | **0.604** | 1.198 | 0.642 | 1.236 | 0.633 |
| **B** 96 | **0.805** | **0.739** | 0.812 | 0.749 | 0.820 | 0.756 |
| 192 | **0.836** | **0.752** | 0.858 | 0.767 | 0.843 | 0.759 |
| 336 | 0.809 | 0.748 | 0.826 | 0.759 | **0.791** | **0.741** |
| 720 | 0.896 | 0.794 | 0.919 | 0.803 | **0.886** | **0.790** |
| **S** 96 | **0.446** | **0.482** | 0.476 | 0.508 | 0.460 | 0.487 |
| 192 | **0.462** | **0.491** | 0.511 | 0.528 | 0.505 | 0.511 |
| 336 | **0.521** | **0.525** | 0.576 | 0.568 | 0.569 | 0.545 |
| 720 | **0.717** | **0.625** | 0.795 | 0.685 | 0.764 | 0.641 |
| AvgWins | **85.0%** | | 2.5% | | 12.5% | |

Table 28: **Means and Stds. Mixed Models - Forecasting** ($\downarrow$)

| Metric | Mean ± Std | |
|---|---|---|
| | MSE | MAE |
| **W** 96 | 0.164 ± 0.0010 | 0.208 ± 0.0012 |
| 192 | 0.208 ± 0.0010 | 0.251 ± 0.0015 |
| 336 | 0.258 ± 0.0012 | 0.290 ± 0.0015 |
| 720 | 0.329 ± 0.0021 | 0.338 ± 0.0015 |
| **E** 96 | 0.178 ± 0.0006 | 0.263 ± 0.0010 |
| 192 | 0.187 ± 0.0021 | 0.273 ± 0.0017 |
| 336 | 0.199 ± 0.0012 | 0.285 ± 0.0017 |
| 720 | 0.238 ± 0.0012 | 0.320 ± 0.0012 |
| **T** 96 | 0.521 ± 0.0010 | 0.301 ± 0.0010 |
| 192 | 0.530 ± 0.0023 | 0.303 ± 0.0012 |
| 336 | 0.555 ± 0.0080 | 0.313 ± 0.0072 |
| 720 | 0.605 ± 0.0097 | 0.337 ± 0.0075 |
| **m1** 96 | 0.328 ± 0.0036 | 0.352 ± 0.0006 |
| 192 | 0.377 ± 0.0021 | 0.383 ± 0.0012 |
| 336 | 0.408 ± 0.0035 | 0.404 ± 0.0021 |
| 720 | 0.470 ± 0.0035 | 0.440 ± 0.0021 |
| **m2** 96 | 0.175 ± 0.0006 | 0.253 ± 0.0010 |
| 192 | 0.247 ± 0.0006 | 0.302 ± 0.0010 |
| 336 | 0.318 ± 0.0006 | 0.348 ± 0.0031 |
| 720 | 0.427 ± 0.0012 | 0.410 ± 0.0067 |
| **h1** 96 | 0.382 ± 0.0025 | 0.395 ± 0.0015 |
| 192 | 0.437 ± 0.0012 | 0.427 ± 0.0006 |
| 336 | 0.490 ± 0.0015 | 0.460 ± 0.0021 |
| 720 | 0.536 ± 0.0031 | 0.512 ± 0.0032 |
| **h2** 96 | 0.294 ± 0.0010 | 0.339 ± 0.0012 |
| 192 | 0.375 ± 0.0025 | 0.391 ± 0.0023 |
| 336 | 0.421 ± 0.0050 | 0.431 ± 0.0031 |
| 720 | 0.610 ± 0.0089 | 0.567 ± 0.0075 |

Table 29: **Mean and Stds. Traffic Only - Specialist Zero-Shot Performance (↓)**

| Metric | | MSE | MAE |
|---|---|---|---|
| | | Mean ± Std | |
| N2 | 96 | 1.194 ± 0.0062 | 0.798 ± 0.0020 |
| | 192 | 1.218 ± 0.0074 | 0.808 ± 0.0023 |
| | 336 | 1.190 ± 0.0153 | 0.804 ± 0.0052 |
| | 720 | 1.117 ± 0.0137 | 0.784 ± 0.0056 |
| N5 | 96 | 0.515 ± 0.0026 | 0.505 ± 0.0012 |
| | 192 | 0.535 ± 0.0051 | 0.514 ± 0.0028 |
| | 336 | 0.524 ± 0.0071 | 0.513 ± 0.0030 |
| | 720 | 0.500 ± 0.0064 | 0.507 ± 0.0032 |
| R | 96 | 1.171 ± 0.0023 | 0.635 ± 0.0019 |
| | 192 | 1.273 ± 0.0090 | 0.673 ± 0.0042 |
| | 336 | 1.232 ± 0.0055 | 0.653 ± 0.0022 |
| | 720 | 1.198 ± 0.0057 | 0.642 ± 0.0041 |
| B | 96 | 0.812 ± 0.0037 | 0.749 ± 0.0025 |
| | 192 | 0.858 ± 0.0025 | 0.767 ± 0.0015 |
| | 336 | 0.826 ± 0.0041 | 0.759 ± 0.0030 |
| | 720 | 0.919 ± 0.0063 | 0.803 ± 0.0037 |
| S | 96 | 0.476 ± 0.0012 | 0.508 ± 0.0012 |
| | 192 | 0.511 ± 0.0005 | 0.528 ± 0.0005 |
| | 336 | 0.576 ± 0.0024 | 0.568 ± 0.0009 |
| | 720 | 0.795 ± 0.0017 | 0.685 ± 0.0012 |

Table 30: **Means and stds. Electricity Only - Specialist Zero-Shot Performance (↓)**

| Metric | | MSE | MAE |
|---|---|---|---|
| | | Mean ± Std | |
| N2 | 96 | 1.193 ± 0.0059 | 0.802 ± 0.0020 |
| | 192 | 1.300 ± 0.0016 | 0.845 ± 0.0003 |
| | 336 | 1.260 ± 0.0162 | 0.837 ± 0.0055 |
| | 720 | 1.234 ± 0.0054 | 0.832 ± 0.0016 |
| N5 | 96 | 0.489 ± 0.0024 | 0.490 ± 0.0011 |
| | 192 | 0.555 ± 0.0012 | 0.527 ± 0.0007 |
| | 336 | 0.538 ± 0.0064 | 0.525 ± 0.0033 |
| | 720 | 0.533 ± 0.0010 | 0.527 ± 0.0006 |
| R | 96 | 1.141 ± 0.0056 | 0.579 ± 0.0028 |
| | 192 | 1.297 ± 0.0162 | 0.652 ± 0.0079 |
| | 336 | 1.247 ± 0.0108 | 0.628 ± 0.0059 |
| | 720 | 1.236 ± 0.0053 | 0.633 ± 0.0070 |
| B | 96 | 0.820 ± 0.0065 | 0.756 ± 0.0034 |
| | 192 | 0.843 ± 0.0042 | 0.759 ± 0.0022 |
| | 336 | 0.791 ± 0.0023 | 0.741 ± 0.0019 |
| | 720 | 0.886 ± 0.0059 | 0.790 ± 0.0020 |
| S | 96 | 0.460 ± 0.0017 | 0.487 ± 0.0010 |
| | 192 | 0.505 ± 0.0017 | 0.511 ± 0.0008 |
| | 336 | 0.569 ± 0.0020 | 0.545 ± 0.0011 |
| | 720 | 0.764 ± 0.0046 | 0.641 ± 0.0014 |

### A.8 Generalist Training Time Comparisons.

|  | TOTEM (Ours) Params | TOTEM (Ours) Training Time on 1 A100 | GPT2 Params | GPT2 Training Time on 1 A100 |
|---|---|---|---|---|
| Imputation | ~345,000 | ~1.5 hours | ~60,700,000 | Several days |
| Anomaly Detection | ~345,000 | ~1.5 hours | ~46,500,000 | Several days |
| Forecasting | ~345,000 for VQVAE, ~1,600,000 for downstream transformer | ~1.5 hours for VQVAE, ~a day for transformer | ~89,000,000 | Several days |

Table 31: Comparison of Parameters and Training Time between TOTEM (Ours) and GPT2 generalist models.

### A.9 Codebook Visualizations.

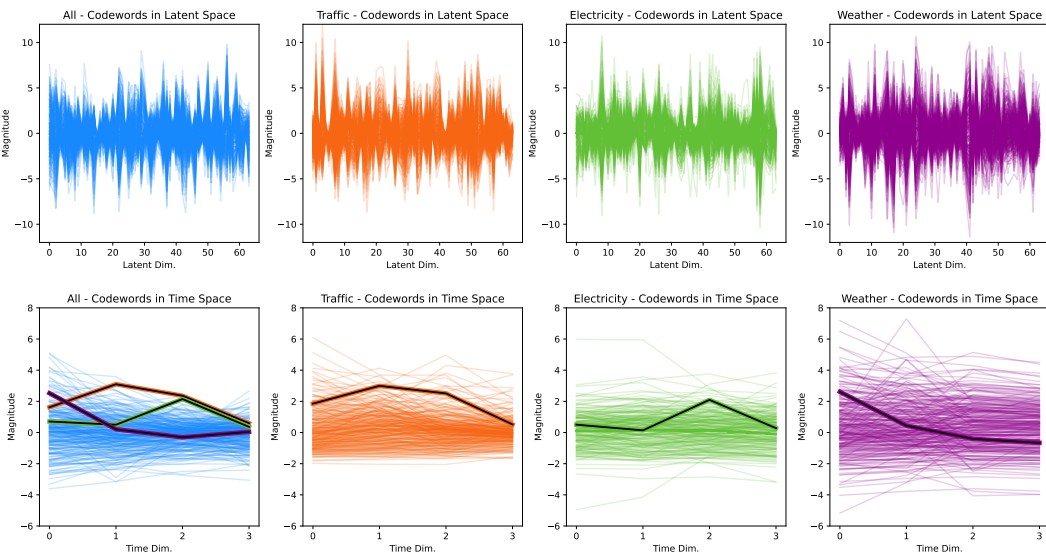

Figure 13: **TOTEM Codebooks.** We visualize all 256 codes for the generalist (All), and three specialists (Traffic, Electricity, and Weather). The top row visualizes codes in the latent space, the bottom row visualizes codes in the decoded time space. We additionally highlight codeword pairs matched via low MSE between All-Traffic, All-Electricity, and All-Weather in the bottom row.

## A.10 TOTEM Examples.

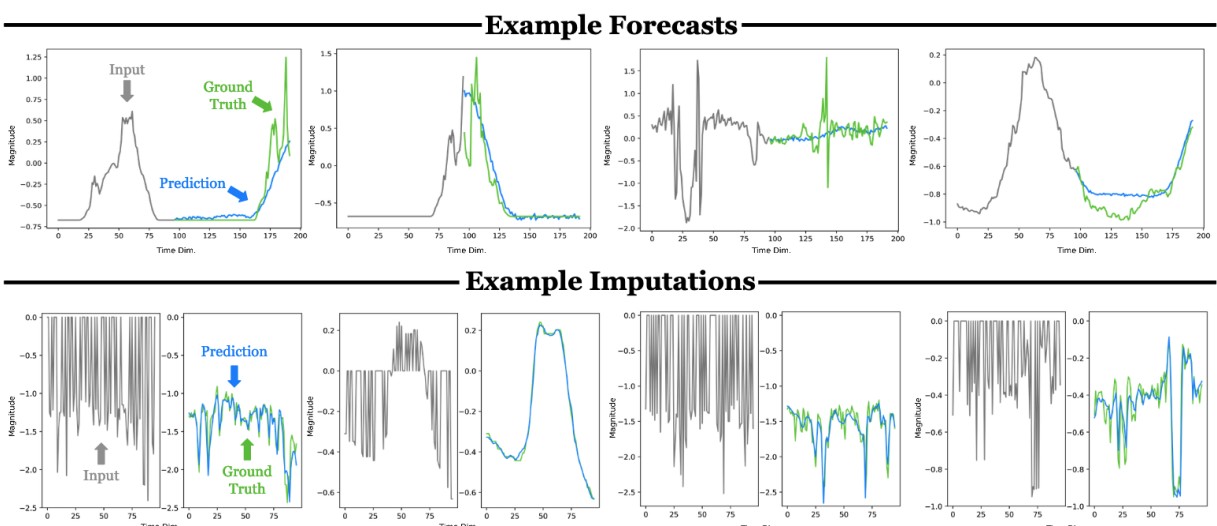

Figure 14: **TOTEM Examples.** In the top row we visualize four weather forecasts for Tin=96 and Tout=96. In the bottom row we visualize four ETTm2 imputations. In all cases the model input is in grey, the predictions are in blue, and the ground truth is in green.

## A.11 Architecture Details.

**VQVAE.** For imputation, anomaly detection, and forecasting the VQVAE's number of residual layers = 2, residual hidden size = 64, and block hidden size = 128 for all datasets. Each residual block has 2 non-causal, non-dilated 1D convolutional layers. The residual blocks are paired with additional non-causal, non-dilated 1D convolutional layers, where the number of additional layers is determined by the desired compression factor. See Table 32 for more hyperparameter details.

Table 32: **VQVAE Hyperparameters** (A) Imputation generalist (All) and specialists. (B) Anomaly detection generalist (All) and specialists. The anomaly %s for all of the zero shot datasets are 2%. (C) Forecasting generalist (All) and specialists.

A. **Imputation**.

| Dataset | LR | Iter. | BS | # CW | CW Dim. | CF |
|---|---|---|---|---|---|---|
| All | 1e-3 | 120000 | 8192 | 512 | 64 | 4 |
| Elec. | 1e-3 | 15000 | 8192 | 512 | 64 | 4 |
| Weather | 1e-3 | 15000 | 8192 | 512 | 64 | 4 |
| ETTm1 | 1e-3 | 15000 | 8192 | 512 | 64 | 4 |
| ETTm2 | 1e-3 | 15000 | 8192 | 512 | 64 | 4 |
| ETTh1 | 1e-3 | 15000 | 8192 | 512 | 64 | 4 |
| ETTh2 | 1e-3 | 15000 | 8192 | 512 | 64 | 4 |

B. **Anomaly Detection**.

| Dataset | LR | Iter. | BS | # CW | CW Dim. | CF | Anomaly % |
|---|---|---|---|---|---|---|---|
| All | 1e-3 | 120000 | 4096 | 1024 | 64 | 4 | Varies by test set. |
| SMD | 1e-3 | 60000 | 4096 | 1024 | 64 | 4 | 0.5 |
| MSL | 1e-3 | 15000 | 4096 | 1024 | 64 | 4 | 2 |
| PSM | 1e-3 | 60000 | 4096 | 1024 | 64 | 4 | 1 |
| SMAP | 1e-3 | 15000 | 4096 | 1024 | 64 | 4 | 1 |
| SWAT | 1e-3 | 15000 | 4096 | 1024 | 64 | 4 | 1 |

C. **Forecasting**.

| Dataset | LR | Iter. | BS | # CW | CW Dim. | CF |
|---|---|---|---|---|---|---|
| All | 1e-3 | 15000 | 4096 | 256 | 64 | 4 |
| Elec. | 1e-3 | 15000 | 4096 | 256 | 64 | 4 |
| Weather | 1e-3 | 15000 | 4096 | 256 | 64 | 4 |
| Traffic | 1e-3 | 15000 | 4096 | 256 | 64 | 4 |
| ETTm1 | 1e-3 | 15000 | 4096 | 256 | 64 | 4 |
| ETTm2 | 1e-3 | 15000 | 4096 | 256 | 64 | 4 |
| ETTh1 | 1e-3 | 15000 | 4096 | 256 | 64 | 4 |
| ETTh2 | 1e-3 | 15000 | 4096 | 256 | 64 | 4 |

**Downstream Forecaster.** The downstream forecaster has two components the transformer encoder that intakes codes and outputs a normalized time forecast, and the feedforward neural network that takes in time and outputs predictions for the forecast's mean and standard deviation. The downstream forecaster is a transformer encoder with a model dimension = 64, hidden dimension = 256, number of heads = 4, number of layers = 4. The transformer encoder applies a sin / cos positional embedding along the time dimension and

applies its attention mechanism to each sensor independently. There is a single linear layer applied after the transformer encoder output. The feedforward neural network takes in the input time steps, and predicts the future's mean and standard deviation.

### A.12   Training Details.

In imputation, anomaly detection, and forecasting the VQVAE is trained with a learning rate of 0.001 using the Adam optimizer, embedding dimension of 64, commitment cost of 0.25, and compression factor of 4; see Table 32 for more hyperparameters. The codewords are uniformly randomly initialized over $[\frac{-1}{K}, \frac{1}{K}]$, where K is the number of codewords and D is the latent dimension. In all tasks there is a global normalization, and local normalization Kim et al. (2021); both are standard throughout prior work. In imputation we only leverage global normalization, in anomaly detection and forecasting we utilize both global and local normalization. In anomaly detection we evaluate the models we run, TOTEM and GPT2, with both local normalized data and non-local normalized data for each method and report whichever schema leads to the best performance. In forecasting the downstream model is a transformer encoder with 4 layers and 4 attention heads and a feed-forward hidden dimension of 256. We train using Adam with a base learning rate of 0.0001 and a one cycle learning rate scheduler in accordance with Nie et al. (2022) on A100s.

