# OpenReview forum: "TOTEM: TOkenized Time Series EMbeddings for General Time Series Analysis"
_TMLR — Accepted by TMLR_

### Review · Reviewer_nhYG · 2024-09-18

**Summary Of Contributions:**

This paper presents a generalist method for time series analysis called Tokenized Time Series Embeddings (TOTEM). The approach makes use of the VQ-VAE at its core, which, in addition to other design aspects, makes TOTEM both a _generalist_ model in the sense that it can be trained on many different domains, and a specialist model in the sense that it can also be trained on a single domain. The bulk of the paper is then spent evaluating TOTEM on a large variety of tasks and settings, including (i) imputation, (ii) anomaly detection, and (iii) forecasting. Beyond these three problems, however, the paper also studies other aspects of the design of TOTEM, such as whether tokens or patches are more useful (Figure 7), or how the side of the codebook impacts performance (Figure 9). Collectively, these ablations and the three analyses on the core problems mentioned above comprise the bulk of the experimental work. The stated findings from the experiments is that TOTEM regularly matches (or outperforms) state-of-the-art.

**Audience:**

Yes

**Broader Impact Concerns:**

None of note.

**Claims And Evidence:**

Yes

**Requested Changes:**

__Writing Suggestions__

Overall:
- My primary suggestion is to focus on improving the strength of the evidence in your experimental data. This does not require any further baselines or settings, as the paper is already extremely broad in that sense. Rather, I believe that some statistical analysis of the data would add a great deal of support to your main claims, so long as the claims made also match the strength of the findings. At the moment, it is very difficult to read the main tables, or to interpret the data, or to identify the strength of the evidence. In its current form the paper's claims do not match the strength of the evidence provided.
- In general, if you are attributing a fact or claim from another piece of work without stating the authors' name(s) directly, use the parenthetical citation: for example, "on a single time series domain Zhou et al. (2023); Wu et al. (2022a); Nie et al. (2022), Figure 1A" should be "on a single time series domain (Zhou et al. 2023; Wu et al. 2022a; Nie et al. 2022), as visualized in Figure 1A.

Other low-level writing suggestions that had less impact on my evaluation:
- I don't quite understand the use of color in the annotation of Figure 1 (why is "in" purple? It appears to be referencing some aspect of the purple color in the visual itself, but I can't decipher it.)
- I believe aspects of the related work discussion could be made more specific, and more useful. The initial blocks of citations (as in " ranging from statistical methods Winters (1960); Holt (1957); Anderson (1976); Hyndman & Athanasopoulos (2018); Taylor & Letham (2018) to multilayer perceptrons (MLPs) Zeng et al. (2023); Li et al. (2023); Das et al. (2023a); Challu et al. (2023); Chen et al. (2023); Zhang et al. (2022); Oreshkin et al. (2019)...") is not all that helpful to the reader. Providing slightly more insight or discussion on these approaches would be valuable, rather than summarizing a vast body of literature with statements like "Many models are hybrid solutions that blend aforementioned approaches." That said, the discussion that follows this paragraph is quite valuable.
- I would move Section 3.2 before 3.1. It is easier to digest the design details when they are contextualized by the problem framing. I believe this would also benefit from moving the "Operating Across the Time Dimension. " paragraph to Section 3.2.
- "any dimension E, S, or T, Figure 2 left." --> "any dimension E, S, or T, visualized in Figure 2."
- I had a hard time parsing this: "this two-step represent then solve-task process "

**Strengths And Weaknesses:**

__Strengths__
- The paper's experimental breadth is staggering, covering a wide variety of baselines, problems, settings, and even a few ablations.
- The basic premise of the work is compelling: the design of a generic time series model that requires minimal fine-tuning is a great premise.
- The visuals in Figure 1 and 4 helps to elucidate aspects of the design of the method.

__Weaknesses__
- While the breadth of experimentation is clearly a strength, the care of reporting results, depth, and experimental rigor are lacking. The paper reports "For all experiments & models in the main paper, we run three seeds and report the mean; standard deviations in the Appendix." Consequently, the experimental data present weak evidence, despite the breadth of baselines and domains. As a result, I find it hard to draw strong conclusions. I encourage the authors to perform a more careful statistical comparison, or at the very least increase the number of seeds used on a smaller number of domains in order to arrive at some statistically meaningful conclusions. As an example, the question of whether tokens or patches are more useful is an interesting one, but the evidence presented is not strong enough to support the claim "TOTEM’s performance demonstrates that tokens, when compared to patches, lead to better performance" simply because there is no indication of how meaningful these data are. Regarding care in reporting the results, the tables are extremely difficult to read. While this is subjective, I would encourage using more white space, and presenting these data in a different way. The imputation performance summaries are relatively easy to process, but B. C. and D. of Table 2 are quite difficult to parse.


Update: Following the response from the authors, I believe the proposed statistical tests and corresponding strength of the evidence in the paper to resolve my concern above, and I am updating my response to the question "Claims and Evidence" in light of that.

---

> ### Author Response · Authors · 2024-10-03
> **Response to Reviewer nhYG 1/2**
>
> We thank the reviewer for their thoughtful comments. We respond in order.
>
> &nbsp;
>
> ### Statistical Significance & Strength of Evidence
>
> We appreciate the suggestion to clarify the statistical significance of our results and the strength of our evidence. We have two main responses.
>
> 1. **Our results are statistically significant with 3 seeds using an exact one-sided permutation test.** The exact one-sided permutation test repeatedly randomly permutes the reported metrics (e.g., MSE) between two competing methods (e.g., TOTEM vs. GPT2) and generates a distribution of all the possible metric assignments. This is an appropriate test because it returns a p-value indicating how rare it is to observe our reported results relative to all the permuted outcomes. Importantly, this test is non-parametric, so it is valid in a low-sample regime. The reason we expect this statistic to return p\<=0.05, even with 3 seeds, is because the training algorithms in this setting are actually quite stable (unlike other areas of ML such as deep RL \[4\]).
>
>    We perform this analysis on the generalist models for each experimental trial/metric (e.g., for the MSE metric of the Weather dataset in the forecasting task), for all tasks (where we compare TOTEM vs. GPT2), and for the token vs. patch analysis (where we compare TOTEM vs. PatchTOTEM). For clarity, we provide the code used to perform the analysis on a small example in the next comment.
>
>    The following tables report the proportion of trials where the p-value is \<= 0.05, i.e., where TOTEM statistically significantly outperforms the competing method (GPT2 or PatchTOTEM). While the original results double-counted the ties between method (e.g., if TOTEM and GPT2 tied, a win was counted for both) to prove the strength of TOTEM, the tables below compare the percentage of experiments in which TOTEM **strictly** outperforms the baseline (we also provide the originally-reported win percentage from the original paper for ease of comparison). It is clear that even in the statistical setting, TOTEM still wins more than baselines, and further, the results are consistent with our original claims, so the interpretation of our results remains the same.
>
> Task Statistical Significance \- Compares Generalists \- TOTEM vs. GPT2
>
> |  | NewPercent TOTEM wins (not including ties) | Original Percent TOTEM wins (not including ties) | OriginalPercent TOTEM wins or ties (as in Tab.2, Fig.5, Fig.6) |
> | :---- | :---- | :---- | :---- |
> | Imputation In Domain | 56.3% | 56.3% | 58.3%  |
> | Imputation Zero Shot | 80.0% | 80.0%  | 80.0%  |
> | Anomaly Detection In Domain | 66.6%  | 80.0% | 80.0% |
> | Anomaly Detection Zero Shot | 73.3% | 73.3% | 73.3% |
> | Forecasting In Domain |   62.5%  | 66.1% | 67.9% |
> | Forecasting Zero Shot |   82.5%  | 87.5% | 90.0% |
>
> Tokens vs. Patches Statistical Significance \- Compares Generalists \- TOTEM vs. PatchTOTEM
>
> |  | NewPercent TOTEM wins (not including ties) | Original Percent TOTEM wins (not including ties) | OriginalPercent TOTEM wins or ties (as in Tab.7) |
> | :---- | :---- | :---- | :---- |
> | In Domain | 66.1% | 76.9% | 78.6% |
> | Zero Shot | 60.0%  | 65.0% | 67.5% |
>
> 2. **Much prior work only reports results on 1 seed \[e.g., 1, 2, 3\].** To statistically analyze specialist models from prior work, we would need to retrain each baseline across all individual datasets with multiple seeds, followed by hundreds of additional trials. Given the breadth of our current results and the above analysis for the generalist case, we believe there is enough evidence to support our main thesis that tokens improve performance over patches without the need for a prohibitive number of further trials.
>
> We are happy to include the above analysis in the paper, and believe this additional analysis further supports our initial claims.
>
> &nbsp;
>
> ### Presentation and Visualization of Results
>
> Thank you for the suggestion regarding clarity of presentation \- we are working on alternatives for visualizing the data. Due to the volume of results, we believed the bar chart summary and coloring the tables would be helpful. We’re happy to increase the white space like you suggest, and maybe move the imputation tables to the Appendix (like they are for anomaly detection and forecasting). Were the bar charts a helpful visualization? We would be happy to incorporate any suggestions into the visualization if you have any specific thoughts.
>
> &nbsp;
>
> ### Other Comments
>
> **Parenthetical Citations.** We will fix these throughout the text.
>
> **Annotation Color in Fig. 1\.** The purple-colored “in” and aqua-colored “domain” in the caption match the purple “in” and aqua “domain” in part b of the Figure. The purple color denotes that the electricity dataset (house) is in-domain inference for the generalist (purple in Figure 1a) and the aqua color denotes that the electricity dataset is in-domain inference for the specialist (aqua in Figure 1a). If it is too confusing, we’re happy to amend the coloring.

---

> > ### Author Response · Authors · 2024-10-03
> > **Response to Reviewer nhYG 2/2**
> >
> > **Improving Related Work.** Thank you for the constructive feedback. We will sharpen the discussion in the first paragraph and incorporate the helpful elements in the remainder of the section to improve the quality of the writing.
> >
> > &nbsp;
> >
> > **Sections 3.1 and 3.2.** We’re happy to move section 3.2 ahead of 3.1. For the “Operating Across the Time Dimension” subsection, we do feel that it belongs in the “Design Decisions” section, but we could move it to the beginning of the section so that it immediately follows the “Task Definitions.” Would this address your concern?
> >
> > &nbsp;
> >
> > **E, S, or T Suggestion.** We will update the draft with your suggestion.
> >
> > &nbsp;
> >
> > **"this two-step represent then solve-task process".** Is the following more clear?
> > > In language modeling, tokenizations that are created independently from the downstream model (e.g., not learnt end-to-end) have been applied with wide success even with the rapid increase in training data quantity and diversity.
> >
> > &nbsp;
> >
> > \[1\] Wu, Haixu, et al. "Timesnet: Temporal 2d-variation modeling for general time series analysis." *arXiv preprint arXiv:2210.02186* (2022).
> >
> > \[2\] Zhou, Tian, et al. "One fits all: Power general time series analysis by pretrained lm." *Advances in neural information processing systems* 36 (2023): 43322-43355. (all results are on 1 seed, except for 1 supplemental experiment with 1 task in 1 testing-paradigm on the 2 smallest datasets using 3 seeds)
> >
> > \[3\] Goswami, Mononito, et al. "Moment: A family of open time-series foundation models." *arXiv preprint arXiv:2402.03885* (2024).
> >
> > \[4\] Henderson, Peter, et al. "Deep reinforcement learning that matters." *Proceedings of the AAAI conference on artificial intelligence*. Vol. 32\. No. 1\. 2018\.
> >
> > &nbsp;
> >
> > ### Statistical Significance & Strength of Evidence \- Code & Example
> >
> > ```
> > # See scipy.stats permutation test
> > # https://docs.scipy.org/doc/scipy/reference/generated/scipy.stats.permutation_test.html#permutation-test
> >
> > import numpy as np
> > import scipy.stats
> >
> > # STEP A - SETUP EXAMPLE DATA
> > # For this example data:
> > # (1) a lower metric means the method performs better (e.g. 0.030 is better than 0.032)
> > # (2) each list corresponds to 4 separate conditions (e.g. weather forecasting for various prediction lengths (PL): [96PL, 192PL, 336PL, 720PL])
> >
> > # Example Data Method 1
> > method1_seed1 = [0.028, 0.03, 0.08, 1]
> > method1_seed2 = [0.028, 0.03, 0.119, 1]
> > method1_seed3 = [0.03, 0.031, 0.081, 1]
> >
> > # Example Data Method 2
> > method2_seed1 = [0.029, 0.032, 0.076, 1]
> > method2_seed2 = [0.029, 0.033, 0.077, 1]
> > method2_seed3 = [0.029, 0.033, 0.078, 1]
> >
> > # STEP B - RESHAPE THE DATA
> > def stack_indiv_model(arr1, arr2, arr3, arr4, arr5, arr6):
> >     '''
> >     Stack all of the data to process it with the scipy stats permutation test.
> >     If 4 conditions (e.g. prediction lengths PL), 3 seeds, and 2 methods the resulting data shape will be (2 x 4 x 3)
> >     '''
> >     method1 = np.stack((arr1, arr2, arr3)).T # shape (4 x 3)
> >     method2 = np.stack((arr4, arr5, arr6)).T # shape (4 x 3)
> >     return np.stack((method1, method2)) # (2 x 4 x 3)
> >
> > # Combine all of the data --> (2 x 4 x 3)
> > data = stack_indiv_model(method1_seed1, method1_seed2, method1_seed3,
> >                          method2_seed1, method2_seed2, method2_seed3)
> >
> > # STEP C - DEFINE METHODS
> > def mean_dif_statistic(method1, method2, axis=0):
> >     return np.mean(method1, axis=-1) - np.mean(method2, axis=-1)
> >
> > def cal_results(arr):
> >     '''
> >     (1) use the mean_dif_statistic defined earlier.
> >     (2) use the 'independent' permutation type because this means the null hypothesis is "the data are randomly sampled from the same distribution".
> >     (3) since we have 6 observations in total with 3 seeds and 6 choose 3 is < n_resamples=9999, we will calculate an exact null distribution.
> >     (4) using 'alternative' less means that the p-values output the percentage of the null distribution that is less than or equal to the observed value of the test statistic. Therefore a low p-value indicates that statistically method1 is lower than method2. In this case because a lower metric is better that means that method1 outperforms method2.
> >     '''
> >     output_less = scipy.stats.permutation_test(arr, statistic=mean_dif_statistic, permutation_type='independent',
> >                     vectorized=True, n_resamples=9999, batch=None,
> >                     alternative='less', axis=-1, random_state=None)
> >
> >     pval_less = output_less.pvalue
> >     print("What were the p values?\n" + str(pval_less) + "\n")
> >
> >     method1_better = pval_less <= 0.05
> >     print("How many times does method 1 win?\n" + str(method1_better.sum()) + " time.")
> >
> > # STEP D - RUN THE PERMUTATION TEST
> > cal_results(data)
> >
> > # Output
> > # What were the p values?
> > # [0.5  0.05 1.   1.  ]
> >
> > # How many times does method 1 win?
> > # 1 time.
> > ```

---

> > > ### Author Response · Authors · 2024-10-14
> > > **Summary of Updates**
> > >
> > > Dear Reviewer nhYG,
> > >
> > > &nbsp;
> > >
> > > As the end of the review period approaches, we would greatly appreciate it if you could confirm whether our response has adequately addressed your concerns.
> > >
> > > &nbsp;
> > >
> > > Given our previous discussion, below is a summary of updates we will make to the paper. We will:
> > > 1. add discussion of the statistical significance and strength of evidence to the text, and add the table to the appendix
> > > 2. increase the white space of the imputation tables, and potentially move them to the appendix (e.g. same presentation as the anomaly detection and forecasting results - bar charts in the main paper, and tables in the appendix).
> > > 3. fix parenthetical citations throughout the text
> > > 4. remove the confusing text coloring in Fig. 1
> > > 5. sharpen the discussion in the first paragraph of related work and incorporate the helpful elements in the remainder of the section
> > > 6. move section 3.2 ahead of 3.1
> > > 7. update the “E, S, or T” wording
> > > 8. update the "this two-step represent then solve-task process" wording
> > >
> > > &nbsp;
> > >
> > > If any questions remain, please let us know, and we will do our best to respond within the remaining time. Otherwise, we kindly thank you for your valuable feedback as it has improved our work.

---

> > > > ### Comment · Reviewer_nhYG · 2024-10-14
> > > > **Response and discussion**
> > > >
> > > > Dear Authors,
> > > >
> > > > Thank you for your detailed, thoughtful response, and for the added code, proposed changes, and statistical tests. I believe that the inclusion of the one-sided permutation test significantly strengthens the evidence in the paper and resolves my primary concern. In light of this update, I will update my review. I will also discuss these proposals with the other reviewers and can follow up if any questions or concerns remain
> > > >
> > > > Thanks,
> > > >
> > > > Reviewer nhYG

---

> > > > > ### Author Response · Authors · 2024-10-15
> > > > > **Thanks for Improving Our Work**
> > > > >
> > > > > Thank you for your constructive feedback, clearly articulated concerns, and thoughtful responses. We feel that our discussions undoubtedly improved and clarified our work. Thanks again!

---

### Review · Reviewer_JMhz · 2024-09-20

**Summary Of Contributions:**

The paper "TOTEM: TOkenized Time Series EMbeddings for General Time Series Analysis" presents TOTEM, a framework specifically designed to tackle general time series analysis by transforming time series data into tokenized embeddings. The proposed method is rigorously evaluated across three different real-world scenarios.

I have done my best to provide a thorough and objective review of the paper. However, as I am not an expert in this specific field, my feedback may appear somewhat superficial or lack the depth that a more experienced reviewer could provide.

**Audience:**

Yes

**Claims And Evidence:**

Yes

**Requested Changes:**

1.The three tasks are only introduced in Section 3.2. To make the paper more accessible to non-expert readers, I suggest briefly introducing these tasks in the introduction.
2. Training time comparison
3. The tables are somewhat difficult to read; I recommend improving their visual presentation for better clarity.

**Strengths And Weaknesses:**

S1: Exploring Time Series Analysis Through Tokenization is Interesting: The application of tokenization, a technique commonly used in natural language processing, to time series analysis is a creative and innovative approach.
S2: Rigorous Evaluation of the Method: The proposed method is not only conceptually interesting but also rigorously tested across various real-world scenarios


W1: Risk of Losing Fine-Grained Temporal Information: While tokenization is a novel and powerful technique, it introduces the risk of losing important fine-grained temporal details. Time series data often contain subtle temporal dependencies and trends that may not be fully captured when discretized into tokens.
w2: Lack of Comparison of Training Time with Competitors: Although the model's performance is evaluated in terms of accuracy and robustness, the paper does not provide a detailed comparison of training time against competing methods. Since deep learning models, especially transformer-based ones, can be computationally expensive and time-consuming to train, it would be important to understand how TOTEM performs in terms of computational efficiency.

---

> ### Author Response · Authors · 2024-10-03
> **Location of Response to Reviewer JMhz**
>
> Somehow our response became a general comment; please find our response here:
> https://openreview.net/forum?id=QlTLkH6xRC&noteId=TiDfhCaRDc

---

> > ### Author Response · Authors · 2024-10-14
> > **Summary of Updates**
> >
> > Dear Reviewer JMhz,
> >
> > &nbsp;
> >
> > As the end of the review period approaches, we would greatly appreciate it if you could confirm whether our response has adequately addressed your concerns.
> >
> >
> > &nbsp;
> >
> > Given our previous discussion, below is a summary of updates we will make to the paper. We will:
> > 1. add the training time comparison table to the appendix
> > 2. add a sentence to paragraph 2 of the Introduction that expands upon the tasks
> > 3. increase the white space of the imputation tables, and potentially move them to the appendix (e.g. same presentation as the anomaly detection and forecasting results - bar charts in the main paper, and tables in the appendix).
> >
> > &nbsp;
> >
> > If any questions remain, please let us know, and we will do our best to respond within the remaining time. Otherwise, we kindly thank you for your valuable feedback as it has improved our work.

---

### Review · Reviewer_FqZp · 2024-10-01

**Summary Of Contributions:**

The paper is well-researched and clearly written supported by elaborate experiments. The paper is novel with a simple idea of using VQVAE based architecture to build a generalized time-series with baselines and datasets.

There are some concerns regarding the choices of architectures and decisions about why VQ-VAE was chosen but overall, it is a good paper and definitely worthy of TMLR.

**Audience:**

Yes

**Claims And Evidence:**

Yes

**Requested Changes:**

The paper is well-written for the most part and the following changes are required.

1. Explanation and motivation behind the choice of VQ-VAE and the reason for having to use Straight-through estimator.
2. More experiments in diverse domains and how these results compare and contrast. Example time series: ECG time series, communication channels, motion activity data

**Strengths And Weaknesses:**

Strengths:

1. The paper is explained and written very clearly and that is the main strength. It builds upon simple concepts and proposes a tokenised model as opposed to a patch-based model
2. Experiments and ablation studies are another big strength for the paper. They are extensive.

Weaknesses:
1. While the experiments are extensive, the domains can be more diverse. E.g: how does the TOTEM that is trained on an EEG dataset translate to ECG and vice versa? These are important questions to be answered.
2. What other architectures were considered?

---

> ### Author Response · Authors · 2024-10-03
> **Response to Reveiwer FqZp 1/2**
>
> We thank the reviewer for their helpful comments.
>
> &nbsp;
>
> ### Weakness 1: Domain Diversity
>
> **Our current results explore many diverse domains in both training and testing that exceeds the domain diversity of prior work**. We believe the diversity of domains is a strength, not a weakness, of our work. Below we would like to clarify the breadth of our diversity in the current zero-shot testing experiments (e.g., your train EEG, test ECG suggestion). Our current experiments prioritize (1) domain diversity, (2) sampling rate diversity, (3) training regime diversity, and (4) exceed what has been done in prior work.
>
> 1\. Regarding domain diversity, we explored many training and testing domains (see the table below). Moreover, we intentionally tested on domains not seen at training time. Here are three examples (of many) in the paper:
> 	(A) train on Traffic, test on Neuroscience \[ECoG\] (See Fig. 11, Tab. 21\)
> 	(B) train on Electricity, test on U.S. Birth Rate (See Fig. 11, Tab. 21\)
> 	(C) train on Servers \+ Mars \+ Soil Moisture \+ Water Treatment, test on River Flow (See Fig. 5, Tab. 7\)
>
> 2\. Regarding sampling rate diversity, we intentionally tested on sampling rates not seen at training time. See Table 3\.
>
> 3\. Regarding training regime diversity, we tested both generalist-trained models and the two largest specialist-trained models (traffic and electricity, see Figure 11\) in zero-shot testing experiments. In both training regimes, the training sets did not share domains with the testing sets.
>
> 4\. Regarding the domain diversity of prior work, our zero shot experiments exceed experiments in much prior work (e.g., \[1,2\] just train specialist forecasters in limited data regimes). See the full scope of our data domains for each task in this table:
>
> | Task | Training Domains Explored (Main Paper) | Training Domains Explored (Appendix) | Zero Shot Testing Domains Explored |
> | :---- | :---- | :---- | :---- |
> | Imputation | Weather, Electricity, Transformer Temperature | Healthcare | Neuroscience (ECoG), River Flow, U.S. Birth Rate, Sunspot  |
> | Anomaly Detection | Server Machines, Mars Science Lab, Soil Moisture, Water Treatment | Insect Feeding, Walking Acceleration, Internal Bleeding, Air Temperature, 3D Gait Phase, Accelerometer on Whale, NASA SpaceCraft Increase Rate, Heart Beat (including ECG datasets), Parkinson’s Asymmetry, Tilt Table Beat | Neuroscience (ECoG), River Flow, U.S. Birth Rate, Sunspot  |
> | Forecasting | Traffic, Weather, Electricity, Transformer Temperature | Tourism, Imports & Exports, Demographics, Real Estate, etc. \[aggregated in the W4, W3, etc. datasets\] | Neuroscience (ECoG), River Flow, U.S. Birth Rate, Sunspot  |
>
> We are happy to add the above table to the paper, if it would help clarify our domain diversity.
>
> &nbsp;
>
> ### Weakness 2: Architectures
>
> The main goal of this work is to study the effect of using tokenization for general time series modeling across many tasks. Given our goal, the study of model architectures is orthogonal to the main thesis of the paper; therefore, we use standard architectural choices from prior works to promote ease of comparison with existing methods in the literature. If new model architectures were added, dozens more models would have to be trained, and hundreds more experiments would need to be run. We believe that the breadth of experiments in this work is sufficient for proving the main claims.
>
> **Representation Learning.** Prior work in time series analysis primarily patches input time series, then feeds the patches through embedding layers that are trained end-to-end with a downstream model \[1, 2, 6, 7, 8\]. Prior work in audio and vision domains leverage vector quantized approaches, commonly VQ-VAEs, with convolutional encoders and decoders \[3, 4, 5\]. Given prior work, we adopt the VQ-VAE as our discrete tokenizer, and the patch \+ embedding layers as a natural ablation. These choices enable ease of comparison to prior work and evaluation of VQ-VAE-based tokens for general time series analysis.
>
> **Downstream Modeling.** Forecasting is the only task that requires a downstream model. Prior work primarily leverages MLPs \[7, 8\] and transformers \[1, 2\]. Again, for ease of comparison, we adopt these downstream model architectures for our forecasting models, as our goal is not to assess which architecture is best, but whether or not discrete tokens boost the performance of existing methods when compared to patch-based representations.

---

> > ### Author Response · Authors · 2024-10-03
> > **Response to Reveiwer FqZp 2/2**
> >
> > ### Requested Changes
> >
> > **VQ-VAEs and the Straight-Through Estimator.** We use a VQ-VAE because vector quantized approaches have been successful in the audio and vision domains \[3, 4, 5\] and can learn discrete representations independently of a downstream model in a self-supervised fashion, similar to tokenization in NLP (see the 2nd to last paragraph on Page 3). As such, they are a natural architecture to consider for large, generalist time series models.
> >
> > The VQ-VAE forward pass employs a non-differentiable discrete codebook lookup through which you cannot backpropagate. This motivates the straight-through estimator \[9\]; the VQ-VAE approximates the gradients in a similar fashion by copying the gradients from the decoder input to the encoder output during training \[3\], which is the method we adopt.
> >
> > We will clarify the role of the VQ-VAE in the main text.
> >
> > &nbsp;
> >
> > **More Experiments.** See the discussion under “Weakness 1.” We believe that the diversity of our domains and breadth of our experiments are quite strong, and the addition of new domains would lead to a prohibitive increase in the number of models to be trained and experiments to be run. If you have more specific concerns regarding data diversity, we are happy to discuss them.
> >
> > &nbsp;
> >
> > \[1\] Zhou, Tian, et al. "One fits all: Power general time series analysis by pretrained lm." *Advances in neural information processing systems* 36 (2023): 43322-43355.
> >
> > \[2\] Liu, Yong, et al. "itransformer: Inverted transformers are effective for time series forecasting." *arXiv preprint arXiv:2310.06625* (2023).
> >
> > \[3\] Aaron Van Den Oord, Oriol Vinyals, et al. Neural discrete representation learning. Advances in neural information processing systems, 30, 2017\.
> >
> > \[4\] Patrick Esser, Robin Rombach, and Bjorn Ommer. Taming transformers for high-resolution image synthesis. In Proceedings of the IEEE/CVF conference on computer vision and pattern recognition, pp. 12873–12883, 2021\.
> >
> > \[5\] Robin Rombach, Andreas Blattmann, Dominik Lorenz, Patrick Esser, and Björn Ommer. High-resolution image synthesis with latent diffusion models. In Proceedings of the IEEE/CVF conference on computer vision and pattern recognition, pp. 10684–10695, 2022\.
> >
> > \[6\] Wu, Haixu, et al. "Timesnet: Temporal 2d-variation modeling for general time series analysis." *arXiv preprint arXiv:2210.02186* (2022).
> >
> > \[7\] Das, Abhimanyu, et al. "Long-term forecasting with tide: Time-series dense encoder." *arXiv preprint arXiv:2304.08424* (2023).
> >
> > \[8\] Ailing Zeng, Muxi Chen, Lei Zhang, and Qiang Xu. Are transformers effective for time series forecasting? Proceedings of the AAAI conference on artificial intelligence, 2023\.
> >
> > \[9\] Yoshua Bengio, Nicholas Léonard, and Aaron Courville. Estimating or propagating gradients through stochastic neurons for conditional computation. arXiv preprint arXiv:1308.3432, 2013\.

---

> > > ### Author Response · Authors · 2024-10-14
> > > **Summary of Updates**
> > >
> > > Dear Reviewer FqZp,
> > >
> > > &nbsp;
> > >
> > > As the end of the review period approaches, we would greatly appreciate it if you could confirm whether our response has adequately addressed your concerns.
> > >
> > > &nbsp;
> > >
> > > Given our previous discussion, below is a summary of updates we will make to the paper. We will:
> > > 1. add the domain diversity table to the appendix
> > > 2. clarify the role of the VQ-VAE in the text
> > >
> > > &nbsp;
> > >
> > > If any questions remain, please let us know, and we will do our best to respond within the remaining time. Otherwise, we kindly thank you for your valuable feedback as it has improved our work.

---

### Author Response · Authors · 2024-09-30
**Response to Reviewer JMhz**

We thank the reviewer for their helpful comments. We will address weaknesses and requested changes in order.

&nbsp;

**W1.** While we agree that capturing fine-grained temporal features is important, we believe that this concern is sufficiently addressed in the paper.

1. **Low token reconstruction error (Fig. 9, left) implies that TOTEM captures fine-grained temporal information.** If the tokens did not capture these details up to the resolution of the training data, then the reconstruction error would be much higher.

2. **Across all tasks, models trained on tokenized representations outperform those trained on patches.** This implies that the full temporal resolution provided by patch representations does not benefit task performance more than the compression offered by tokenized representations.

3. **This performance gap holds over a diverse range of temporal resolutions.** The datasets studied in this work exhibit temporal resolutions as fine as 1/0.002Hz and as coarse as once per day (Table 3, Appendix A1). Further, this holds even over the datasets used in the zero-shot trials, which implies that TOTEM helps, not hurts, for capturing high-resolution temporal features.

&nbsp;

**W2:** Below are comparisons of both training time and model size between the TOTEM and GPT2 generalists demonstrating that TOTEM is more computationally efficient on both fronts. We are happy to add this new analysis to our work.

|  | TOTEM (Ours) Params  | TOTEM (Ours) Training Time on 1 A100 | GPT2 Params | GPT2 Training Time on 1 A100 |
| :---- | :---- | :---- | :---- | :---- |
| Imputation | **\~345,000** | **\~1.5 hours** | \~60,700,000 | Several days |
| Anomaly Detection | **\~345,000** | **\~1.5 hours** | \~46,500,000 | Several days |
| Forecasting | **\~345,000 for VQVAE, ~1,600,000 for downstream transformer**  | **\~1.5 hours for VQVAE, ~ a day for transformer** | \~89,000,000 | Several days |

&nbsp;

**RQ1.** We are happy to add a sentence to paragraph 2 of the Introduction that expands upon the tasks. We will also consolidate the existing discussion in the Related Work on 'Time Series Tasks' to clarify this earlier in the paper.

&nbsp;

**RQ2.** See above response.

&nbsp;

**RQ3.** Thank you for the suggestion - we are working on alternatives for visualizing the data. Due to the volume of results we thought the bar chart summary and coloring the tables would be helpful. We could move the imputation tables to the Appendix (like they are for anomaly detection and forecasting) and increase the white space; were the bar charts a helpful visualization? We would be happy to incorporate any suggestions into the visualization if you have any specific thoughts.

---

### Decision · Action_Editor_s4gx · 2024-11-12

**Recommendation:** Accept as is

**Comment:**

The paper was reviewed by three expert reviewers. The reviewers initially raised concerns about the  diversity of the explored domains, the information loss due to discretization and the significance of the reported results. These concerns were addressed by the authors in their response, and two reviewers recommended acceptance of the paper, while one reviewer recommended weak acceptance. I thus think that the paper is now ready for publication.

To further strengthen the paper, I would encourage the authors to discuss the following:
- Language models rely on the structure of language which is shared across all corpora on which these models are trained. However, time series data collected from different domains can be very diverse with different patterns occurring in the different series. Why are those models expected to learn patterns useful for other time series domains?
- Would models that do not produce discrete data representations, but are trained on many data domains simultaneously also achieve strong zero-shot performance and why?

I also request the authors to fix the issue with the parenthetical citations raised by Reviewer nhYG.

**Audience:**

The findings of this paper will be of interest to some individuals in TMLR's audience.

**Claims And Evidence:**

This paper presents TOTEM, a time series model which leverages discrete data representations and can be trained on many data domains simultaneously. The main claim made in the submission is that TOTEM can achieve strong zero-shot performance. The claim is indeed supported by extensive experimental results which demonstrate its strong zero-shot capabilities.

---

> ### Author Response · Authors · 2024-12-16
> **Thank you for the engaging discussion**
>
> Thank you for your comments - we have updated the paper accordingly. To briefly address your two points:
>
> - We believe that time series data collected across different domains do share underlying structure which our generalist models exploit. In the “Codebook Visualizations” section in the Appendix, we can see that the generalist codebook is distinct from three of the specialist codebooks, but still retains codes present in each specialist. This implies that there are waveforms (which we explicitly tokenize) that are common across domains, which may help explain TOTEM's strong performance.
>
> - The GPT2 generalist baseline that we train does not utilize discrete data representations but is trained on many data domains simultaneously. Our experiments show that TOTEM outperforms GPT2 despite its smaller parameter count on all tasks and for both the in-domain and zero-shot testing regimes.
>
> We thank you and the reviewers for the many productive comments and discussions, which greatly improved the quality of our work.